# Black carbon and mineral dust in snow cover on the Tibetan Plateau

Yulan Zhang[1], Shichang Kang[1,2*], Michael Sprenger[3], Zhiyuan Cong[2], Tanguang Gao[4], Chaoliu Li[2], Shu Tao[5], Xiaofei Li[1], Xinyue Zhong[1], Min Xu[1], Wenjun Meng[5], Bigyan Neupane[1], Xiang Qin[1], Mika Sillanpää[6]

[1]State key laboratory of Cryospheric Science, Northwest Institute of Eco-Environment and Resources, Chinese Academy of Sciences, Lanzhou730000, China
[2]CAS Center for Excellence in Tibetan Plateau Earth Sciences, Beijing 100101, China
[3]Institute for Atmospheric and Climate Science, ETH Zurich, CH-8092 Zurich, Switzerland
[4]Key Laboratory of Western China's Environmental System (Ministry of Education), College of Earth and Environmental Sciences, Lanzhou University, Lanzhou 730000, China
[5]Department of Environmental Science, Laboratory for Earth Surface Processes, Peking University, Beijing, China
[6]Laboratory of Green Chemistry, Lappeenranta University of Technology, Mikkeli 50130, Finland

*Correspondence to*: Prof. Shichang Kang (shichang.kang@lzb.ac.cn)

**Abstract.** Snow cover plays a key role for sustaining ecology and society in mountainous regions. Light-absorbing particulates (including black carbon, organic carbon, and mineral dust) deposited on snow can reduce surface albedo and contribute to the near-worldwide melting of snow and ice. This study focused on understanding the role of black carbon and other water-insoluble light-absorbing particulates in the snow cover of the Tibetan Plateau (TP). The results found that the black carbon, organic carbon, and dust concentrations in snow cover generally ranged from 202–17,468 ng g$^{-1}$, 491–13,880 ng g$^{-1}$, and 22–846 μg g$^{-1}$, respectively, with higher concentrations in the central to northern areas of the TP. Back trajectory analysis suggested that the northern TP was influenced mainly by air masses from Central Asia with some Euro-Asia influence, and air masses in the central and Himalayan region originated mainly from Central and South Asia. The relative biomass burning sourced black carbon contributions decreased from ~50 % in the southern TP to ~30 % in the northern TP. The relative contribution of black carbon and dust to snow albedo reduction reached approximately 37 % and 15 %, respectively. The effect of black carbon and dust reduced the snow cover duration by 3.1±0.1 days to 4.4±0.2 days. Meanwhile, the black carbon and dust had important implications for snowmelt water loss over the TP. The findings indicate that the impacts of black carbon and mineral dust need to be properly accounted for in future regional climate projections, particularly in the high-altitude cryosphere.

## 1 Introduction

Black carbon (BC), organic carbon (OC), and mineral dust (hereafter: dust) are the main constituents of light-absorbing particulates (LAPs) in snow (Andreae and Gelencsér, 2006; Bond et al., 2013; Di Mauro et al., 2015; Doherty et al., 2010; Painter et al., 2010). LAPs can reduce snow and ice albedo and trigger positive feedback that contributes to the near-worldwide melting of snow and ice (Dumont et al., 2014; Flanner et al., 2007; Hadley and Kirchstetter, 2012; Hansen and Nazarenko, 2004; Painter et al., 2013; Zhang et al., 2017). Atmospheric BC is a distinct type of carbonaceous material from incomplete

combustion of biomass/biofuel and fossil fuel. A large fraction of BC is emitted from anthropogenic activities (Bond et al., 2013). Because BC can absorb solar radiation, influence cloud processes, and alter the melting of snow cover and glaciers, it has been considered to be the second most important climate forcer in the Earth's climate system only after carbon dioxide (Andreae and Ramanathan, 2013; Bond et al., 2013; Ramanathan and Carmichael, 2008). Model studies in recent decades

indicated that BC in seasonal snow may be responsible for a significant impact on warming and albedo reduction over the Arctic (Flanner, 2013; Qian et al., 2014), Greenland (Dumont et al., 2014), the Alps (Painter et al., 2013; Yasunari et al., 2015), and the Tibetan Plateau (TP) (He et al., 2014; Jacobi et al., 2015; Ji et al., 2015; Ménégoz et al., 2014). The snow albedo feedback of BC may also be an important driver accelerating glacial retreat and snow cover melt (Painter et al., 2013; Skiles et al., 2015; Xu et al., 2009).

Light-absorbing OC in snow can contribute significantly to the absorption of ultraviolet/visible light (Andreae and Gelencsér, 2006; Yan et al., 2016). Bahadur et al. (2012) studied aerosol optical properties derived from the Aerosol Robotic Network (AERONET) measurements and estimated that, in California, the OC absorption at wavelength of 440 nm was approximately 40 % of the BC absorption. Measurements of hundreds of snow samples revealed that approximately 40 % of the light absorption from impurities in Arctic snow and sea ice is due to non-BC constituents (Doherty et al., 2010). In western North

America, Dang and Hegg (2014) suggested that OC was responsible for more than 10 % of the total light absorption. A global model by Lin et al. (2014) estimated the role of OC in the reduction of snow albedo and suggested that the radiative forcing (RF, it means the difference between insolation (sunlight) absorbed by the snow surface and energy radiated back to space in this study.) of OC deposited to land snow and sea ice ranges from +0.0011 to +0.0031 W m$^{-2}$, contributing as much as 24 % of the forcing caused by BC in snow and ice. The RF of snow pit dissolved organic carbon (DOC) from a Tibetan glacier was

calculated to be 0.43 W m$^{-2}$, indicating that DOC in snow needs to be taken into consideration for the accelerating glacial melt on the TP (Yan et al., 2016).

Mineral dust, another important LAP component in snow cover, can also change the cryospheric environment and hydrological cycle due to its light-absorbing properties (Di Mauro et al., 2015; Ji et al., 2016; Painter et al., 2010, 2013; Wu et al., 2016; Zhang et al., 2015a). In the European Alps, observed reflectance provided evidence that seasonal input of dust can strongly

decrease the spectral properties of snow, especially for wavelength of 350 to 600 nm (Di Mauro et al., 2015, 2017). In a study in the Swiss Alps, Gabbi et al. (2015) showed that mineral dust lowered the mean annual albedo by 0.006−0.011, depending on the location on the glacier, causing a reduction of approximately 142–271 mm water equivalent (mm w.e.) in annual mass loss. In the San Juan Mountains of the USA, snow cover duration in a seasonal snow-covered region was found to be shortened by 18 to 35 days during ablation period due to surface RF caused by deposition of disturbed desert dust (Painter et al., 2007).

In the upper Colorado River Basin, the daily spring dust RF ranged from 30−65 W m$^{-2}$, advancing melt by 15−49 days (Skiles et al., 2015). A study of the Mera glacier on the southern slope of the Himalayas indicated that, when dust concentrations were high, dust dominates absorption, snow albedo reduction, and RF, and the impact of BC may be negligible (Kaspari et al., 2014). The presence of dust in snow suggests it also plays an important role in darkening the glacier surface.

As an important component of the hydrology, snow cover plays a key role for sustaining ecology and society in mountainous regions (Beniston et al., 2017). Snowpack changes are not only crucial for estimating impacts on water resources but also important to anticipate natural hazards related to snow avalanches/disasters (Beniston et al., 2017; Brown and Mote, 2009). Covered by a large volume of snow/ice in the mid-low latitudes, the TP is considered to be a sensitive and readily visible indicator of climate change (Yao et al., 2012a). Recently, the TP cryosphere has undergone rapid changes (Kang et al., 2010), including glacial shrinkage (Xu et al., 2009; Yao et al., 2012b), permafrost degradation (Cheng and Wu, 2007), and reduction in the annual duration of snow cover days (Ménégoz et al., 2014; Xu et al., 2017). Studies on the TP indicate that BC has been a significant contributing factor to the observed cryospheric change (Li et al., 2017; Ming et al., 2009; Niu et al., 2017; Xu et al., 2009;Yang et al., 2015; Zhang et al., 2017) and a major forcer of climate change (Qian et al., 2015; Ji, 2016). Observations and simulations also found that dust deposition on snow/ice can change the surface albedo and perturb the surface radiation balance (Ji, 2016; Qu et al., 2014), resulting a decrease of 5–25 mm snow water equivalent over the western TP and Himalayas (Ji, 2016).

Changes in snow cover over the TP have attracted much attention in recent years owing to climate change (Xiao and Duan, 2016; Xu et al., 2017). Snow cover on the TP plays an important role in the Asian summer monsoon, and serves as a crucial water source for several major Asian rivers (Bai and Feng, 1994; Lau et al., 2010; Vernekar et al., 1995; Yao et al., 2012a; Zhao and Moore, 2004). Atmospheric heating related to BC and dust deposition can lead to widespread enhanced warming over the TP and accelerate snowmelt in the western TP and Himalayas (Lau et al., 2010; Ramanathan and Carmichael, 2008; Zhang et al., 2015b). Simulation studies of BC in snow over the TP have inherent uncertainties due to the physics/chemistry/transport scheme used in the models (Gertler et al., 2016; Yasunari et al., 2010, 2013). For example, Kopacz et al. (2011) estimated RF of 5 to 15 W m$^{-2}$ due to BC within the snow-covered areas of Himalaya and the TP whereas Flanner et al. (2007) and Qian et al. (2011) estimated peak values of BC effects exceeding 20 W m$^{-2}$ for some parts of the TP. Menon et al. (2010) and Ménégoz et al. (2014) proposed that BC in snow caused a significant part of the decrease of the snow cover extent or duration observed on the TP during the last decade. Ji (2016b) found a positive surface RF was induced by dust, which caused a decrease of 5–25 mm w.e. over the western TP, Himalayas, and Pamir Mountains from December to May. Large-area observations of BC data in snow cover, which are still sparse in the TP, can be useful for model evaluation and calibration in the future.

To further understand the role of BC and dust in TP snow cover, more measurements and an extended LAP dataset of snow cover is needed to study their effects on cryospheric change in future of this region. This study is an attempt to fill this gap. Snow samples were collected across the TP in winter seasons (Fig. 1, Table S1), which will provide the first large-area survey of BC and dust in seasonal snow cover over the TP. We further investigate albedo reduction and RF, and estimate changes of snow cover duration days caused by observed LAPs. We use back trajectory analysis coupled with BC fire emission inventories to approximate natural/anthropogenic contributions. The results of this study will thus increase and broaden our understanding of BC and dust in snow cover across the TP, enable improved climate modeling, and inform for mitigation actions around the Tibetan cryospheric region.

## 2 Methodology

### 2.1 Study area

The TP covers an area of more than 2.5 million km$^2$ with an average elevation greater than 4000 m a.s.l. (Yao et al., 2012a). The TP exerts profound influences on atmospheric circulation and climate through mechanical and thermal effects because of

its large area, unique topography, and geographical location (Yanai and Wu, 2006). Extended glacial coverage (over 100,000 km$^2$) and snow cover have the potential to modify regional hydrology and trigger natural hazards that can impact local populations, agriculture, and water resources in and around the region (e.g., Immerzeel et al., 2010; Xu et al., 2009; Yao et al., 2012a).

Moderate Resolution Imaging Spectroradiometer (MODIS) data from 2000–2006 showed that the spatial distribution of snow

cover is quite variable over the TP because of the complex terrain (Pu et al., 2007). Maximum snow accumulation and melting times vary over the year but are generally later as the elevation increases. Additionally, large interannual variabilities occur in the late autumn and winter months. Heating by light-absorbing aerosols produces an atmospheric dynamic feedback, which increases moisture, cloud cover, and deep convection over northern India, and enhances the rate of snow melt in the Himalayas and the TP (Lau et al., 2010; Ramanathan and Carmichael, 2008). The accelerated melting of snow is confined mostly to the

western TP, beginning slowly in early April and then progressing more rapidly from early to mid-May; the snow cover remains reduced from mid-May through early June (Lau et al., 2010). The decadal change in the spring snow depth over the TP can impact the Asian summer monsoon and a close relationship exists between the interdecadal increase of snow depth over the TP during March–April, increased summer rainfall over the Yangtze River valley, and a dryer summer along the southeast coast of China and the Indochina peninsula (Zhang et al., 2004). Based on observations from 103 meteorological stations

across the TP, the duration of snow cover exhibited a significant decreasing trend (1.2 days per decade), which was jointly controlled by a later snow starting time (1.6±0.8 days per decade) and an earlier snow ending time (−1.9±0.8 days per decade) consistent with a response to climate change (Xu et al., 2017).

### 2.2 Sample collection

For this study, snow samples were collected from 37 sites (Fig. 1a) and one glacial river basin (Laohugou (LHG) region,

including 10 sites as in Fig. 1b) across the TP in December 2014 and November 2015 (Table S1). As shown in Figure 1a, these field campaigns can be grouped into three regions (Region I, II, and III), as was done in previous studies by $\delta^{18}O$ in precipitation (Yao et al., 2013). These three distinct domains were associated with Indian monsoon (Southern TP), westerlies (Northern TP), and transition in between (Yao et al., 2013). In this study, Region I (Southern TP) includes 27 sites and is mainly affected by the Indian monsoon. Region II (Central TP) includes 10 sites and is affected by both the westerlies and the Indian monsoon.

In Region III dominated by westerlies, a total of 34 seasonal snow samples were collected from the LHG glacierized region of Qilian Mountain from December 2015 to June 2016.

To minimize the effects of local sources, we selected sampling sites more than 1 km away from roads and usually 30 km from villages and cities. Generally, snow samples were collected from the top 5−10 cm with cleaned stainless-steel utensils, and placed in Whirl-Pak® bags (Fig. S1). These snow samples were kept frozen until they underwent filtration. During sampling, snow depth, snow density and grain size were also observed. Snow depth was measured using a ruler, and snow density was

measuring by weighing a specific volume of snow (Table S1). For the snow grain size observation, we sprayed the snow grains on MIG paper and took a photo using a portable digital microscope (Anyty 3R-MSV500) to calculate the major and minor axis (Fig. S2). Based on field measurements from previous studies (Mugnai and Wiscombe, 1980; Grenfell and Warren, 1999), two assumptions were proposed: 1) The snow grain particle is an inequilateral spheroid, and 2) the major axis (a) and minor axis (b) of the inequilateral spheroid are denoted by a, and b, respectively. In such way, the relationship a=2b exists (Hao,

2009). According to Kokhanovsky and Zege (2004), the effective snow grain radius ($R_{ef}$) can be calculated as equal to the radius of the volume-to-surface equivalent sphere by the following equation (1):

$$R_{ef} = \frac{3\bar{V}}{4\bar{A}} \tag{1}$$

Where $\bar{V} = \frac{4}{3}\pi r^3$ and $\bar{A} = \pi r^2$, are the average volume and the average cross-section (geometric shadow) area of the snow grains, respectively. In addition, $r$ is the radius of geometric optics, $r = \frac{a+b}{4}$.

Thus, $R_{ef} \approx 0.35a \tag{2}$

## 2.3 LAP measurements

In the laboratory, the snow samples were melted rapidly in a hot-water bath to minimize the melt time. Immediately after melting, the snow samples were filtered through pre-baked quartz filters (Whatman[TM]) using an electric pump to create a partial vacuum. Then, sample meltwater was saved in polyethylene bottles, and refrozen for chemical analysis.

Because the mineral dust mass (unit, μg g[-1]) is much larger (about 2−3 orders of magnitude) than the BC and OC mass (unit, ng g[-1]) in the snow samples compared to aerosol samples (Wang et al., 2012; Zhang et al., 2016), dust concentrations were calculated by taking the differences in the weight of the quartz filter before and after filtration. Quartz filters were then used to analyze BC and OC with a thermal-optical carbon analyzer (DRI 2001A model, Desert Research Institute, NV, USA) following a methodology (IMPROVE_A protocol) used in previous studies (Cao et al., 2003; Chow et al., 2004; Xu et al.,

2009; Yang et al., 2015). Its function relies on the fact that OC components can be volatilized from the sample and deposited in a non-oxidizing helium atmosphere, in which BC can be combusted with an oxidizer. The sample punch was placed in a quartz holder oriented in the direction of the carrier gas flow, and heated gradually to a specific temperature plateau (typically 140, 280, 480, 550 ℃) in a pure helium atmosphere, which results in OC concentrations from the sample. Due to the influence of dust on the quantitative analysis of BC and OC in snow when using a thermal optical method (Wang et al., 2012), we limited

the initial temperature plateau (550 ℃) to reduce the time that BC was exposed to a catalyzing atmosphere (Yang et al., 2015). The sample was reheated further in a stepwise fashion to near 840 ℃ in an oxygen-containing atmosphere (usually 2 %

oxygen and 98 % helium) to burn out all remaining BC. The carbon released from each temperature plateau is converted to methane by passing the flow through a methanator (firebrick impregnated with nickel catalyst at ~550 ℃ in a stream of hydrogen) and measured by a flame ionization detector (FID) (Chow et al., 2004). Under ideal conditions, the minimum detection limit of total carbon is 0.93 μg C cm$^{-2}$, with a range of 0.2−750 μg C cm$^{-2}$. During the experiments, the filter blank (detected by blank filters) was 1.23±0.38 μg C cm$^{-2}$.

To identify uncertainty stemming from instrumental instability and the uneven distribution of carbon particles in the filters, duplicates of ~30 % of the samples were analyzed separately, with consistent results (BC r$^2$=0.95, OC r$^2$=0.87, dust r$^2$=0.90). Because we analyzed the filters through which the melted snow samples passed, the OC measured in our study was water-insoluble OC.

## 2.4 Back trajectory analysis

Calculation of air parcel trajectories is a widely used approach for determining source regions of various atmospheric species. Conceptually, back trajectory analysis was considered as changes in concentration at the receptor site can be qualitatively attributed to different upwind source areas along the backward trajectories (Skiles et al., 2015). To determine the potential origins of the LAPs deposited on the snow cover of the TP, back trajectory analysis were performed using the European Centre for Medium-Range Weather Forecasts (ECMWF) analysis fields with the Lagrangian analysis tool LAGRANTO (Sprenger and Wernli, 2015), launched every six hours for six selected sampling sites as receptors (including three sites of MYL, NMC, and SETP in region I, two sites of TGL and NETP in region II, and LHG in region III) during the winter season. The ECMWF fields (horizontal and vertical wind components) were retrieved on 137 model levels and then interpolated onto a 0.25 °×0.25 ° latitude-longitude grid. Trajectory starting positions can be defined easily and flexibly based on different geometrical and meteorological conditions; after the computation of the trajectories, a versatile selection is offered based on single or combined criteria (Sprenger and Wernli, 2015). First, starting positions are initialized with a suitable domain over the TP at 12:00 UTC on November 1, 2015 (or 2014). For this case, a domain from 0 to 75 °N and 60 °E to 120 °E was chosen. We choose starting positions in this domain that are horizontally equidistant with 80 km horizontal spacing and extend vertically from 1030 to 790 hPa with 30 hPa vertical spacing. The trajectories are calculated from all starting positions 96 h backward in time. Finally, biomass burning emission data is traced along the calculated trajectories to estimate whether an air parcel at the receptor site is influenced by BC fire emissions or not. In this study, the Fire INventory from NCAR (FINN) v1.5 global fire emissions flux in 2012−2014, speciated with the GEOS-chem mechanism, was used to estimate contributions of BC fire emission at the six selected receptor sites. FINN emission estimates are based on the framework described by Wiedinmyer et al. (2011). FINN used satellite observations of active fires and land cover, together with emission factors and estimated fuel loadings to provide daily, highly resolved (1 km) open biomass burning (includes wildfire, agricultural fires, and prescribed burning, and dose not included biofuel use and trash burning) emissions estimates for use in regional and global chemical transport models. The same calculation was performed for the Eclipse V5 inventory for the anthropogenic contributions of BC. The Eclipse V5 inventory was widely used in the simulations. The historical data for the period 1990−2010 were revised compared to preceding

sets using the latest IAE (the International Energy Agency) and FAO (the Food and Agriculture Organization) statistics extending to 2010, as well as recent country reporting where available. Note that this analysis is qualitative and does not take into account loss from wet and dry deposition. We also assumed that the anthropogenic emissions have not changed significantly from 2010 compared to the period from 2012−2014. Despite these uncertainties, the relative differences between the BC contributions are used to provide information on regional differences in this study.

## 2.5 Reflectance measurements and albedo simulations

For the selected sites, a general-purpose spectroradiometer (Analytical Spectral Devices (ASD), FieldSpec 4, Inc.) was used to measure the reflectance of snow cover. The ASD FieldSpec 4 instrument is a general-purpose spectroradiometer that is useful for applications requiring the measurement of reflectance, transmittance, radiance, or irradiance. The instrument is specifically designed for field environment remote sensing to acquire visible and near-infrared and shortwave infrared spectra. It has 3 nm spectral resolution on the visible/near infrared detector (wavelength of 350–1050 nm, silicon photodiode array), and 10–12 nm resolution on the shortwave infrared detectors (wavelength of 900–2 500 nm, InGaAs). In the field, reflectance was measured at two sites (24K and MD in Table S1) with FieldSpec 4 under clear-sky conditions. These measurements of reflectance were calculated using the standard solar irradiance, which were then used for comparison with simulated albedos. Albedo reduction caused by LAPs is calculated by the SNow ICe and Aerosol Radiation (SNICAR) model (Flanner et al., 2007), which utilizes the two-stream radiative transfer solution of Toon et al. (1989) and has been widely used in Arctic snow (e.g., Flanner et al., 2007, 2009; Hadley and Kirchstetter, 2012). This simulator provides hemispheric reflectance of snow for unique combinations of impurity content (black carbon, dust, and volcanic ash), snow grain size and incident solar flux characteristics. The input fields include incident radiation, solar zenith angle, surface spectral distribution, effective snow grain radius, snow cover thickness and density, albedo of the underlying ground and concentrations of impurities. Based on our observations, the effective snow grain radius ranged from approximately 100 to 1500 μm for different snow samples based on calculation of Eq. (2), and snow density ranged from 150 to 400 kg m$^{-3}$. These two factors varied in low, medium and high scenarios in the model runs (Table 1).

In terms of the albedo calculation, RF due to BC and dust can be obtained by using Eq. (3) (Kaspari et al., 2014; Yang et al., 2015):

$$\text{RF} = \sum_{0.325\,\mu m}^{2.505\,\mu m} E(\lambda, \theta)\,(\alpha_{(r,\lambda)} - \alpha_{(r,\lambda,imp)})\Delta\lambda \tag{3}$$

Where α is the modeled snow albedo with or without the impurities (imp) of BC and/or dust; E is the spectral irradiance (W m$^{-2}$); r is the snow optical grain size (μm); λ is wavelength (μm); and θ is the solar zenith angle for irradiance ($^\circ$).

In general, the snow cover is vertically inhomogeneous, and sometimes it is optically thin (Voisin et al., 2012). The stratified structure of snow cover will lead to the discrepancies between computed spectral albedo from semi-infinite snow cover and measured albedo (Aoki et al., 2000; Grenfell, 1994; Kuipers Munneke et al., 2008; Zhou et al., 2003). In this study, we usually collected snow samples from the top 5−10 cm. Because the SNICAR model online is a single-layer implementation model

(Flanner et al., 2007, 2009), the BC concentration input is necessarily uniform with depth. The error in doing so depends entirely on the environmental conditions, for example, under strong melt where hydrophobic impurities accumulated near the top, the concentrations can vary strongly with depth.

## 2.6 Estimates of changes in snow cover duration

To estimate snow melt due to BC and dust, a method was constructed in which the absorptivity of the snow was multiplied by the daily average incoming shortwave radiation from the automatic weather stations (AWS) set up at the meteorological stations near the sampling sites (Fig. S3) (Schmale et al., 2017). Factors derived from meteorological stations in the adjacent areas included daily temperature and daily solar shortwave radiation. Average albedo reductions for snow cover by BC and dust were derived from the SNICAR model. To calculate the amount of snow melted based on the enthalpy of fusion of water

(334 J g$^{-1}$), the snow cover was assumed to be at 0 ℃ and then snowmelt (Melt$_{snow}$, mm w.e.) was calculated by using Eq. (4) (Schmale et al., 2017):

$$\text{Melt}_{snow} = \text{N}_{Tht0} \times \Delta\alpha \times SW \tag{4}$$

where N$_{Tht0}$ is the number of days with temperature greater than 0 ℃; $\Delta\alpha$ is the albedo reduction caused by BC and dust in snow, and by different snow grain sizes due to snow aging for clean snow; and $SW$ is the shortwave radiation (W m$^{-2}$). From

Eq. (4), we can estimate changes of snow cover duration ($\Delta$D, days) based on the observed snow depth (mm w.e.) during sampling (Table S1) and previous studies (Che et al., 2012; Xu et al., 2017; Zhong et al., 2016). We expressed the results for each case as low, medium, and high scenarios, based on albedo variations.

## 3 Results and discussion

### 3.1 Distributions of LAPs

Across the TP, most of the samples we took were old wind-packed snow. Generally, the ground layer of snow was usually depth hoar, similar to that in Arctic snow (Doherty et al., 2010) and central North American snow (Doherty et al., 2014, 2016). For many of these sampling sites, the snow cover was intermittent (Fig. S1). During the winter season, surface soil can be carried aloft by strong winds and deposited on snow surface, which affects the concentrations of LAPs (Wang et al., 2013). Our results indicated that the spatial distributions of BC, OC, and dust in snow over the TP generally ranged from 202–17,468

ng g$^{-1}$, 491–13,880 ng g$^{-1}$, and 22–846 μg g$^{-1}$, respectively (Fig. 2). Variations in the concentrations of LAPs were higher in the central to northern TP (Region II and III) than in the southern TP (Region I) (Table 2). The highest concentrations may be related to the lower snow depth and dirtier snow that resulted from post-deposition processes (e.g., snow at some sites was melting). In Region I (southern TP), the snow was cleaner and had lower concentrations of LAPs comparable to concentrations in the old snow of glaciers (Kaspari et al., 2014).

The previous studies also examined concentrations of LAPs in snow from central North America (Doherty et al., 2014, 2016; Skiles and Painter, 2016; Zatko et al., 2016), northern China (Huang et al., 2011; Wang et al., 2013; Ye et al., 2012), Japan

(Kuchiki et al., 2015), Europe (Chýlek et al., 1987), the Arctic (Doherty et al., 2010; Forsström et al., 2013), Greenland (Aoki et al., 2014; Zatko et al., 2013), and Antarctic (Warren and Clark, 1990; Zatko et al., 2013). Concentrations of BC and other water-insoluble LAPs from the TP were mostly similar to those in the snow from northern China (Wang et al., 2013), but higher than those from the Himalayas (Lim et al., 2014) (Table 2). In comparison (Table 2), LAPs in snow of the TP showed

larger concentrations which can be attributed to more deposition from nearby regions around the TP (e.g., South Asia, East Asia, and/or western China) (Lu et al., 2012; Ramanathan and Carmichael, 2008), or to impacts from the soil in vicinity to the sampling sites.

Ratios of OC to BC (OC/BC) were used to examine the possible impact of biomass burning in previous studies (Watson et al., 2001; Bond et al., 2013; Cong et al., 2015a, 2015b). Usually, the aerosols emitted from biomass burning have higher OC/BC

ratios than that from fossil fuels combustion. For example, Watson et al. (2001) reported an OC/BC ratio of 14.5 for forest fires; whereas for fossil fuel, the OC/BC ratio was approximately 1. The OC/BC ratio in our sampled surface snow ranged from 0.64 to 3.31 over the TP, generally decreasing from south to north (Fig. 2d). The average ratios for Regions I, II, and III were 1.82, 1.31, and 1.14, respectively (Table 2), indicating a decreasing impact of biomass-sourced aerosol deposition in snow over the TP. The slightly higher OC/BC in region I may be due to its proximity to South Asian combustion sources

dominated by biomass burning (Cong et al., 2015a, 2015b; Li et al., 2016). For example, in the LHG region, lower LAP concentrations in fresh clean snow were observed at LHG3 and LHG6 (Fig. 1b); whereas near the glacier at LHG1 and LHG2, the collected wind packed, aged snow samples were quite dirty, and had higher LAP concentrations (Fig. 3a, b, and c). Meanwhile, OC/BC ratios were lower at sites near the glacier (Fig. 3d), implying predominantly non-biomass sourced combustion. Higher ratios of OC/BC were found at the sites near human settlements (LHG7–10) (Fig. 3d), where there was

extensive biofuel combustion (e.g., yak dung or straw burning). Because of the variations in sampling periods, snow types and depths, sources of dust and carbonaceous aerosol deposition, and/or strong winds, these results were expected to vary with large uncertainties. In general, however, anthropogenic activities play an important role in the ratios of OC/BC in snow cover. Figure 4 shows the back trajectory analysis for selected sites across the TP in winter. The northern TP sites (LHG and NETP) are influenced mainly by air masses from Central Asia and partly by air masses from Euro-Asia, which are similar to

the sources of dust reaching the TP (e.g., Zhang et al., 2016). These results are also consistent with those of He et al. (2014), Kopacz et al. (2011), and Lu et al. (2012), who modeled contributions of BC emissions in the same region. In the central (TGL and NMC) and southern region (MYL and SETP) of the TP, the air masses originate mainly from Central and South Asia. Anthropogenic BC emissions dominate the BC snow albedo as shown in previous studies (Li et al., 2016; Lu et al., 2012). Contributions of BC from wildfire, agricultural fires, and prescribed burning decrease from the southern to the

northern TP (Fig. 5). In the Himalayan region, BC from biomass burning sources accounts for about 50 % of the BC deposition on snow cover, reflecting the proximity to large sources in South Asia. In the central TP, BC from biomass burning accounts for approximately 30 % of the total, less than the evidence from aerosols (Li et al., 2016), maybe indicating a lack of biofuel contributions from back trajectory analysis; whereas in the northern TP, anthropogenic BC emissions (more than 70 %) dominate the BC albedo reduction. Li et al. (2016), using the $\Delta^{14}C/\delta^{13}C$ compositions of BC isolated from

aerosols and snowpit samples in the TP, also found equal contributions from biomass and fossil fuel combustion in the Himalayan region (near the southern TP), whereas BC in the remote northern TP is predominantly derived from fossil fuel combustion (66±16 %). These differences can also be seen from the BC emissions that arrived at the selected sites based on BC inventory data for the year of 2013 (PKU-BC-2013, Wang et al., 2014) in Figure S3. The amount of BC deposition is

larger in the southern TP (MYL) than in the northern TP (NETP) due to the nearby BC sources in south Asia, known to be a regional hotspot of BC-induced atmospheric solar heating (Ramanathan and Carmichael, 2008); while more BC deposition in the central TP (NMC) is evident than in the NETP. This is plausible as pollutants from the southern side of the Himalayas can traverse the high mountain range not only through the major north-south river valleys but also by being lifted and advected over the Himalayas to reach to the inland TP (Cong et al., 2015b; Lüthi et al., 2015).

**3.2 Impacts of LAPs on albedo reduction and radiative forcing**

To estimate the albedo change and surface RF from BC and dust in snow cover, the SNICAR model was used (Flanner et al., 2007). Effective snow grain size and snow density was considered at the low, medium, and high scenarios (Table 1) and the SNICAR model was run for clear-sky conditions on the specific sampling days (Table S2).

The SNICAR model simulated albedo (caused by BC and dust with snow grain effective radius=150 μm) matches observed

reflectance in both the visible and absorptive near-infrared (Fig. 6). The BC strongly reduces visible reflectance (in particular from wavelength of 350 to 800 nm) but has negligible influence at wavelengths beyond 1 μm, since the impact of snow grain size is large in this wavelength range (Flanner et al., 2007; Gardner and Sharp, 2010). The presence of dust also decreases the albedo in the visible wavelength, which is in agreement with a study of the impact of dust on snow radiative properties conducted in the European Alps (Di Mauro et al., 2015). The difference between measured and simulated

reflectance by dust may be caused by the upper boundary of dust size in the SNICAR model (Flanner et al., 2007). The broadband albedo is comparable to albedo simulated for the wavelengths from wavelength of 350 to 2500 nm, with albedo differences less than 0.0058 (~2.3 %) (Table 3). For the same wavelengths (350–2500 nm), albedo differences between measurements and simulations is less than 0.0155 (~4 %) (Table 3).

For different scenarios defined by the effective snow grain size and snow density in Table 1, clean snow albedos resulting

from snow aging range from 0.5992 to 0.7751 (Table 4). If BC or dust is the only impurity in the snow cover, contributions to albedo reduction can reach approximately 37 % or 15 %, respectively, with associated larger RF due to BC. On average, BC and dust contribute to albedo reduction at approximately 0.18±0.01 and 0.06±0.004, respectively (Fig. 7). The total albedo reduction contributed by BC and dust is approximately 38 %, with the associated RF of 18–32 W m$^{-2}$ (Fig. 7). This highlights the importance of LAPs in snow.

Large uncertainties exist in the estimation of BC snow albedo forcing over the TP. Simulation results using the Community Atmosphere Model version 3.1 showed that the aerosol-induced snow albedo perturbations generate surface radiative forcing changes of 5–25 W m$^{-2}$ in spring (Qian et al., 2011). Using the SNICAR model coupled a general circulation model, Flanner et al. (2007) showed that, during some spring months, snow-only forcing exceeded 10 and 20 W m$^{-2}$ over parts of the TP. RF

in the snow-covered regions due to BC by the GEOS-Chem model induced the snow-albedo effect to vary from 5–15 W m$^{-2}$ over the TP, an order of magnitude larger than RF as a result of the direct effect (Kopacz et al., 2011). Recently, simulation results using a global chemical transport model in conjunction with a stochastic snow model and a radiative transfer model showed an annual mean BC snow albedo forcing to be 2.9 W m$^{-2}$ averaged over snow-covered plateau regions, which is a factor of three larger than the value over global land snow cover (He et al., 2014). Surface RF (simulated by a regional climate model coupled with an aerosol–snow/ice feedback module) induced by dust-in-snow showed positive values in the range of 0–6 W m$^{-2}$ over the interior of the TP, the Himalayas, Pamir, and Tienshan Mountains (Ji, 2016b). BC-in-snow/ice can cause positive surface RF (3.0–4.5 W m$^{-2}$) over the western TP in the monsoon season, with the maximum RF (5–6 W m$^{-2}$) simulated in the Himalayas and southeastern TP in the non-monsoon season (Ji et al., 2016). Limited measurements of LAPs in snow cover over the TP have hindered a more robust estimate of the forcing by BC and dust-in snow in this region (Bond et al., 2013; Qian et al., 2015). The mean forcing of BC and dust in snow in our study is slightly higher than previous results from snowpits (e.g., Ming et al., 2009) and comparable to some parts of the TP in spring (20 W$^{-2}$) (Flanner et al., 2007), which indicates that more frequent impurity measurements in terms of different years and locations over the TP are necessary for model assessment.

Light-absorbing OC in atmospheric aerosols has various origins, e.g., soil humics, humic-like substances (HULIS), tarry materials from combustion, bioaerosols, etc. (Andreae and Gelencsér, 2006). The chemical and optical properties of OC may differ due to the nature of the OC source or atmospheric processing (Chen and Bond, 2010). Contrarily, in comparison to BC and mineral dust, the optical properties of OC in snow/ice have hardly been examined. Due to the lack of reliable OC optical properties that span the dimensions of snow grain size and OC particle size, the SNICAR model currently does not support the calculation of OC-in-snow forcing in the same way as BC. Thus, this study doesn't incorporate the effect of OC on albedo reduction and RF.

### 3.3 Changes in snow cover duration days

Changes in snow cover duration were calculated using a model documented by Schmale et al. (2017) (section 2.6). In Eq. (4), monthly shortwave radiation (*SW*) were obtained from the Automatic Weather Station (AWS) near the snow sampling sites across the TP (Fig. S3). Table S3 shows monthly shortwave radiation data from AWS during the snow melting season (March-May) when the temperature began to increase. On average, *SW* in March, April, May, and June were approximately 238, 269, 292, and 271 W m$^{-2}$ respectively. *SW* in March showed the minimum value in Tanggula of the central TP (210 W m$^{-2}$, lower than the average), whereas in May it showed the maximum value in Nam Co (314 W m$^{-2}$, higher than the average). Based on these data during the melt season, three scenarios (220, 270, and 310 W m$^{-2}$) were defined as the minimum, median, and maximum scenarios of *SW* to estimate its impact on changes of snow cover duration. The average snow depth water equivalent (*SD*, mm w.e.) was assumed from previous studies in this region (Che et al., 2012; Xu et al., 2017; Zhong et al., 2016). In general, the number of reduction days in snow cover duration is larger for low *SD* scenarios than for medium and high scenarios (Fig. 8 and Table 5), particularly when *SW* is lower (220 W m$^{-2}$). If SD was 40 mm w.e., the average snow cover duration days

would be reduced by 1.26 ±0.05 days ($SW$=310 W m$^{-2}$) to 1.77 ±0.07 days ($SW$=220 W m$^{-2}$) by BC and dust (Fig. 8a, c, and e). If SD was 100 mm w.e., the average snow cover duration days would be shortened by 3.14 ±0.13 days to 4.43 ±0.18 days from the effect of BC and dust, with a maximum of approximately 9.4 days (Fig. 8b, d, and f), indicating advanced melt during ablation season. The result indicated that estimation of changes of snow cover duration days caused by the same level of LAPs was also affected by different $SW$ and $SD$. For different scenarios, dust contributes only approximately 1 day to the reduction of snow cover duration days (Fig. 8 and Table 5). These results imply that the reduction in snow cover duration related to LAPs is due mainly to BC. Note that no significant differences in reductions of snow cover duration were found across the TP (Fig. S4).

Against the background of climate change, changes of snow cover duration have important consequences for surface energy and water budgets over a range of scales, as well as for cryospheric ecosystems (e.g., Euskirchen et al., 2007; Yuan et al., 2016; Zeeman et al., 2017). Based on measurements and modeling, several studies have documented recent trends toward accelerated snowmelt and changes in snow cover duration caused by BC and dust across the Himalayan-TP in response to enhanced albedo reduction (Ménégoz et al., 2014; Menon et al., 2010; Xu et al., 2009). Ji et al. (2016) and Ji (2016) noted that BC and dust in snow/ice decreased surface albedo and caused the snow water equivalent to decrease by 5–25 mm over the TP. Ménégoz et al. (2014) has estimated that both dry and wet BC depositions affect the snow cover, reducing its annual duration by 1 to 8 days, which may potentially influence sustaining seasonal water availability (Immerzeel et al., 2010; Tahir et al., 2015).

## 3.4 Limitations and implications for snow cover and glaciers on the Tibetan Plateau

We calculated the effect of LAPs on snow cover duration with a range of simplistic assumptions based on simulations of albedo reduction and RF. We did not consider meteorological conditions, including clouds, precipitation, etc. Snow metamorphism was also not considered, and in the simulation, snow grains were assumed to be spherical, as previous studies (Flanner et al., 2007; Ginot et al., 2013; Schmale et al., 2017). These samples were collected during winter season when the snow cover was more stable and continuous, which might not represent the true impurities during the ablation season. It is difficult to sample during ablation season due to poor accessibility. Discontinuous snow cover during the melting season also make it difficult to represent the true impurities in snow. Besides, considerable heterogeneity in the topography and climate has led to complex spatial and temporal snow cover patterns on the TP (Xu et al., 2017).

We also did not consider the BC morphology and its mixing state, and light-absorbing properties of OC and dust, which may also significantly affect the light absorption in the simulation (Andreae and Gelencsér, 2006; Kaspari et al., 2014; Di Mauro et al., 2015; Painter et al., 2010; Schmale et al., 2017). BC in the atmosphere is mixed with OC and inorganic salts during aging enhancing its absorption (Gustafsson and Ramanathan, 2016). He et al. (2014) noted that BC-snow internal mixing increases the albedo forcing by 40–60 % compared with external mixing, and coated BC increases the forcing by 30–50 % compared with uncoated BC aggregates, whereas Koch snowflakes (Liou et al., 2014) reduce the forcing by 20–40 % relative to spherical snow grains. A lack of OC consideration due to the limitation of the SNICAR model also affected the results in this study. Due to its light-absorbing properties (Andreae and Gelencsér, 2006; Bahaur et al., 2012; Yan et al., 2016), OC in

snow cover can also absorb solar radiation and reduce the surface albedo. Previous studies indicated that OC was responsible for more than 10–40 % of the light absorption (Dang and Hegg, 2014; Doherty et al., 2010; Lin et al., 2014; Wang et al., 2015). Not considering OC in snow will underestimate the LAP impact on albedo reduction and RF, and this should not be ignored in the future.

Dust optical properties depend strongly on source material and dust properties are assumed to represent "global-mean" characteristics as closely as possible (Flanner et al., 2007). The light absorption by mineral dust in snow is thought to be due to iron oxides (Wang et al., 2013), which are efficient at light scattering and absorption and can enhance absorption at UV and visible wavelengths (Di Mauro et al., 2015; Moosmüller et al., 2012; Zhang et al., 2015a). Goethite and hematite are the most abundant forms of iron oxide in dust and the major light absorbers in the shortwave spectrum in snow (Wu et al., 2016). The

SNICAR model applies the Maxwell-Garnett approximation for combining indices of refractions, assuming a mixture of quartz, limestone, montmorillonite, illite, and hematite. We should note that some dust particles (e.g., those containing a large proportion of strongly absorbing hematite) can have a larger impact on snow albedo than the dust applied in this work (Kaspari et al., 2014; Painter et al., 2007).

It is further important to note that a large fraction of such uncertainty can also result from uncertainty in the simulated snow-

cover fraction/snow depth across the plateau. The TP covers a large area (more than 2.5 million km$^2$) with an average elevation exceeding 4000 m a.s.l. (Yao et al., 2012a). Considerable heterogeneity in the topography and climate has led to complex spatial and temporal snow cover patterns (Xu et al., 2017). Most studies on the snow cover distribution were based on satellite-based observations (Che et al., 2008, 2012). Surface observations of snow depth showed an increase over the TP from 1957–1998 (Ma and Qin, 2012). However, snow cover depth and the number of snow covered days during the current decade under

intense climate warming showed a negative trend mainly occurring in the southeast TP in winter. Whereas in spring, snow cover depth showed a positive trend in the eastern TP (Xu et al., 2017). These differences can also affect the estimates in this study. Nevertheless, the method provides a theoretical approach for evaluating how the presence of LAPs affects the lower parts of the glacier subjected to summer melt.

Snow cover durations were shortened during the melt season from 1.26−9.4 days as estimated in this study (Fig. 8), and the

glaciers over western China have been undergoing extensive recession, with an estimated loss of about ~450 km$^3$ w.e. in volume and 10 % reduction in area during the last 40 years (Yao et al., 2012b). Recently, the annual mass loss of glaciers over the TP was estimated at approximately 15–20 Gt yr$^{-1}$ based on GRACE and ICESat (Gardner et al., 2013; Neckel et al., 2014). These melt waters may affect water resources and social development in the surrounding areas, e.g., India and China (Immerzeel et al., 2010). The total amount of snowmelt caused by LAPs can be estimated over the whole TP using the model

described in Sec. 2.6. The number of days with average temperature above 0 ℃ was assumed to be 70 (±25 %) based on the observations from AWS during the melting season. Thus, based on the daily snowmelt (0.76±0.40, 0.67±0.36, and 0.54±0.28 cm w.e. d$^{-1}$ for $SW$ at 220, 270, and 310 W m$^{-2}$, respectively) caused by BC and dust in this study, the estimation shows that approximately 4.61±2.49 Gt yr$^{-1}$ (range of 3.80±1.97 to 5.36±2.78 Gt yr$^{-1}$) of snowmelt water has been lost due to the effect of BC and dust in snow (Fig. 9). This indicates that glacial snowmelt caused by LAPs contributes ~20 % of mass loss and that

BC and dust may play an important role in recent accelerated snow and glacial melt. However, this result has large uncertainties associated with the lack of sufficient *in situ* and distributed observations of spectral albedo, concentrations of LAPs, and coincident changes in glacier mass observations, as discussed by many previous studies (e.g., Ginot et al., 2014; Ménégoz et al., 2014; Qian et al., 2015; Schmale et al., 2017).

Our study confirms that BC and dust in snow and ice can darken the surface, and further reduce snow albedo, and increase the speed of snow cover melt (e.g., Flanner et al., 2007; Di Mauro et al., 2015; Painter et al., 2007; Qian et al., 2015; Skiles et al., 2015). Therefore, the impacts of BC and other LAPs need to be properly accounted for in future regional climate projections, particularly in the high-altitude cryosphere. It is also important to note that, without large scale, *in situ* observation data applied to calibrate BC forcing and constrain simulated values, the impact of BC and other LAPs on the albedo feedback would have

uncertainties. Schmale et al. (2017) pointed out that, due to the decreasing trend of dust emissions in Central Asia (Xi and Sokolik, 2015), dust may become less important, while BC deposition may increase as a result of projected increased emissions (Wang et al., 2014). Constraints on BC sources/deposition and climatic/hydrological effects will provide guidance for effective mitigation actions.

## 4 Conclusions

This study was a large area survey of LAP in snow cover over the TP across regions with different climatology and sources of LAPs. Systematically higher BC concentrations in snow were observed in the central to northern TP (Regions II and III) than in the southern TP (Region I). This finding may be related to shallower snow depths and dirtier snow in Regions II and III as a result of post-depositional processes (e.g., snowdrift, or melting). In the southern TP (Region I), the collected snow was cleaner with lower concentrations of LAPs comparable to concentrations in the glacial aged snow. The ratios of OC/BC

in the surface snow samples ranged from 0.64 to 3.31 across the TP, generally decreasing from south to north, which may indicate that anthropogenic activities played an important role on OC/BC ratios in snow cover in the TP region. Back trajectory analysis also indicated that anthropogenic sourced BC accounted for approximately 70 % of total BC deposition in the central and northern TP. Estimates of albedo reduction contributed by BC and dust were approximately 38 %, with an associated RF of 18–32 W m$^{-2}$.

The reported shortage of snow cover duration days across the TP relied on single site observations rather than a large area survey. For different scenarios with different *SD* and *SW*, the advanced melt days of snow cover duration related to LAPs was mainly due to BC, which reduced snow cover from several days to more than one week. Dust contributed only approximately 1 day to the reduction of snow cover duration days. No significant differences among reductions in snow cover duration were found across the whole TP. The changing snow cover duration may potentially influence seasonal water

availability and sustainability.

Furthermore, the estimated effects of BC and dust on glaciers and snowmelt across the TP were based on the simulations of albedo reduction and RF. It was found that glacial snowmelt water caused by LAPs contributed ~20 % of the mass loss. These findings revealed that the effects of both BC and non-BC absorbers need to be properly accounted for in future

regional climate projections, in particular on the high-altitude cryosphere.

**Data availability**

Supplementary information on black carbon and other light absorbing data are available online or upon request to Professor Shichang Kang (shichang.kang@lzb.ac.cn) and Dr. Yulan Zhang (yulan.zhang@lzb.ac.cn).

**Author contribution**

Y. Zhang, S. Kang, and Z. Cong designed the experiments. M. Sprenger performed the back trajectory simulations. Y. Zhang, T. Gao, C. Li, X. Li, X. Zhong, and M. Xu, carried out the experiments. S. Tao and W. Meng analyzed the black carbon inventory. M. Sillanpää and X. Qin gave valuable comments. Y. Zhang prepared the manuscript with contributions from all co-authors.

**Acknowledgement**

We acknowledge the support provided by the National Natural Science Foundation of China (41421061, 41630754, and 41671067), State Key Laboratory of Cryospheric Sciences (SKLCS-ZZ-2017), Chinese Academy of Sciences (KJZD-EW-G03-04), the Foundation for Excellent Youth Scholars of the Northwest Institute of Eco-Environment and Resources, CAS, and the China Scholarship Council. We also thank Dr. M Flanner for help with simulating reflectance with the SNICAR model, 15 and the reviewers who helped to improve this manuscript.

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

21     **Table 1 Snow effective grain sizes and snow density used for the albedo calculation with the SNICAR model.**

| Description | Low scenario | Central scenario | High scenario |
|---|---|---|---|
| Snow effective grain radius ($\mu$m) | 150 | 500 | 1500 |
| Snow density (kg m$^{-3}$) | 150 | 250 | 400 |

 **Table 2 Summary of BC, OC and dust concentrations in snow over the Tibetan Plateau and other regions.**

| Sites | BC (ng g$^{-1}$) | Dust (μg g$^{-1}$) | OC (ng g$^{-1}$) | OC/BC | Method | References |
|---|---|---|---|---|---|---|
| Southern TP (Region I) | 1423 ±242 | 119.8 ±18.97 | 2145 ±325 | 1.82 ±0.17 | DRI | This study |
| Central TP (Region II) | 5624 ±1500 | 295 ±67.50 | 6119 ±1257 | 1.31 ±0.19 | DRI | This study |
| Northern TP (Region III) | 1484 ±626 | 93.20 ±27.05 | 974 ±197 | 1.14 ±0.15 | DRI | This study |
| Himalayan Khumbu Valley | 0.1-70 | | | | SP2 | Jacobi et al., 2015 |
| Himalayas | 24.3 ±20.1 | 1.32 ±0.84 | 359 ±185.2 | ~10 | Sunset | Lim et al., 2014 |
| Qilian mountain | 1550 | | | | ISSW | Wang et al., 2013 |
| Border of Siberia, China | 117 | | | | ISSW | Wang et al., 2013 |
| Northeast China | 1220 | | | | ISSW | Wang et al., 2013 |
| Inner Mongolia, China | 340 | | | | ISSW | Wang et al., 2013 |
| Northwest China | 10-150 | | | | ISSW | Pu et al., 2016 |
| Sapporo, Japan | 7-2800 | 10-1300 | 0.14-260 | | Sunset | Kuchiki et al., 2015 |
| Central North America | 5-70 | | | | ISSW | Doherty et al., 2014 |
| Uintah basin, Utah, USA | 5-100 | | | | ISSW | Zatko et al., 2016 |
| Sierra Nevada, North America | 11 ±7.7 | | | | DRI | Hadley et al., 2010 |
| French Alps | 4.3 ±4.5 | 3.06 ±5.25 | 59.6 ±77.6 | | SP2 | Lim et al., 2014 |
| Greenland Summit | 3.1 ±1.4 | 0.262 ±0.117 | 142.6 ±82.6 | | SP2 | Lim et al., 2014 |
| Greenland | Sumit station: 1.4 | | | | ISSW | Zatko et al., 2013 |
| Arctic | Scandinavia: 88 | | | | Sunset | Forsström et al., 2013 |
| | Svalbard: 11-14 | | | | | |
| | Fram Strait: 7-42 | | | | | |
| | Barrow: 9 | | | | | |
| Antarctic | Near Dome C: 2.1 | | | | ISSW | Zatko et al., 2013 |
| | Far from Dome C: 0.6 | | | | | |

Note:

(1) DRI-DRI 2001A model thermal-optical carbon analysis

(2) SP2: Single Particle Soot Photometer

(3) ISSW- ISSW spectrophotometer

(4) Sunset: Sunset Lab OC-EC Aerosol Analyzer

**Table 3 Comparison between measured albedo by ASD and simulated albedo (SA) by the SNICAR model.**

| Selected site (24K) | SA clean snow | SA BC | SA Dust | SA BC+Dust |
|---|---|---|---|---|
| SNICAR: Broadband (305-4495 nm) | 0.7824 | 0.7027 | 0.7511 | 0.6892 |
| SNICAR: Wavelength (350-2500 nm) | 0.7728 | 0.6985 | 0.7453 | 0.6864 |
| ASD measurement (350-2500 nm) | 0.6709±0.0019 | | | |

**Table 4 Simulated albedo (SA) and radiative forcing (RF) by BC and dust in snow cover across the Tibetan**
**Plateau.**

| Different scenarios | SA clean snow | SA BC | SA Dust | SA BC+Dust | Contribution to albedo reduction | RF BC | RF Dust | RF BC+Dust |
|---|---|---|---|---|---|---|---|---|
| Low | 0.7751±0.0193 | 0.6517±0.0702 | 0.7387±0.0250 | 0.6413±0.0710 | 37 % | 16.32±0.0931 | 4.78±0.0310 | 17.64±0.0952 |
| Medium | 0.6943±0.0313 | 0.5104±0.0900 | 0.6372±0.0382 | 0.4992±0.0900 | 39 % | 24.20±0.1250 | 7.57±0.0483 | 25.86±0.1233 |
| High | 0.5992±0.0435 | 0.3683±0.0935 | 0.5197±0.0490 | 0.3582±0.0889 | 38 % | 30.96±0.1362 | 10.70±0.0653 | 32.36±0.1315 |

**Table 5 Average reductions of snow cover duration by BC and dust for different short wavelengths (220, 270,**
**and 310 W m$^{-2}$, respectively) and snow cover depth water equivalent (mm).**

| SD / Shortwave | SD=40 mm | | | SD=100 mm | | |
|---|---|---|---|---|---|---|
| | BC | Dust | BC+dust | BC | Dust | BC+dust |
| SW=220 W m$^{-2}$ | 1.69±0.07 | 0.74±0.04 | 1.77±0.07 | 4.24±0.18 | 1.84±0.10 | 4.43±0.18 |
| SW=270 W m$^{-2}$ | 1.38±0.06 | 0.60±0.03 | 1.44±0.06 | 3.45±0.15 | 1.50±0.08 | 3.61±0.15 |
| SW=310 W m$^{-2}$ | 1.20±0.05 | 0.52±0.03 | 1.26±0.05 | 3.01±0.13 | 1.31±0.07 | 3.14±0.13 |

**Figure 1**

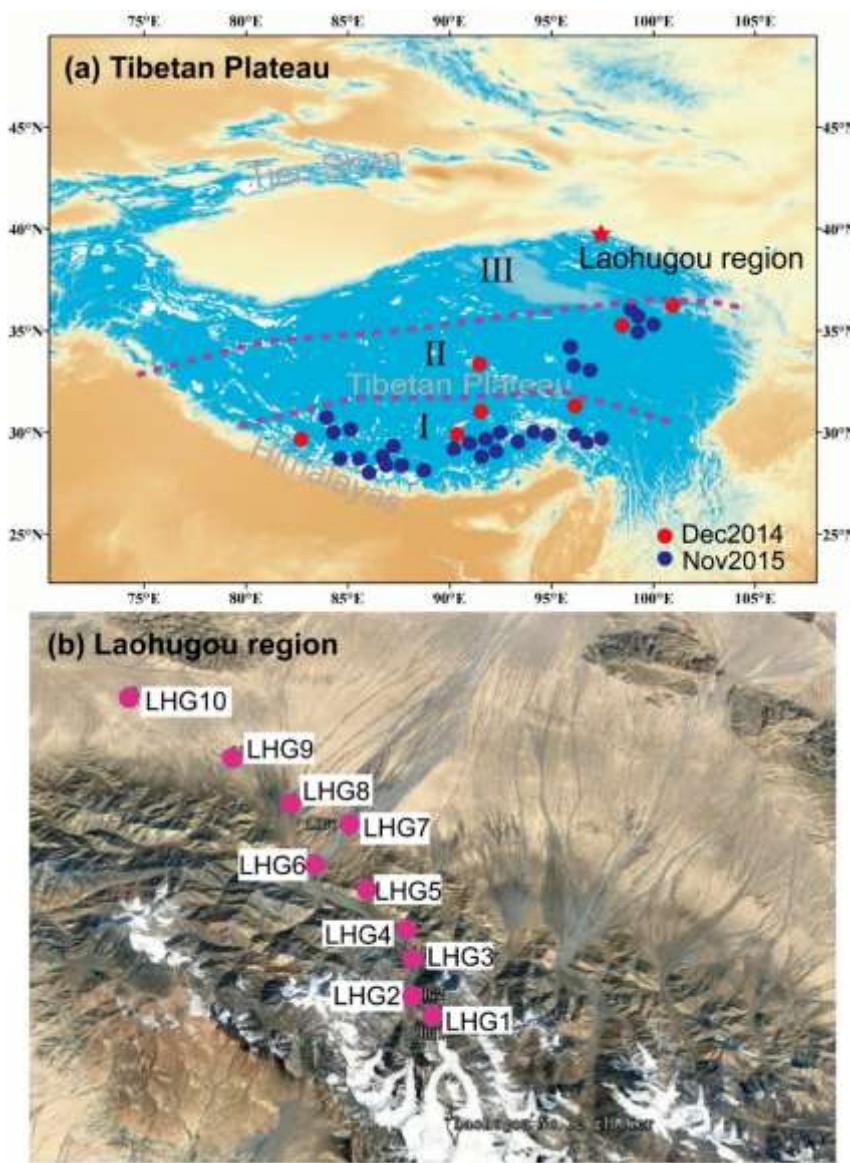

**Figure 1: Snow sampling sites over (a) the Tibetan Plateau and (b) the Laohugou region.**

**Figure 2**

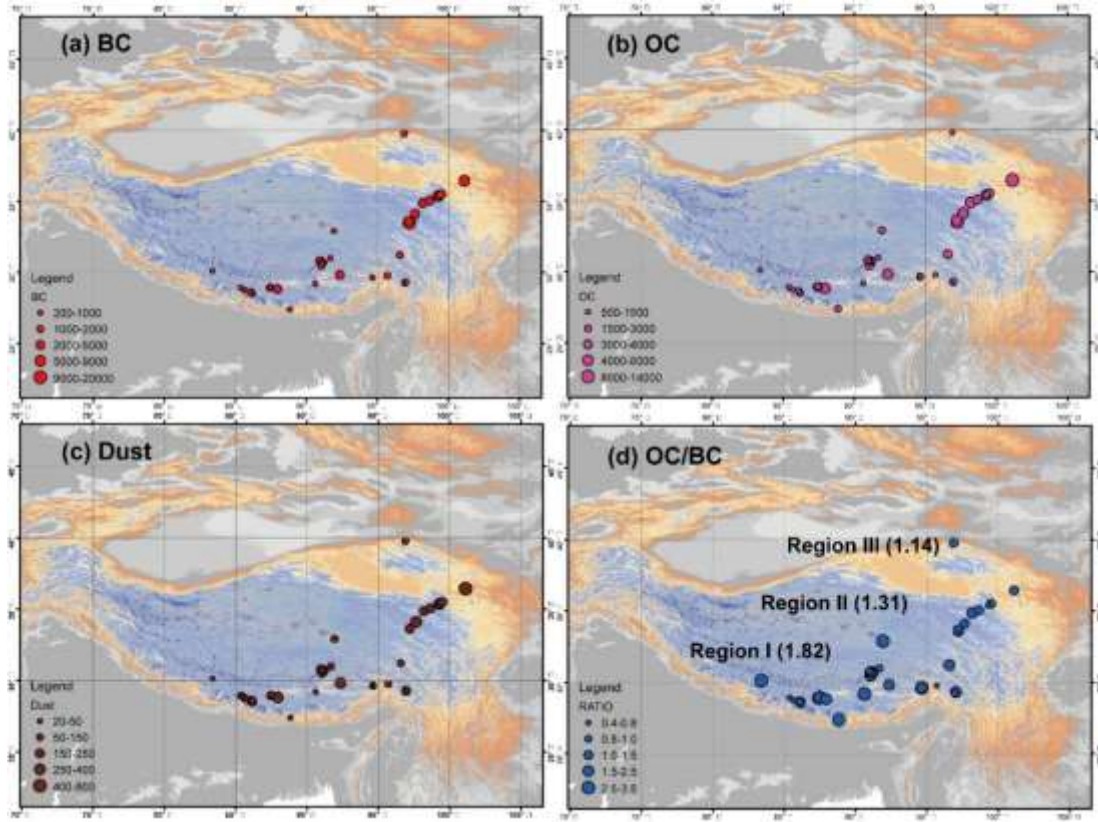

Figure 2: Spatial distributions of light-absorbing impurities in snow cover for each sampling site over the Tibetan Plateau. (a) BC, (b) OC, (c) Dust, and (d) ratio of OC to BC (OC/BC).

## Figure 3

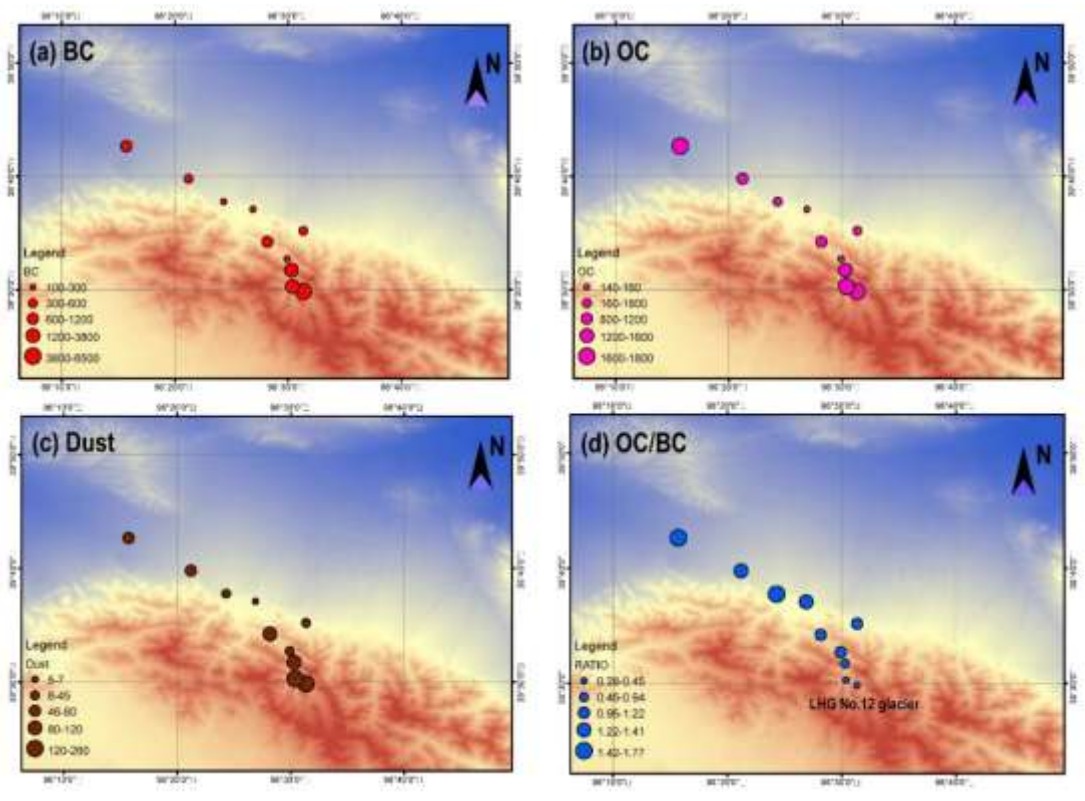

**Figure 3. Spatial distributions of light-absorbing impurities in snow at the Laohugou region. (a) BC, (b) OC, (c) Dust, and (d) ratio of OC to BC (OC/BC).**

**Figure 4**

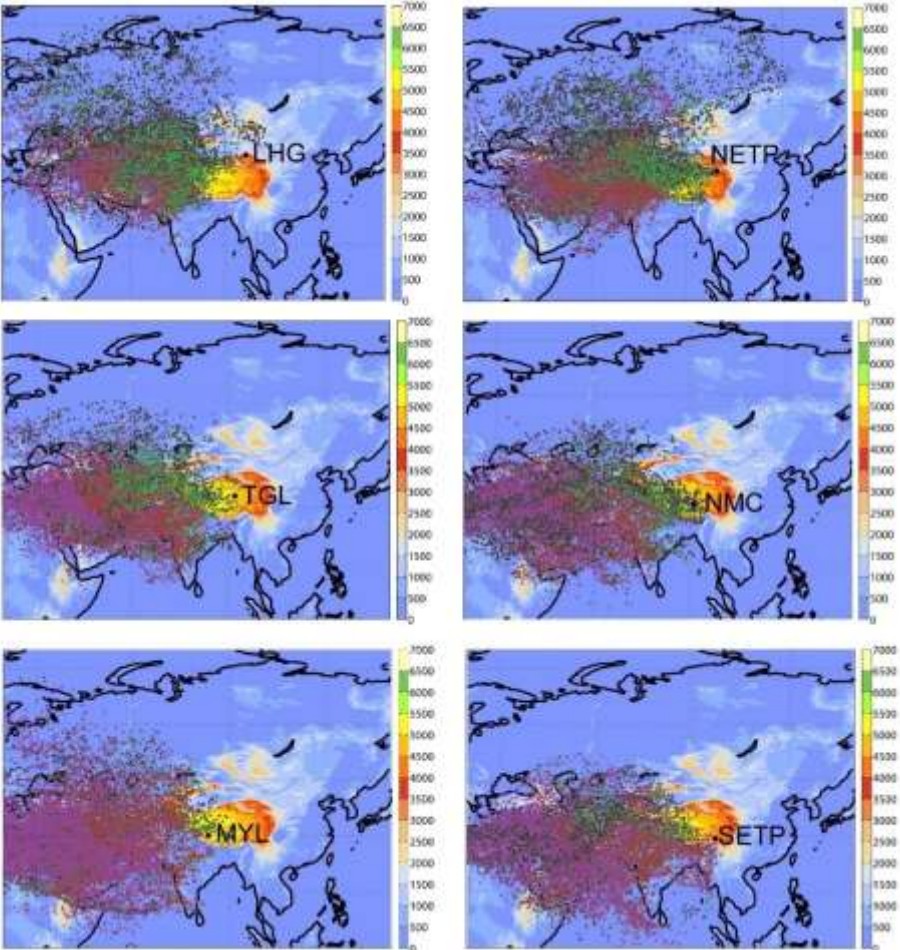

**Figure 4: Back trajectory analyses for six selected sites over the Tibetan Plateau during the winter season (Nov 2015-Feb 2016). LHG and NETP in the northern Tibetan Plateau (Region III), TGL and NMC in the central Tibetan Plateau (Region II), and MYL and SETP in the southern Tibetan Plateau (Region I). (The right color bar means the height (m a.s.l.). The trajectories starting below 500 hPa were taken into account. Black dots: the air parcel did not pass near a fire during the 96 h prior to arrival at the studied site; Green dots: the air parcel did pass by a fire between -96 h and -48 h, but not afterwards, i.e., the 'contact' to a fire lies back at least 48 h before the air parcel arrived at Renlongba glacier; Magenta dots: contact with fires occurred between -48 h and 0 h before arriving at site; Red dots: contact with a fire before and after -48 h.)**

**Figure 5**

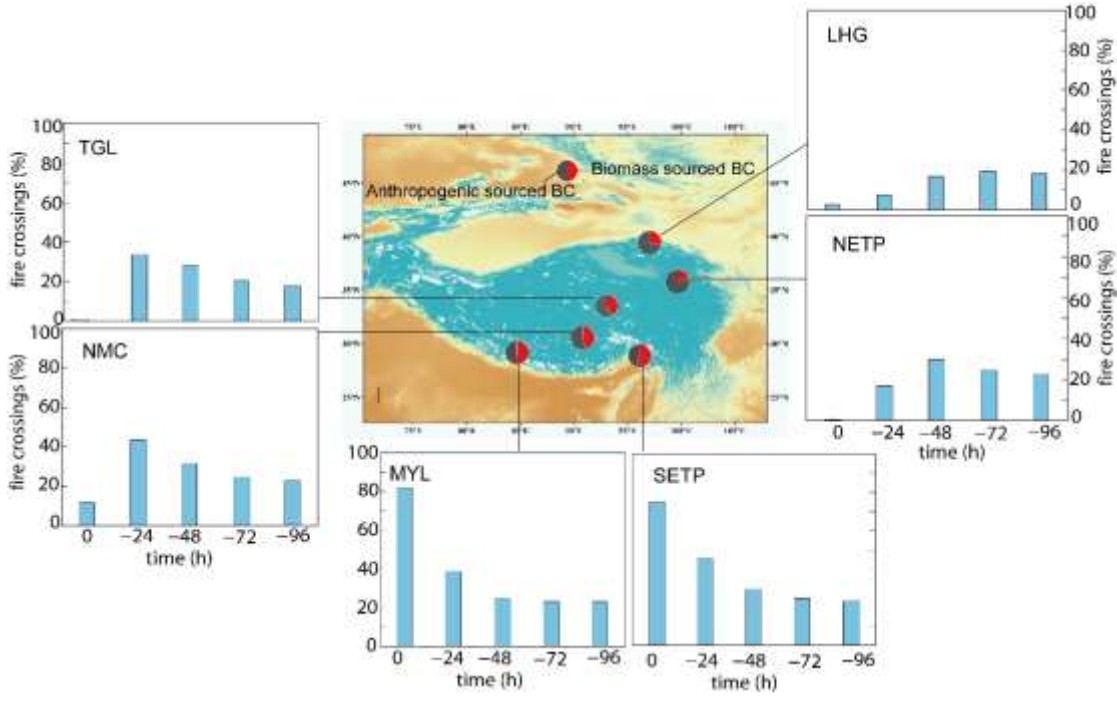

**Figure 5: Different source contributions to BC deposition on the snow cover across the Tibetan Plateau.**

**Figure 6**

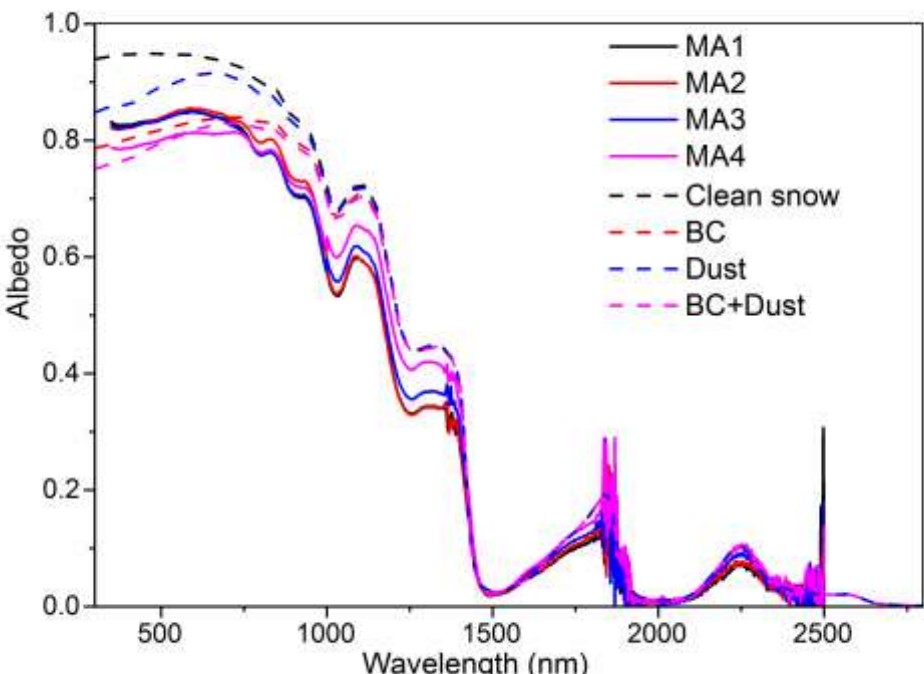

**Figure 6: Measured albedo by ASD (solid lines, MA1-4) and simulated effects of BC and dust on albedo**
**(dashed lines) at the selected snow site on the Tibetan Plateau.**

**Figure 7**

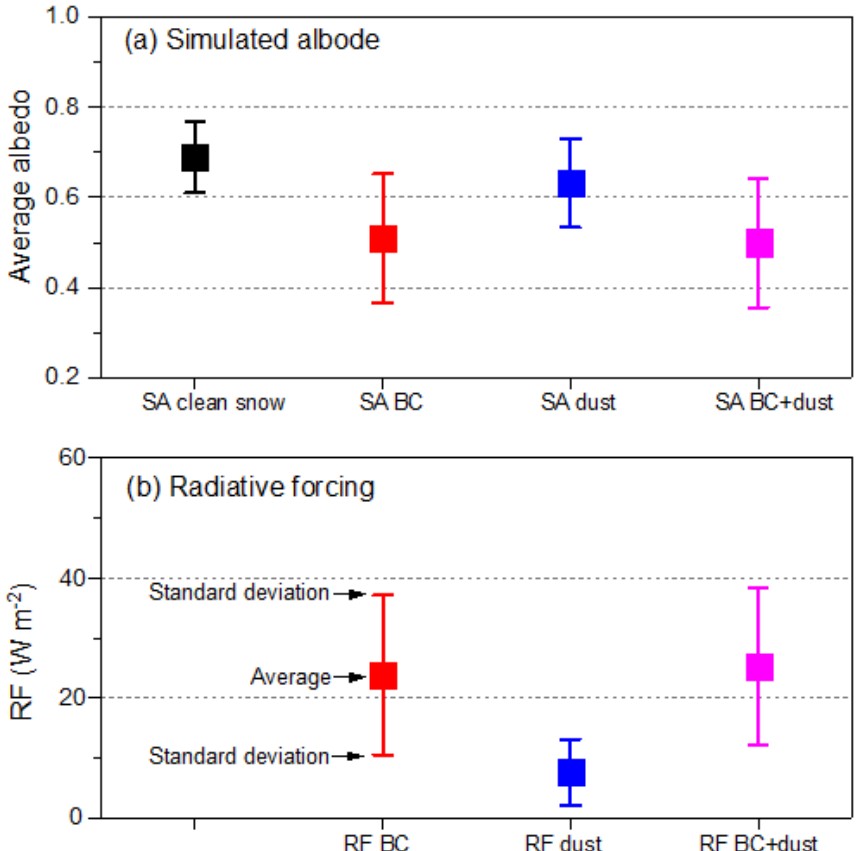

**Figure 7: (a) Average albedo (solid rectangles) of snow cover due to aging only and impurities relative to the**

**clean snow and (b) resulting clear sky instantaneous radiative forcing (solid rectangle means the average).**

**(Note: SA means simulated albedo, SA BC means albedo only with BC in snow, SA dust means albedo only**

**with dust in snow, and SA BC+dust means albedo with BC and dust in snow. RF means radiative forcing, RF**

**BC means the RF caused by BC in snow, RF dust means the RF caused by dust in snow, and RF BC+dust**

**means the RF caused by BC and dust in snow.The solid rectangles are the central average estimate whereas**

**the bars show the standard deviation range of albedo and RF)**

# **Figure 8**

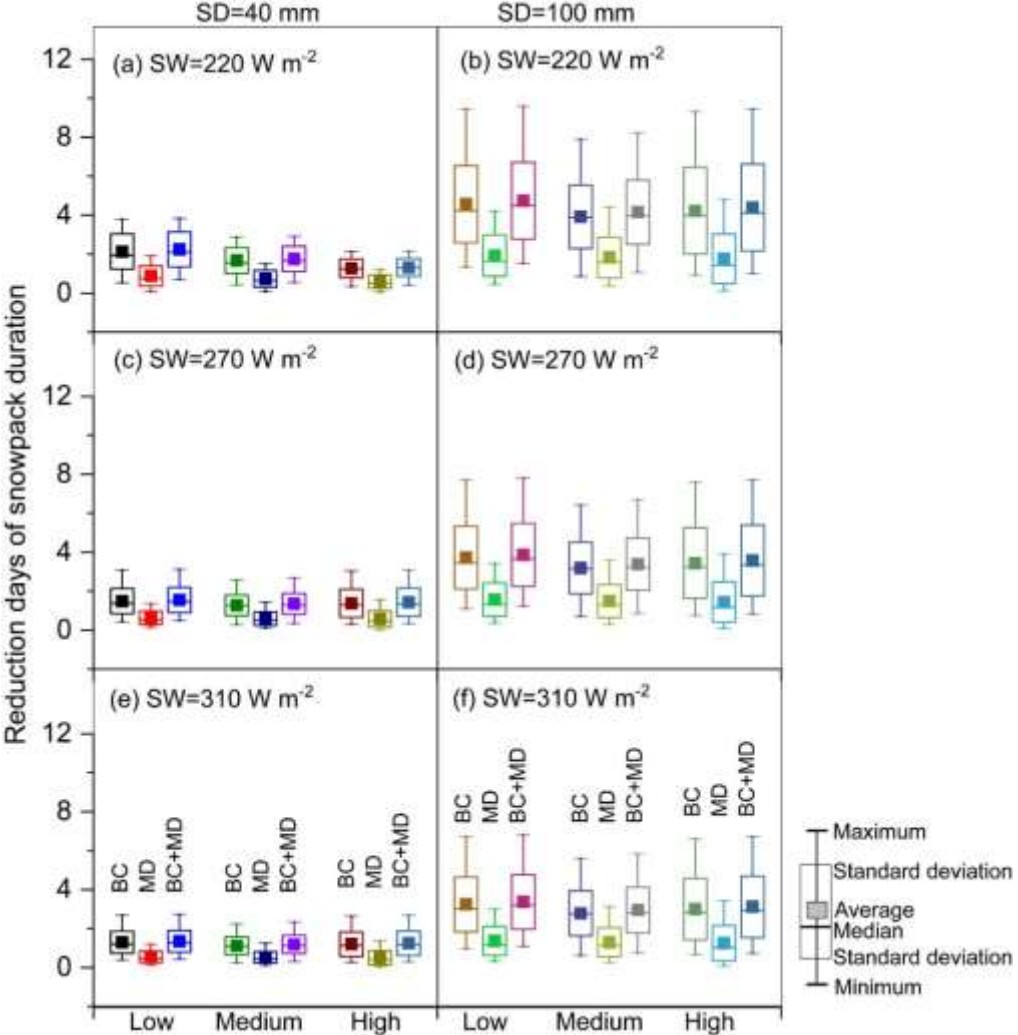

**Figure 8: Reduction in days of snow cover duration by BC and dust for low, medium and high scenarios due to albedo reduction at different snow depth (SD=40 and 100 mm, respectively). a and b were simulated based on the daily shortwave at 220 W m-2. c and d were simulated based on the daily shortwave at 270 W m-2, e and f were simulated based on the daily shortwave at 310 W m-2. (Solid rectangles are average values; solid line in the box means the median; outliers of boxes are the one standard deviations of the average; and the bars are maximum and minimum values.)**

**Figure 9**

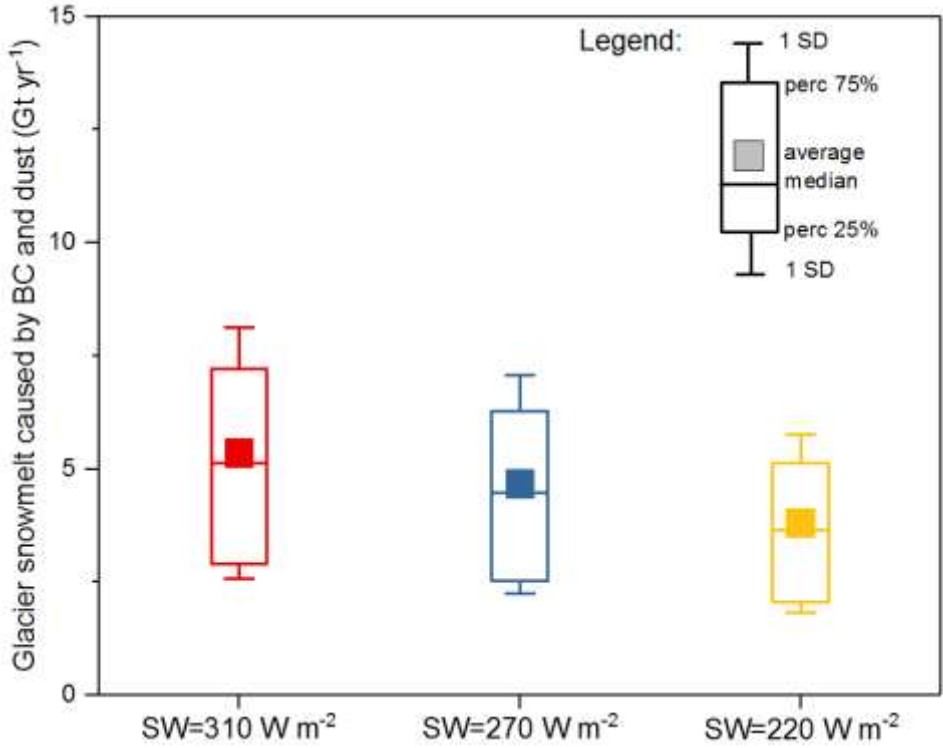

**Figure 9: Estimates of total glacial snowmelt based on daily snowmelt caused by BC and dust for shortwave**

**radiation at 220, 270, and 310 W m⁻², respectively. The days of daily temperature above 0 °C value is 100**

**(±25 %).**
