# Peer review of "Black carbon and mineral dust in snow cover on the Tibetan Plateau"

_The Cryosphere, 2017_

## Referee Comment (RC1) · Anonymous Referee #1 · 12 Aug 2017

Review for TC-2017-111 Title: Black carbon and mineral dust in snow cover on the Third Pole

General opinion: The authors report the observation of light-absorbing impurities in snow over Tibetan Plateau. Based on these field observed data, they calculated the albedo reduction induced by black carbon and mineral dust, and the corresponding impact on snow energy budget. The field data reported in this work are valuable for quantifying the impact of light-absorbing particles on snow albedo and the analysis based on this field data are informative. However, some discussion and conclusions given in this paper are not accurate and need to be modified. This manuscript also contains a lot of typos/grammar errors (some obvious errors are listed below) that need to be corrected before this manuscript can be considered for publication. Here are the

suggestions/comments that the authors may find useful.

Comments and Questions

Page 1 Line 30-32: BC is recognized as an important climate forcer not only because it absorbs sunlight, but also because a large fraction of BC are emitted from anthropogenic sources. Please be more accurate at here.

Page 2 Line 16: "... snow covered range" → snow covered region?

Line 20: "Confirming radiative transfer modeling" . It is not very clear to readers what does the authors mean. Please consider revise this.

Line 24: "..., but also to" → but also important/crucial to

Line 26: "... is the most sensitive" how could tell it is the most sensitive? Please remove most or provide supporting data.

Line 33: "... and result in perturbation in" → ...and perturb

Line 34: "5 – 25 mm in the snow water equivalent..." → 5-25 mm snow water equivalent

Page 3 Line 1: "snowpack on the TP is associated with the ..." by associated you mean the snowpack on the TP is influenced by summer monsoon? Or the snowpack will influence summer monsoon? or both? Please be more accurate here.

Line 4-5: "Simulation studies of BC in snow over TP have inherent uncertainties because of the lack of large-area observations of BC data in seasonal snow cover". This is not correct. Model simulations have inherent uncertainties due to the physics/chemistry/transport schemes used in the models - such as uncertainties in BC emissions sources or deposition rate. Large-area observation of BC in TP will be useful for model evaluation, but it will not help with the inherent uncertainty of the model simulation. Please revise it here.

Page 4 Line10: " ... and is dominated by the Indian monsoon" → A region cannot be

dominated by monsoon. Do you mean "affected by"?

Line 11: "Region III has one site (LHG) is located in the northeastern part of the TP" → "Region III has one site (LHG) that is located in the northeastern part of the TP"

Line 15: it seems the snow samples were only collected from the top 5 cm, so how did you calculate the snow albedo for snowpacks thicker than 5 cm? Did you assume the BC/MD mass mixing ratio is constant through the entire snow column? Please clarify this.

Line 17: how did you measure snow grain size in the field? Please clarify this.

Line 24-26: what is the accuracy of the weight you used to measure filter before and after filtration? It seems the author assume BC/OC/MD are the only insoluble particles deposited on the quartz filter, could you please provide more evidence about this? If MD were the only other particle in snow (besides OC and BC), do all MD absorb sunlight? It might be a good idea to include these discussions in the uncertainty analysis.

Page 5: Line 6: what is the filter blank for? is this a blank filter? It seems there is some particles on filter blank as well since it weighs more than 0 C/cm-2.

Line 8: "separately analyzed" → analyzed separately

Line 18: "... is or not influenced by BC emissions" → is influenced by BC emissions or not

Line 23: what is "down-sun"?

Line 24: "when the weather was clear"→ when the sky was clear. It seems the albedo measurement were made for clear sky only, but later in the albedo comparison, the measured albedo is compared against the albedo calculated using SNICAR for all-sky case. Is this a typo or this is wrong?

Line 28: "e.g. Doherty et al., 2010": Doherty et al 2010 did not use SNICAR, it only reported the observation in the Arctic. Please remove this citation.

Page 6: Line 9-11: Please revise this sentence. Is this a model or is this just the method you used to calculate snow water melt in this work?

Line 16: "for clean snow..." what is cause of albedo reduction for clean snow case? a different snow grain size due to snow aging? Please be more accurate here.

Line 18: "assumed snow depth" what is the assumed snow depth? How did you assume it?

Line 22: "older wind-packed snow" → old wind-packed snow

Page 7: Line 2: Please consider cite paper Doherty et al., 2016. It also reports LAIs in snow in North America. Doherty,S. J., D. A. Hegg, J. E. Johnson, P. K. Quinn, J. P. Schwarz, C. Dang, and S. G. Warren, (2016), Causes of variability in light absorption by particles in snow at sites in Idaho and Utah. J. Geophys. Res. Atmos., 121, no. 9: 4751-4768.

Line 6-8: How could you tell the LAIs in snow over TP were generated from biomass burning in surrounding region from Table 2? And from Table 2, how could you tell if the fraction of LAIs emitted by biomass/fossil fuel burning in TP is larger than that in the other regions? Please provide more explanations.

Line 9-11: How could the ratio of OC/BC be used as a standard to determine the emission sources of biomass burning? What is the OC/BC ratio if all LAIs were emitted from biomass burning or non-biomass burning sources?

Line 15: "LHG3 AND LHG6 (Figure 3)", do you mean Figure 1b?

Line 15-21: It seems Figure S2 is an important figure for discussions in this part. If there is no restriction on the number of figures, please consider include this figure in the paper.

Line 28: "Open burning sourced BC"→ BC emitted from open burning sources. Does the BC emitted from open burning sources contain BC emitted from biomass burning?

or are they the same? Please be more specific about what you mean by "open burning".

Page 8 Line 3-4: "In the southern TP. . . . due to influence of BC sources in the south Asia". This statement is too vague. Do you mean the total emission is larger in southern Asia? or the BC deposition rate is larger? or both? This is a major conclusion you have in this work, please be more specific.

Line 8: You used SNICAR model to calculate snow albedo for all-sky cases, but the model set up is not stated. For example, what is the cloud fraction for all sky case? As mentioned in the previous comments, the albedo measurement was performed under clear sky, so in what accuracy do you expect this to agree with the SNICAR calculation (Figure 5)?

Line 14: "The deviation between measured and simulated reflectance by MD may be a result of the upper boundary of the SNICAR model in particle dimension". This does not make sense, is the deviation due to upper boundary condition (and what upper boundary condition)? or MD particle size? Please revise.

Line 18-19: "This result is import, showing that the SNICAR model simulations can represent albedo changes of snow cover in the Third Pole region". This is a really strong conclusion. I don't think the authors can make this conclusion based on the results shown in this paper. Especially it is unclear to the readers that how did the authors set up the SNICAR calculation. Please remove this or include more details about SNICAR calculation.

Line 26. As the author said, the BC snow albedo forcing over TP is highly uncertain partly due to the uncertainty in simulated BC concentration. But it is also important to point out that a large fraction of such uncertainty is resulted form uncertainty in simulated snow-cover fraction/snow depth. Please consider including this in the discussion.

Page 9 Section 3.3: Why did you pick SW flux of 220, 270 and 310 W m-2? Line 27: "SD plays an import role" → SD plays an important role? What do you mean by SD

plays a role here? Deeper snow is supposed to melt slower given the same amount of radiative forcing; it is not because snow depths play a role here. Please clarify this point.

Page 10: Line 13: "However, the results presented in this study .... for which these assumptions are not critical": This is not true. All the quantities listed in this paragraph will influence the snow albedo and most of them will influence the albedo reduction induced by BC. For example: BC-snow internal mixing increases the albedo forcing by 40-60% compared with external mixing (He et al. (2014). The author should discuss the uncertainty of this study resulted from the assumptions they made, instead of claim these quantities will not impact their results.

References: He, C., Q. Li, K. N. Liou, Y. Takano, Y. Gu, L. Qi, Y. Mao, and L. R. Leung, 2014: Black Carbon Radiative Forcing over the Tibetan Plateau. Geophy. Res. Lett., 41, 7806-7813, doi: 10.1002/2014GL062191.

Line 32: "Our study confirms that ..... and further reduces snow albedo,....": this is not true. BC and other LAI can reduce the snow albedo even if the snow aging process is not accelerated. Please revise this.

Figures3: Is the color bar showing height? Please define the color bar and unit.

Figure 5: Are MA1-4 measured albedo? Are the dashed lines albedo calculated using SNICAR? Please clarify these details and modify the corresponding text.

Figure 6-8: please clarify the figure convention in each figure. Do the boxes represent average values from central estimate? For example, in Figure 6, you say the rectangles are central estimate, so what does the box mean? standard deviation of central estimate? maximum and minimum of central estimate?

---

## Referee Comment (RC2) · Anonymous Referee #2 · 25 Aug 2017

This manuscript presents results from snow sampling, laboratory analysis, and related modeling efforts for the presence and impact of light absorbing impurities in the Tibetan Plateau region. Although the manuscript is not wrong in stating more measurements are valuable in constraining our understanding of the spatial and temporal variability of light absorbing particulates, the techniques presented in this paper are unclear and as written does not contribute to the state of knowledge of light absorbing particulates in snow. Major changes are needed before this paper can be reviewed again for publication.

I will not, at the point, do line by line corrections because they are numerous. The authors need to revisit the writing in each section for editing and to clarify there justification, methods, and results. Particularly, snow sampling and automatic weather

stations need to be described in significantly more detail.

My first and foremost concern is that samples were collected in November and December, and yet they are attempting to quantify the impacts of light absorbing particulates on melt. This does not make sense to me and if I misunderstood this it is because it is not made clear in the manuscript. Although the sample collection timing may be after the summer monsoon, these samples do not represent the impurities that are present during the ablation season- and therefore it is inappropriate to use these values to quantify reduction in snowpack duration. Particularly for dust, which tends to deposit in the spring when source regions dry out (peak radiative forcing by dust in snow is observed from MODIS imagery over the Himalayan region in April and May).

The backtrajectory footprint modeling was also very unclear to me, how the model runs were carried out needs to be described better, but it is unclear to me why the model runs were only completed for the winter. And were they run continuously? Typically particulates are deposited in episodic events so running it continuously does not inform the source region, it just informs of the regional synoptics. I suggest seeing Skiles et al., 2015 (Hydrological Processes) for how that study produced and described back-trajectory footprints.

The albedo measurements need to be better described. And how was snow effective grain radius retrieved? This should be an optically equivalent grain size, not an observable grain size. If an effective/optical grain size was used, the retrieval should be described. If observed grain sizes were used, the large error introduced by this (see Painter et al., 2007 in Journal of Glaciology) needs to be mentioned.

Estimates of changes in snow cover duration should be removed or significant more detail and justification needs to be made for the timing of sample collection. Section 3.3 and 3.4 are generally confusing- was shortwave radiation (uplooking and downlooking pyranometers) actually measured? This analysis seems far too simplified. Furthermore, the discussion of snow depth is misleading. This is a straightforward energy

balance calculation, so less snow will melt faster than more snow for an equal amount of forcing- this is basic mass balance. So snow depth does not itself play an important role in the reduction of snow cover by particulates. This is a mixing up of forcing and state functions.

The citations are also outdated or incorrect in many cases, and I suggest the authors revisit these.

I take issue with the use of the 'Third Pole' term, this is not universally recognized and is somewhat politicized, why not just use Tibetan Plateau or High Mountain Asia? Also light absorbing impurity is an outdated term, the community has moved toward the use of light absorbing particulates. Also mineral dust and the acronym MD are confusing- you can simply say dust- which needs no acronym. Similarly, please be consistent in terminology, for example, albedo and reflectance are not the same thing.

Uncertainties in modeling are not only due to lack of observations, and increasing our number of sampling points alone will not reduce our model uncertainty. To state this is misleading.

---

## Author Comment (AC1) · 11 Oct 2017

Oct10, 2017

Dear editor,

The manuscript: Black carbon and mineral dust in snow cover on the Tibetan Plateau by Zhang Y.L. et al.

The manuscript number: tc-2017-111

We greatly appreciate the reviewers' constructive comments to improve the paper. We have revised our manuscript according to these comments (blue color in the main text), and hope the revised manuscript is suitable for publication in The Cryosphere.

The "point to point" response to comments are listed as below.

Sincerely yours, Yulan Zhang and Shichang Kang

Response to Referee #1

General opinion: The authors report the observation of light-absorbing impurities in snow over Tibetan Plateau. Based on these field observed data, they calculated the albedo reduction induced by black carbon and mineral dust, and the corresponding impact on snow energy budget. The field data reported in this work are valuable for quantifying the impact of light-absorbing particles on snow albedo and the analysis based on this field data are informative. However, some discussion and conclusions given in this paper are not accurate and need to be modified. This manuscript also contains a lot of types/grammar errors (some obvious errors are listed below) that need to be corrected before this manuscript can be considered for publication. Here are the suggestions/comments that the authors may find useful. Answer: Thank you very much for the comments. We have carefully revised according to the comments. The manuscript has also been improved by a native speaker to reduce the types/grammar errors.

Comments and Questions Page 1 Line 30-32: BC is recognized as an important climate forcer not only because it absorbs sunlight, but also because a large fraction of BC are emitted from anthropogenic sources. Please be more accurate at here. Answer: We have revised as following in the main text (Page 1-2): Atmospheric BC is a distinct type of carbonaceous material from incomplete combustion of biomass/biofuel and fossil fuel. A large fraction of BC is emitted from anthropogenic activities (Bond et al., 2013). Because BC can absorb solar radiation, influence cloud processes, and alter the melting of snow cover and glaciers, it has been considered to be the second most important climate forcer in the Earth's climate system only after carbon dioxide (Andreae and Ramanathan, 2013; Bond et al., 2013; Ramanathan and Carmichael, 2008).

References: Andreae, M. O., and Ramanathan, V.: Climate's dark forcings, Science, 340, 280−281, 2013. Bond, T. C., Doherty, S. J., Fahey, D. W., Forster, P. M., Berntsen, T., DeAngelo, B. J., Flanner, M. G., Ghan, S., Kärcher, B., Koch, D., Kinne, S., Kondo, Y., Quinn, P. K., Sarofim, M. C., Schultz, M. G., Schultz, M., Venkataraman, C., Zhang, H., Zhang, S., Bellouin, N., Guttikunda, S.K., Hopke, P. K., Jacobson, M. Z., Kaiser, J. W., Klimont, Z., Lohmann, U., Schwarz, J. P., Shindell, D., Storelvmo, T., Warren, S. G., Zender, C. S.: Bounding the role of black carbon in the climate system: A scientific assessment, J. Geophys. Res. Atmos., 118, 5380–5552, doi:10.1002/jgrd.50171, 2013. Ramanathan, V. and Carmichael, G.: Global and regional climate changes due to black carbon, Nat. Geosci., 1, 221-227, doi:10.1038/nego156, 2008.

Page 2 Line 16: ": : : snow covered range" ! snow covered region? Answer: It is "snow covered region". (Page 2, Line 26)

Line 20: "Confirming radiative transfer modeling". It is not very clear to readers what does the authors mean. Please consider revise this. Answer: The sentence has been revised as following (Page 2, Line 23-31): In the European Alps, observed reflectance provided evidence that seasonal input of dust can strongly decrease the spectral properties of snow, in particular, from 350 to 600 nm (Di Mauro et al., 2015, 2017). In a study of two sites on Claridenfirn of the Swiss Alps, Gabbi et al. (2015) showed that mineral dust lowered the mean annual albedo by 0.006–0.011, depending on the location on the glacier, causing a reduction of approximately 142–271 mm in annual mass balance. In the San Juan Mountains of the USA, snow cover duration in a seasonal snow-covered region, was found to be shortened by 18 to 35 days during ablation through surface shortwave RF caused by deposition of disturbed desert dust (Painter et al., 2007). In the upper Colorado River Basin, the daily spring dust RF ranged from 30-65 W m-2, advancing melt by 15-49 days (Skiles et al., 2015). A study of the Mera glacier on the southern slope of the Himalayas even indicated that, when dust concentrations are high, dust dominates absorption, snow albedo reduction, and RF, and the impact of BC may be negligible (Kaspari et al., 2014). The presence of dust in snow

suggests a relevant role for BC in darkening the glacier surface.

References: Di Mauro, B., Fava, F., Ferrero, L., Garzonio, R., Baccolo, G., Delmonte, B., Colombo, R.: Mineral dust impact on snow radiative properties in the European Alps combining ground, UAV, and satellite observations, J. Geophys. Res. Atmos., 120, 6080–6097, doi:10.1002/2015JD023287, 2015. Di Mauro, B., Baccolo, G., Garzonio, R., Giardino, C., Massabò, D., Piazzalunga, A., Rossini, M., Colombo, R.: Impact of impurities and cryoconite on the optical properties of the Morteratsch glacier (Swiss Alps), Cryosphere Discuss., doi:10.5194/tc-2017-66, 2017. Kaspari, S., Painter, T. H., Gysel, M., Skiles, S. M., Schwikowski, M.: Seasonal and elevational variations of black carbon and dust in snow and ice in the Solu-Khumbu, Nepal and estimated radiative forcings, Atmos. Chem. Phys., 14, 8089–8103, 2014.

Line 24: "..., but also to" ! but also important/crucial to Answer: Has been revised. (Page 2, Lin35)

Line 26: ": : : is the most sensitive" how could tell it is the most sensitive? Please remove most or provide supporting data. Answer: Removed the "most".

Line 33: ": : : and result in perturbation in" ! : : :and perturb Answer: Has been revised. (Page 3, Line 7)

Line 34: "5 – 25 mm in the snow water equivalent: : :"!5-25 mm snow water equivalent Answer: Has been revised. (Page 3, Line 8)

Page 3 Line 1: "snowpack on the TP is associated with the : : :" by associated you mean the snowpack on the TP is influenced by summer monsoon? Or the snowpack will influence summer monsoon? or both? Please be more accurate here. Answer: Changes in snow cover over the TP have attracted much attention in recent years owing to climate change (Xu et al., 2017). Previous studies have indicated that changes in snow cover across the TP can markedly affect summer monsoons and precipitation over the Indian Ocean (Bai and Feng, 1994; Chen et al., 2000; Vernekar et al.,

1995). Through the analysis of spatial and temporal variability of Tibetan snow cover and its relationship with the Indian summer monsoon, Zhao and Moore (2004) showed that there existed an east-west dipole-like correlation pattern between snow cover over the TP and Indian monsoon rainfall that underwent a change in sign around 1985. They also argued that variability in the TP monsoon was responsible for the spatial and temporal variability in the relationship between Tibetan snow cover and the Indian summer monsoon. The relationship between TP snow cover and the East Asian summer monsoon indicated that winter snow cover played an important role in cooling local air temperature through the snow–albedo effect (Xiao and Duan, 2016). However, data analysis demonstrated that persistent effects of winter snow cover were limited to the period from winter to spring over most parts of the central and eastern TP. Therefore, the preceding snow cover over these regions exerted little influence over either the in situ summer atmospheric heat source or the East Asian summer monsoon, because of its short duration. In contrast, the effects of winter or spring snow cover anomalies over the western TP and the Himalayas can last until summer, and these anomalies further influenced the East Asian summer monsoon by modulating moisture transport to eastern China and favoring eastward-propagating synoptic disturbances that were generated over the TP. Here in the main text has been revised as following (Page3, Line 10-13): Changes in snow cover over the TP have attracted much attention in recent years owing to climate change (Xiao and Duan, 2016; Xu et al., 2017). Snow cover on the TP plays an important role on the Asian summer monsoon, and serves as a crucial water source for several major rivers of Asia (Bai and Feng, 1994; Lau et al., 2010; Vernekar et al., 1995; Yao et al., 2012a; Zhao and Moore, 2004).

References: Bai, Y., and Feng, X.: Introduction to some research work on snow remote sensing, Remote Sens. Technol. Appl., 12(2), 59−65, 1994. Chen, Q., Gao, B., Li W., Liu, Y.: Studies on relationships among snow cover winter over the Tibetan Plateau and droughts/floods during Meiyu season in the middle and lower reaches of the Yangtze River as well as atmosphere/ocean. Acta Metall. Sin., 58(5), 582−592, 2000. Lau, W. K. M., Kim, M.-K., Kim, K.-M., Lee, W.-S.: Enhanced surface warming and accelerated snow melt in the Himalayas and Tibetan Plateau induced by absorbing aerosols, Environ. Res. Lett., 5, 025204, doi:10.1088/1748-9326/5/2/025204, 2010. Vernek, A. D., Zhou, J., Shukla, J.: The effect of Eurasian snow cover on the Indian Monsoon, J. Clim., 8, 248–266, 1995. Xiao, Z., and Duan, A.: Impacts of Tibetan Plateau snow cover on the interannual variability of the East Asian summer monsoon, J. Clim., 29, 8495−8514, doi:10.1175/JCLI-D-16-0029.1, 2016. Xu, W., Ma, L., Ma, M., Zhang, H., Yuan, W.: Spatial-Temporal variability of snow cover and depth in the Qinghai-Tibetan Plateau, J. Clim., 30, 1521−1533, doi:10.1175/JCLI-D-15-0732.1, 2017. Yao, T., Thompson, L.G., Mosbrugger, V., Zhang, F., Ma, Y., Luo, T., Xu, B., Yang, X., Joswiak, D. R., Wang, W., Joswiak, M. E., Devkota, L. P., Tayal, S., Jilani, R., Fayziev, R.: Third Pole Environment (TPE), Environ. Development, 3, 52–64, doi: 10.1016/j.envdev.2012.04.002, 2012a. Zhao, H., and Moore, G.W.K.: On the relationship between Tibetan snow cover, the Tibetan Plateau monsoon and the Indian summer monsoon, Geophys. Res. Lett., 31(L14204), doi:10.1029/2004GL020040, 2004.

Line 4-5: "Simulation studies of BC in snow over TP have inherent uncertainties because of the lack of large-area observations of BC data in seasonal snow cover". This is not correct. Model simulations have inherent uncertainties due to the physics/chemistry/transport schemes used in the models - such as uncertainties in BC emissions sources or deposition rate. Large-area observation of BC in TP will be useful for model evaluation, but it will not help with the inherent uncertainty of the model simulation. Please revise it here. Answer: We agree that large-area observation of BC in TP will be useful for model evaluation. The sentence has been changed to be as following (Page 3, Line15-23): Simulation studies of BC in snow over the TP have inherent uncertainties due to the physics/chemistry/transport scheme used in the models (Gertler et al., 2016; Yasunari et al., 2010, 2013). For example, Kopacz et al. (2011) estimated RF of 5 to 15 W m−2 due to BC within the snow-covered areas of Himalaya and the TP, whereas Flanner et al. (2007) and Qian et al. (2011) estimated peak values of BC effects exceeding 20 W m−2 for some parts of the TP. Menon et al. (2010)

and Ménégoz et al. (2014) proposed that during the last decade BC in snow caused a significant part of the decrease of the snow cover extent or duration observed on the TP during the last decade. Ji et al. (2016b) found a positive surface RF was induced by dust, which caused a decrease of 5–25 mm w.e. over the western TP, Himalayas, and Pamir Mountains from December to May. Large-area observations of BC data in snow cover, which are still seldom in the TP, can be useful for model evaluation and calibration in the future.

Page 4 Line10: " : : : and is dominated by the Indian monsoon" ! A region cannot be dominated by monsoon. Do you mean "affected by"? Answer: It should be "affected by". The sentence has been changed in the main text (Page4, Line27).

Line 11: "Region III has one site (LHG) is located in the northeastern part of the TP"! "Region III has one site (LHG) that is located in the northeastern part of the TP" Answer: Has changed in the main text as following (Page4, Line28-29). Intensified sampling has been carried out in the LHG region within Region III, which is located in the northeastern part of the TP.

Line 15: it seems the snow samples were only collected from the top 5 cm, so how did you calculate the snow albedo for snowpacks thicker than 5 cm? Did you assume the BC/MD mass mixing ratio is constant through the entire snow column? Please clarify this. Answer: We did not measure the BC/MD mass mixing ratio through the entire snow column. Usually, snow samples were collected from the top 5−10 cm. In general, the snow cover is vertically inhomogeneous, and sometimes it is optically thin (Voisin et al., 2012). The stratified structure of snow cover will lead to the discrepancy between computed spectral albedo from semi-infinite snow cover and measured albedo (Grenfell, 1994; Aoki et al., 2000; Zhou et al., 2003; Kuipers Munneke et al., 2008). Recently, the propagation of light in snow with impurities has been extensively investigated with different physical-based numerical models: WW (Grenfell et al., 1994; Jin et al., 2008; Nair et al., 2013), SNICAR (Flanner et al., 2005), PBSAM (Aoki et al., 2000; 2012), GOSWIM (Yasunari et al., 2012; 2014), TARTES (Libios et al., 2013). In this study,

we use SNICAR model to simulate the albedo. This simulator is a single-layer implementation of the Snow, Ice, and Aerosol Radiation model (Flanner et al., 2007, 2009), which utilized the two-stream radiative transfer solution (Toon et al., 1989). Therefore, in this study, we only focused on the surface albedo based on SNICAR model which requests a snow depth of 5 cm. When the snow cover was thicker than 5 cm, the input parameter of "snowpack thickness" (Fig. R1) was the observed snow cover depth.

In the main text (section 2.5), we have added the related information. (Page7, Line24-28): In general, the snow cover is vertically inhomogeneous, and sometimes it is optically thin (Voisin et al., 2012). The stratified structure of snow cover will lead to the discrepancy between computed spectral albedo from semi-infinite snow cover and measured albedo (Grenfell, 1994; Aoki et al., 2000; Zhou et al., 2003; Kuipers Munneke et al., 2008). In this study, we usually collected snow samples from the top $5-10$ cm. Because the SNICAR model is a single-layer implementation model (Flanner et al., 2007, 2009), the input parameter of "snowpack thickness" was the observed snow cover depth.

Figure R1. The information of SNICAR model input parameters.

References: Aoki, T., Aoki, T., Fukabori, M., Hachikubo, A., Tachibana, Y., Nishio, F.: Effects of snow physical parameters on spectral albedo and bidirectional reflectance of snow surface, J. Geophys. Res, 105(D8), $10219-10236$, 2000. Flanner, M. G., Zender, C. S., Hess, P. G., Mahowald, N. M., Painter, T. H., Ramanathan, V., Rasch, P. J.: Springtime warming and reduced snow cover from carbonaceous particles, Atmos. Chem. Phys., 9, 2481–2497, 2009. Flanner, M. G., Zender, C. S., Randerson, J. T., Rasch, P. J: Present-day climate forcing and response from black carbon in snow. J. Geophys. Res., 112, D11202, 2007. Grenfell, T. C., Warren, S. G., Mullen, P. C.: reflection of solar radiation by the Antarctic snow surface at ultraviolet, visible, and near-infrared wavelengths, J. Geophys. Res, 99(D9), $18669-18684$, 1994. Kuipers Munneke, P., Reijmer, C. H., van den Broeke, M. R., König-Langlo, G., Stammes, P., Knpa, W. H.: Analysis of clear-sky Antarctic snow albedo using ob-

servations and radiative transfer modelling, J. geophys. Res- Atmos., 113(D17118), doi:10.1029/2007JD009653, 2008. Voisin, D., Jaffrezo, J.-L., Houdier, S., Barret, M., Cozic, J., King, M. D., France, J. L., Reay, H. J., Grannas, A., Kos, G., Ariya, P. A., Beine, H. J., Domine, F.: Carbonaceous species and humic like substances (HULIS) in Arctic snowpack during OASIS field campaign in Barrow, J. Geophys. Res., 117(D00R19), doi:10.1029/2011JD016612, 2012. Zhou, L., Dickinson, R. E., Tian, Y., Zeng, X., Dai, Y., Yang, Z.-L., Schaaf, C. B., Gao, F., Jin, Y., Strahler, A., Myneni, R. B., Yu, H., Wu, W., Shaikh, M.: Comparison of seasonal and spatial variations of albedos from Moderate-Resolution Imaging Spectradiomater (MODIS) and Common Land Model, J. Geophys. Res, 108(D15), doi:10.1029/JD003326, 2003.

Line 17: how did you measure snow grain size in the field? Please clarify this. Answer: In the field, the snow grains were sprayed on the MIG paper. Then we took a photo by using portable digital microscope (Anyty 3R-MSV500) (Fig. R2a). In the lab, we can measure the snow grain shape and obtain the lengths of a and b (Fig. R2b). Mugnai and Wiscombe (1987) demonstrated that a collection of unoriented non-spheroids produce the same scattering results as spheres, and Grenfell and Warren (1999) showed that the radius of a non-spherical particle was equal to that of a spherical particle that has the same volume-to-surface-area (V/A) ratio. Consequently, V/A ratio was used to transfer the measured snow grain size into the effective grain size. On this basis of field measurement, Hao (2009) proposed two assumptions: 1) The snow particle is an inequilateral spheroid; and 2) The major axis, minor axis, and height of the inequilateral spheroid is denoted by a, b, and c, respectively, in such a way that the relationship a = 2b exists. According to Kokhanovsky and Zege (2004), the effective snow grain radius (Ref) can be calculated equal to the radius of the volume-to-surface equivalent sphere by the following equation (1): $R\_ef=(3V \v{I}\v{E})/(4A \v{I}\v{E})$ (1) Where, $V \v{I}\v{E}=4/3 \pi r^3$ and $A \v{I}\v{E}=\pi r^2$, are the average volume and the average cross-section (geometric shadow) area of the snow grains, respectively. And, r is the radius of geometric optics, $r=(a+b)/4$. Thus, $R\_ef \approx 0.35a$ Eq.1 was then employed to calculate the effective snow grain size.

[Figure]

We have added related information in the main text (Page 5, Line2-12): For the snow grain size observation, we sprayed the snow grains on MIG paper and took a photo using a portable digital microscope (Anyty 3R-MSV500) to calculate the major and minor axis (Fig. S2). Based on field measurements from previous studies (Mugnai and Wiscombe, 1980; Grenfell and Warren, 1999), two assumptions were proposed: 1) The snow grain particle is an inequilateral spheroid, and 2) The major axis (a), minor axis (b), and height (c) of the inequilateral spheroid are denoted by a, b, and c, respectively, in such a way that the relationship a=2b exists (Hao, 2009). According to Kokhanovsky and Zege (2004), the effective snow grain radius (Ref) can be calculated as equal to the radius of the volume-to-surface equivalent sphere by the following equation (1): R_ef=(3V Ì Ě)/(4A Ì Ě ) (1) where V Ì Ě=4/3 $\pi r^3$ and A Ì Ě=$\pi r^2$, are the average volume and the average cross-section (geometric shadow) area of the snow grains, respectively. In addition, r is the radius of geometric optics, r=(a+b)/4. Thus R_ef≈0.35a (2)

Figure R2. Snow grain size observation in the field (a) and measurement in the lab (b).

References: Grenfell, T. C., and Warren, S. G.: Representation of a nonspherical ice particle by a collection of independent sphere for scattering and absorption of radiation, J. geophys. Res., 104(D24), 31697–31709, 1999. Hao, X.: Retrieval of alpine snow cover area and grain size basing on optical remote sensing. PhD thesis, Cold and Arid Regions Environment Engineering Research Institute, Lanzhou, China, pp. 103–104, 2009. Kokhanovsky, A. A., and Zege, E. P.: Scattering optics of snow, Appl. Optics, 43(7), 1589–1602, 2004. Mugnai, A., and Wiscombe, W. J.: Scattering of radiation by moderately nonspherical particles. J. Atmos. Sci., 37, 1291–1307, 1987.

Line 24-26: what is the accuracy of the weight you used to measure filter before and after filtration? It seems the author assume BC/OC/MD are the only insoluble particles deposited on the quartz filter, could you please provide more evidence about this? If MD were the only other particle in snow (besides OC and BC), do all MD absorb sunlight? It might be a good idea to include these discussions in the uncertainty analysis.

Answer: The accuracy of the weight for measurement of filter before and after filtration is less than ±5%. We assumed BC/OC/MD are the only insoluble particles deposited on the quartz filter. Not all MD absorb sunlight. Iron oxide minerals are efficient light scattering and absorption materials which can enhance absorption at UV and visible wavelengths (Di Mauro et al., 2015; Moosmuller et al., 2012; Zhang et al., 2015). Goethite and hematite are the most abundant forms of iron oxides in dust and the major light absorbers in the shortwave spectrum in snow (Wu et al., 2016a). The light absorption by mineral dust in snow is thought to be due to iron oxides (Wang et al., 2013). Two peaks in the first derivative value of the spectra at 430 and 560 nm were determined to be goethite and hematite, respectively. When the iron content reaches a threshold, the iron oxides have little or no impact on the reflectance spectra (Wu et al., 2016a). The fine fraction of glacier dust has a greater abundance of iron (2016b), and the first derivative values of hematite are higher than goethite, indicating that hematite might be concentrated in the fine fraction (2016a). Dust optical properties depend strongly on source material and these properties are designed to represent "global-mean" characteristics as closely as possible (Flanner et al., 2007). The SNICAR model applies the Maxwell-Garnett approximation for combining indices of refractions, assuming a mixture of quartz, limestone, montmorillonite, illite, and hematite. We agree that some dust particles (e.g., those containing a large proportion of strongly-absorbing hematite) can have a larger impact on snow albedo than the dust applied in this work (Aoki et al., 2006; Painter et al., 2007).

We have added related discussion in the section 3.4 as following (Page12, Line25-33): We also have to pay attention to the fact that dust optical properties depend strongly on source material and their properties are designed to represent "global-mean" characteristics as closely as possible (Flanner et al., 2007). The light absorption by mineral dust in snow is thought to be due to iron oxides (Wang et al., 2013), which are efficient light scattering and absorption materials that can enhance absorption at UV and visible wavelengths (Di Mauro et al., 2015; Moosmüller et al., 2012; Zhang et al., 2015). Goethite and hematite are the most abundant forms of iron oxide in dust and

the major light absorbers in the shortwave spectrum in snow (Wu et al., 2016). The SNICAR model applies the Maxwell-Garnett approximation for combining indices of refractions, assuming a mixture of quartz, limestone, montmorillonite, illite, and hematite. We should note that some dust particles (e.g., those containing a large proportion of strongly absorbing hematite) can have a larger impact on snow albedo than the dust applied in this work (Aoki et al., 2006; Painter et al., 2007).

References: Aoki, T., Motoyoshi, H., Kodama, Y., Yasunari, T. J., Sugiura, K., Kobayashi, H.: Atmospheric aerosol deposition on snow surfaces and its effect on albedo, SOLA, 2013–2016, doi:10.2151/sola.2006-004, 2006. Baccolo, G., Di Mauro, B., Massabò, D., Clemenza, M., Nastasi, M., Delmonte, B., Prata, M., Prati, P., Previtali, E., Maggi, V.: Cryoconite as a temporary sink for anthropogenic species stored in glaciers, Sci. Rep., 7, 9623, doi:10.1038/s41598-017-10220-5, 2017. Di Mauro, B., Fava, F., Ferrero, L., Garzonio, R., Baccolo, G., Delmonte, B., Colombo, R.: Mineral dust impact on snow radiative properties in the European Alps combining ground, UAV, and satellite observations, J. Geophys. Res. Atmos., 120, 6080–6097, doi:10.1002/2015JD023287, 2015. Moosmüller, H., Engelbrecht, J. P., Skiba, M., Frey, G., Chakrabarty, P. K., Arnott, W. P.: Single scattering albedo of fine mineral dust aerosols controlled by iron concentration, Journal of Geophys. Res., 117(D11210), doi:10.1029/2011JD016909, 2012. Painter, T. H., Barrett, A. P., Landry, C. C., Neff, J. C., Cassidy, M. P., Lawrence, C. R., McBride, K. E., Farmer, G. L.: Impact of disturbed desert soils on duration of mountain snow cover, Geophys. Res. Lett., 34, L12502, doi:10.1029/2007GL030284, 2007. Wang, X., Doherty, S., Huang, J.: Black carbon and other light-absorbing impurities in snow across Northern China, J. Geophys. Res. Atmos., 118, 1471-1492, doi:10.1029/2012JD018291, 2013. Wu, G., Xu, T., Zhang, X., Zhang, C., Yan, N.: The visible spectroscopy of iron oxide minerals in dust particles from ice cores on the Tibetan Plateau, Tellus B, 68, 29191, doi: 10.3402/tellusb.v68.29191, 2016a. Wu, G., Zhang, X., Zhang, C., Xu, T.: Mineralogical and morphological properties of individual dust particles in ice cores from the Tibetan Plateau, J. Glaciol., 62(231), 46–53, 2016b. Zhang, X., Wu, G., Zhang, C., Xu, T.,

Zhou, Q.: What is the real role of iron oxides in the optical properties of dust aerosols? Atmos. Chem. Phys., 15, 12159–12177, 2015.

Page 5: Line 6: what is the filter blank for? is this a blank filter? It seems there is some particles on filter blank as well since it weighs more than 0 C/cm-2. Answer: We used the filter blank to detect the sampling and storage processes. It is a blank filter. The blank filters were brought to the field and stored with the filters together. The results showed that the filter blank is much less (1.23±0.38 $\mu$g C cm–2) than those in the sampled filters (more than 10 $\mu$g C cm–2), indicating the OC/BC data were reasonable.

Line 8: "separately analyzed" ! analyzed separately Answer: Has been changed.( (Page6, Line2)

Line 18: ": : : is or not influenced by BC emissions" ! is influenced by BC emissions or not Answer: Has been changed.( (Page6, Line21)

Line 23: what is "down-sun"? Answer: It's type wrong. Spectral albedo measurements were collected using an ASD FieldSpec 4 (FS4) standard-resolution spectroradiometer in the range of 350–2500 nm using the ASD Remote Cosine Receptor (RCR) foreoptic. The RCR foreoptic was mounted to the end of a 91.4-cm rectangular aluminium boom and levelled at a height of 80 cm above the snow surface using a bubble level located at the observer's end of the boom, collecting sky irradiance when pointed upward and snow irradiance when pointed downward. Spectral albedo was obtained by dividing snow irradiance by sky irradiance from ten consecutive upward and downward measurements collected at site around local solar noon determined using the NOAA SolarNoonCalculator.

We have changed in the main text (Page7, Line2-9). For the selected sites, a general-purpose spectroradiometer (Analytical Spectral Devices (ASD), FieldSpec 4, Inc.), was used to measure the reflectance of snow cover. The ASD FieldSpec 4 instrument is a general-purpose spectroradiometer that is useful for applications requiring the

measurement of reflectance, transmittance, radiance, or irradiance. The instrument is specifically designed for field environment remote sensing to acquire visible and near-infrared and shortwave infrared spectra. It has 3 nm spectral resolution on the visible/near infrared detector (350–1050 nm, silicon photodiode array), and 10–12 nm resolution on the shortwave infrared detectors (900–2 500 nm, InGaAs). In the field, reflectance was measured at two sites (24K and MD in Table S1) with FieldSpec 4 under clear-sky conditions. These measurements of reflectance were calculated using the standard solar irradiance to get the albedos, which were then used for comparison with simulated albedos.

Line 24: "when the weather was clear"! when the sky was clear. It seems the albedo measurement were made for clear sky only, but later in the albedo comparison, the measured albedo is compared against the albedo calculated using SNICAR for all-sky case. Is this a typo or this is wrong? Answer: Has been changed. The "all-sky" is a type wrong (see Table S2 in supplementary material), we have changed in the main text.

Line 28: "e.g. Doherty et al., 2010": Doherty et al 2010 did not use SNICAR, it only reported the observation in the Arctic. Please remove this citation. Answer: Has removed.

Page 6: Line 9-11: Please revise this sentence. Is this a model or is this just the method you used to calculate snow water melt in this work? Answer: It is a method we used to calculate snow water melt. We have changed the sentence in the main text (Page7, Line31-32). To estimate snow melt due to BC and dust, a method was constructed in which the absorptivity of the snow was multiplied by the daily average incoming shortwave radiation from the automatic weather stations (AWS) set up at the meteorological stations near the sampling sites (Fig. S3) (Schmale et al., 2017).

Line 16: "for clean snow: : :" what is cause of albedo reduction for clean snow case? A different snow grain size due to snow aging? Please be more accurate here. Answer:

Yes. For clean snow, different snow grain size played an important role on the albedo reduction. We have revised the sentence in the main text (Page8, Line7-8): $\Delta\alpha$ is the albedo reduction caused by BC and dust, for clean snow mainly by different snow grain sizes due to snow aging;

Line 18: "assumed snow depth" what is the assumed snow depth? How did you assume it? Answer: In the following analysis, we assumed the snow depth to be consistent (which is not the reality) for estimating the variations of snow cover durations days in section 3.3. This assumption was based on the snow depth observations from the TP in previous studies (Che et al., 2012; Xu et al., 2017; Zhong et al., 2016). We have changed this sentence in the main text (Page8, Line8).

References: Che, T., Dai, L., Wang, J., Zhao, K., Liu, Q.: Estimation of snow depth and snow water equivalent distribution using airborne microwave radiometry in the Binggou Watershed, the upper reaches of the Heihe River basin, Int. J. Appl. Earth Obs. Info., 17, 23–32, doi:10.1016/j.jag.2011.10.014, 2012. Xu, W., Ma, L., Ma, M., Yuan, W.: Spatial-temporal variability of snow cover and depth in the Qinghai-Tibetan Plateau, J. Clim., doi:10.1175/JCLI-D-150732.1, 2017. Zhong, X., Zhang, T., Zheng, L., Hu, Y., Wang, H., Kang, S.: Spatiotemporal variability of snow depth across the Eurasian continent from 1966 to 2012, Cryosphere Discuss., doi:10.5194/tc-2016-182, 2016.

Line 22: "older wind-packed snow" ! old wind-packed snow Answer: Has been changed.( Page8, Line12)

Page 7: Line 2: Please consider cite paper Doherty et al., 2016. It also reports LAIs in snow in North America. Doherty,S. J., D. A. Hegg, J. E. Johnson, P. K. Quinn, J. P. Schwarz, C. Dang, and S. G. Warren, (2016), Causes of variability in light absorption by particles in snow at sites in Idaho and Utah. J. Geophys. Res. Atmos., 121, no. 9: 4751-4768. Answer: Agree, and has cited this paper.

Line 6-8: How could you tell the LAIs in snow over TP were generated from biomass burning in surrounding region from Table 2? And from Table 2, how could you tell if

the fraction of LAIs emitted by biomass/fossil fuel burning in TP is larger than that in the other regions? Please provide more explanations. Answer: Table 2 shows the concentrations of LAIs in snow cover in the TP and other regions. These comparisons indicate deposition of LAIs in the snow over the TP have higher values than other regions. In this section, we will not to discuss their fraction and different sources. Thus, we have revised this sentence in the main text as following (Page8, Line27-29): In comparison (Table 2), the LAIs in snow cover of the TP showed larger values which can be attributed to more deposition from nearby regions around the TP (e.g., South Asia, East Asia, and/or western China) (Lu et al., 2012; Ramanathan and Carmichael, 2008), or to impacts by the soil/dust near the sampling sites.

Line 9-11: How could the ratio of OC/BC be used as a standard to determine the emission sources of biomass burning? What is the OC/BC ratio if all LAIs were emitted from biomass burning or non-biomass burning sources? Answer: Usually, the aerosols emitted from biomass burning have higher OC/EC ratios. For example, Watson et al. (2001) have reported an OC/EC ratio of 14.5 for forest fires; while for the fossil fuel, OC/EC ratio was about 1 (Watson et al., 2001). The OC/EC ratios at Mt. Everest station range from 1.91 to 43.8 with an average of 6.69. Such high ratios are considered to be mainly affected by the strong influence of biomass-burning emissions (Cong et al., 2015a), which is also evidenced by the atmospheric organic acids (Cong et al., 2015b). In the southeastern TP, OC was found to be more abundant during the monsoon season (with OC/EC ratios are 17.67) than in other seasons (e.g. pre-monsoon and winter with OC/EC ratios of 6.29 and 6.45, respectively). These trends can be explained by the emission of plant spores and pollen as well as the formation of greater quantities of secondary organic carbon (SOC) in the periods with the higher OC loadings (Zhao et al., 2013). We have added related information in the main text as following (Page8, Line30-32): Ratios of OC to BC (OC/BC) were used to represent the possible impact of biomass burning in previous studies (Watson et al., 2001; Bond et al., 2013; Cong et al., 2015a, 2016b). Usually, the aerosols emitted from biomass burning have higher OC/EC ratios. For example, Watson et al. (2001) reported an OC/EC ratio of 14.5 for

forest fires; whereas for fossil fuel, the OC/EC ratio was approximately 1.

References: Cong, Z., Kawamura, K., Kang, S., Fu, P.: Penetration of biomass-burning emissions from South Asia through the Himalayas: new insights from organic acids, Sci. Rep., 5, 9580, doi:10.1038/srep09580, 2015b. Cong, Z., Kang, S., Kawamura, K., Liu, B., Wan, X., Wang, Z., Gao, S., Fu, P.: Carbonaceous aerosols on the south edge of the Tibetan Plateau: concentrations, seasonality and sources, Atmos. Chem. Phys., 15, 1573-1584, 2015a. Watson, J. G., Chow, J. C., Houck, J. E.: PM2.5 chemical source profiles for vehicle exhaust, vegetative burning, geological material, and coal burning in Northwestern Colorado during 1995, Chemos., 43, 1141–1151, 2001. Zhao, Z., Cao, J., Shen, Z., Xu, B., Zhu, C., Chen, L.-W. A., Su, X., Liu S., Han, Y., Wang, G., Ho, K.: Aerosol particles at a high-altitude site on the Southeast Tibetan Plateau, China: Implications for pollution transport from South Asia, J. Geophys. Res-Atmos., 118, 11360–11375, 2013.

Line 15: "LHG3 AND LHG6 (Figure 3)", do you mean Figure 1b? Answer: Yes, it should be Figure 1b.

Line 15-21: It seems Figure S2 is an important figure for discussions in this part. If there is no restriction on the number of figures, please consider include this figure in the paper. Answer: Agree, we have included this figure in the main text as Figure 3.

Line 28: "Open burning sourced BC"! BC emitted from open burning sources. Does the BC emitted from open burning sources contain BC emitted from biomass burning? or are they the same? Please be more specific about what you mean by "open burning". Answer: They are not exactly the same. Here in this study, the results was based on the data of FINN fire emission. Thus, we mean "fire spots or open fire", kind of biomass burning. We have changed this sentence in the main text as following (Page9, Line19-23): Contributions of BC from open fire burning (a kind of biomass burning) decrease from the southern to the northern TP (Fig. 5). In the Himalayan region, BC from biomass burning sources accounts for half of the BC deposition on snow cover,

reflecting the proximity to large sources in South Asia. In the central TP, BC from open fire burning accounts for approximately 30 % of the total, less than from biomass burning evidence from aerosols and glacial snow (Li et al., 2016), maybe indicating a lack of biofuel contributions from our calculation.

Page 8 Line 3-4: "In the southern TP: : :. due to influence of BC sources in the south Asia". This statement is too vague. Do you mean the total emission is larger in southern Asia? or the BC deposition rate is larger ? or both? This is a major conclusion you have in this work, please be more specific. Answer: Here we mean the total emission is larger in southern Asia (see from the AOD in Fig. R3) rather than the BC deposition rate. We have changed this sentence in the main text (Page9, Line28-33): In the southern TP (MYL), the amount of BC deposition is larger than in the northern TP (NETP) due to the nearby BC sources in south Asia, known to be a regional hotspot of BC-induced atmospheric solar heating (Ramanathan and Carmichael, 2008). Whereas in the central TP (NMC), the amount of BC deposition was higher than in the NETP. This may be because pollutants from the southern side of the Himalayas can traverse the high mountain range not only through the major north-south river valleys but also by being lifted and advected over the Himalayas to reach to the inland TP (Cong et al., 2015b; Lüthi et al., 2015).

Figure R3. MODIS atmospheric optical depth (AOD) fields derived using dark target algorithms over the TP and its surroundings. (http://giovanni.sci.gsfc.nasa.gov).

Line 8: You used SNICAR model to calculate snow albedo for all-sky cases, but the model set up is not stated. For example, what is the cloud fraction for all sky case? As mentioned in the previous comments, the albedo measurement was performed under clear sky, so in what accuracy do you expect this to agree with the SNICAR calculation (Figure 5)? Answer: Here is not the all-sky cases, it should be clear sky. We have changed in the main text.

Line 14: "The deviation between measured and simulated reflectance by MD may be

a result of the upper boundary of the SNICAR model in particle dimension". This does not make sense, is the deviation due to upper boundary condition (and what upper boundary condition)? or MD particle size? Please revise. Answer: The difference between measured and simulated reflectance by MD may be caused by the upper boundary of particle size (as shown in Fig. R1, the dust concentrations) in the SNICAR model (Flanner et al., 2007). Thus, we have changed this sentence in the main text (Page10, Lin10-11).

Line 18-19: "This result is import, showing that the SNICAR model simulations can represent albedo changes of snow cover in the Third Pole region". This is a really strong conclusion. I don't think the authors can make this conclusion based on the results shown in this paper. Especially it is unclear to the readers that how did the authors set up the SNICAR calculation. Please remove this or include more details about SNICAR calculation. Answer: Agree, and removed this sentence.

Line 26. As the author said, the BC snow albedo forcing over TP is highly uncertain partly due to the uncertainty in simulated BC concentration. But it is also important to point out that a large fraction of such uncertainty is resulted form uncertainty in simulated snow-cover fraction/snow depth. Please consider including this in the discussion. Answer: Agree, we have added this section in the discussion (Page13, Line1-10). It is further important to note that a large fraction of such uncertainty can also result from uncertainty in the simulated snow-cover fraction/snow depth across the plateau. The TP covers a large area (more than 2.5 million km2) with an average elevation exceeding 4000 m a.s.l. (Yao et al., 2012a). Considerable heterogeneity in the topography and climate has led to complex spatial and temporal snow cover patterns (Xu et al., 2017). Most studies on the snow cover distribution were based on satellite-based observations (Che et al., 2008, 2012). Surface observations of snow depth showed an increase over the TP from 1957–1998 (Ma and Qin, 2012). However, snow cover depth and the number of snow covered days during the current decade under intense climate warming showed a decreasing trend mainly occurring in the southeast TP in winter.

Whereas in spring, snow cover depth showed an increasing trend in the eastern TP (Xu et al., 2017). These differences can also affect the estimation in this study. Nevertheless, the method provides a theoretical approach for evaluating how the presence of LAIs affects the lower parts of the glacier subjected to summer melt.

References: Che, T., Dai, L., Wang, J., Zhao, K., Liu, Q.: Estimation of snow depth and snow water equivalent distribution using airborne microwave radiometry in the Binggou Watershed, the uppwe reaches of the Heihe River basin, Int. J. Appl. Earth Obs. Info., 17, 23–32, doi:10.1016/j.jag.2011.10.014, 2012. Che, T., Li, X., Jin, R., Armstrong, R., Zhang, T.: Snow depth derived from passive microwave remote-sensing data in China, Ann. Glaciol., 49, 145–154, doi:10.3189/172756408787814690, 2008. Ma, L., Qin, D.: Temporal–spatial characteristics of observed key parameters of snow cover in China during 1957–2009, Sci. Cold Arid Reg., 4, 384–393, doi:10.3724/SP.J.1226.2012.00384, 2012. Xu, W., Ma, L., Ma, M., Zhang, H., Yuan, W.: Spatial-Temperal variability of snow cover and depth in the Qinghai-Tibetan Plateau, J. Clim., 30, $1521-1533$, doi:10.1175/JCLI-D-15-0732.1, 2017. Yao, T., Thompson, L.G., Mosbrugger, V., Zhang, F., Ma, Y., Luo, T., Xu, B., Yang, X., Joswiak, D. R., Wang, W., Joswiak, M. E., Devkota, L. P., Tayal, S., Jilani, R., Fayziev, R.: Third Pole Environment (TPE), Environ. Development, 3, 52–64, doi: 10.1016/j.envdev.2012.04.002, 2012a.

Page 9 Section 3.3: Why did you pick SW flux of 220, 270 and 310 W m-2? Answer: We selected short-wave radiation from several automatic weather stations (Fig. R4) near the snow sampling sites in region I, II, and III. Table R1 showed monthly short-wave radiation from automatic weather stations near the snow sampling sites during the snow melting season (March-May) when the temperature began to increase. On average, shortwave radiations in March, April, May, and June is about 238, 269, 292, and 271 W m–2, respectively. Short-wave radiation showed the minimum value in March in Tanggula (210 W m–2, lower than the average) and the maximum value in May in Namco (314 W m–2, higher the average). Thus, in order to estimate the impact of input short-wave radiation on the snow cover duration, we gave a range of shortwave radiation from 220 to 310 W m–2.

We have added related information in the main text as following (Page11, Line14-23): Changes of snow cover duration are calculated using a model documented by Schmale et al. (2017) (section 2.6). In Eq. (4), monthly shortwave radiation (SW) input data were obtained from the Automatic Weather Station (AWS) near the snow sampling sites across the TP (Fig. S3). Table S3 shows monthly shortwave radiation data from AWS during the snow melting season (March-May) when the temperature began to increase. On average, shortwave radiation in March, April, May, and June is approximately 238, 269, 292, and 271 W m–2, respectively. Short-wave radiation in March showed the minimum value in Tanggula of the central TP (210 W m–2, lower than the average), whereas in May it showed the maximum value in Namco (314 W m–2, higher the average). Based on these data during the melt season, three scenarios (220, 270, and 310 W m–2)were defined as the minimum, median, and maximum scenarios of input shortwave radiation to estimate its impact on changes of snow cover duration. We have added this information in the supplementary material as Table S3.

Figure R4. Automatic weather stations selected in the Tibetan Plateau.

Table R1. Monthly short-wave radiation from automatic weather stations near the snow sampling sites during the snow melting season.

Line 27: "SD plays an import role" ! SD plays an important role? What do you mean by SD plays a role here? Deeper snow is supposed to melt slower given the same amount of radiative forcing; it is not because snow depths play a role here. Please clarify this point. Answer: We agree that deeper snow is supposed to melt slower given the same amount of radiative forcing. In this study, we calculated the changes of snow cover duration days rather than the total days of snow cover duration. We mean that SD plays an important role on our estimation of numbers of snow cover duration days. Thus, we have changed the sentence in the main text as following (Page11, Line28-29). The result indicated that estimation of changes of snow cover duration
days caused by the same level of LAIs was also affected by different SD.

Page 10: Line 13: "However, the results presented in this study : : :. for which these assumptions are not critical": This is not true. All the quantities listed in this paragraph will influence the snow albedo and most of them will influence the albedo reduction induced by BC. For example: BC-snow internal mixing increases the albedo forcing by 40-60% compared with external mixing (He et al. (2014). The author should discuss the uncertainty of this study resulted from the assumptions they made, instead of claim these quantities will not impact their results. References: He, C., Q. Li, K. N. Liou, Y. Takano, Y. Gu, L. Qi, Y. Mao, and L. R. Leung, 2014: Black Carbon Radiative Forcing over the Tibetan Plateau. Geophy. Res. Lett., 41, 7806-7813, doi: 10.1002/2014GL062191. Answer: Agree, and we have added related discussion in this section in the main text as suggested.

Line 32: "Our study confirms that : : :... and further reduces snow albedo,: : :.": this is not true. BC and other LAI can reduce the snow albedo even if the snow aging process is not accelerated. Please revise this. Answer: Agree, and have changed (Page13, Line26). Our study confirms that BC and other water-insoluble LAIs in snow on land and ice can darken the surface, and further reduces snow albedo, and increases the speed of snow cover melt

Figures3: Is the color bar showing height? Please define the color bar and unit. Answer: Yes, the color bar shows the height. We have added the information in the figure caption.

Figure 4: Footprint analyses for six selected sites over the Tibetan Plateau during the winter season (Nov 2015-Feb 2016). LHG and NETP in the northern Tibetan Plateau (Region III), TGL and NMC in the central Tibetan Plateau (Region II), and MYL and SETP in the southern Tibetan Plateau (Region I). (The right color bar means the height (m a.s.l.). The trajectories starting below 500 hPa were taken into account. Black dots: the air parcel did not pass near a fire during the 96 h prior to arrival at the studied

Interactive
comment

[Figure]

site; Green dots: the air parcel did pass by a fire between -96 h and -48 h, but not afterwards, i.e., the 'contact' to a fire lies back at least 48 h before the air parcel arrived at Renlongba glacier; Magenta dots: contact with fires occurred between -48 h and 0 h before arriving at site; Red dots: contact with a fire before and after -48 h.)

Figure 5: Are MA1-4 measured albedo? Are the dashed lines albedo calculated using SNICAR? Please clarify these details and modify the corresponding text. Answer: In this figure, MA1-4 (solid lines) are measured albedo. Dashed lines mean the albedo calculated using SNICAR. We have added the information in the figure caption.

Figure 6: Measured albedo by ASD (solid lines, MA1-4) and simulated effects of BC and dust on albedo (dashed lines) at the selected snow site on the Tibetan Plateau.

Figure 6-8: please clarify the figure convention in each figure. Do the boxes represent average values from central estimate? For example, in Figure 6, you say the rectangles are central estimate, so what does the box mean? standard deviation of central estimate? maximum and minimum of central estimate? Answer: We have added related information and legend for each figure in the main text.

Please also note the supplement to this comment: https://www.the-cryosphere-discuss.net/tc-2017-111/tc-2017-111-AC1-supplement.pdf

**SNICAR-Online: Snow albedo simulation**

Documentation

1a. Incident radiation:
Direct: ◉
Diffuse: ◯
1b. Solar zenith angle, if incident radiation is direct (0‐89 degrees): 60   degrees

2. Surface spectral distribution:
Mid-latitude winter, clear-sky: ◉
Mid-latitude winter, cloudy: ◯
Summit Greenland, clear-sky: ◯
Summit Greenland, cloudy: ◯

3. Snow grain effective radius (30‐1500 microns): 100   μm

4. Snowpack thickness: 1   meters

5. Snowpack density: 200   kg/m³

6. Albedo of underlying ground:
Visible (0.3‐0.7 μm): 0.2   Near-infrared (0.7‐5.0 μm): 0.4

7. Black carbon concentration (ppb, or nanograms of BC per gram of ice):
Uncoated: 0   ppb.   MAC scaling factor (experimental): 1.0
Sulfate-coated: 0   ppb

8. Dust concentration (ppm, or micrograms of dust per gram of ice):
Size 1 (0.1‐1.0 μm diameter): 0   ppm
Size 2 (1.0‐2.5 μm diameter): 0   ppm
Size 3 (2.5‐5.0 μm diameter): 0   ppm
Size 4 (5.0‐10.0 μm diameter): 0   ppm

9. Volcanic ash concentration (ppm, or micrograms of ash per gram of ice): 0   ppm

10. Experimental particle 1 concentration (ppb, or nanograms of particle per gram of ice): 0   ppb

Click "Submit" to display spectral albedo and solar broadband (0.3‐5.0 μm) albedo.
Submit  Reset

**Fig. 1.**

[Figure]

[Figure]

(a) Observe snow grain size

(b) Measure snow grain size

**Fig. 2.**

[Figure]

(a) Summer

Time averaged map of Aerosol Optical Depth 550 nm (Dark Target) Monthly 1deg. (MODIS-Terra MOD08_M3 v051)
over 2015Jul-2015 Aug, Region 66.4453E, 20.0391N, 105.1172E, 45N

(b) Winter

Time averaged map of Aerosol Optical Depth 550 nm (Dark Target) Monthly 1deg. (MODIS-Terra MOD08_M3 v051)
over 2014 Dec-2015 Feb, Region 66.4453E, 20.0391N, 105.1172E, 45N

**Fig. 3.**

[Figure: Topographic map of the Tibetan Plateau region with sampling sites marked by red stars: Laohugou (~40°N, 97°E), Tanggula (~33°N, 92°E), Namco (~30°N, 90°E), and Southeastern TP (~29°N, 97°E). Map extends from 75°E to 105°E longitude and 25°N to 45°N latitude.]

**Fig. 4.**

Table R1. Monthly short-wave radiation from automatic weather stations near the snow sampling sites during the snow melting season.

| Sites | Month of the year | Monthly Short-wave radiation (W·m⁻²) | Month of the year | Monthly Shortwave radiation (W·m⁻²) |
|---|---|---|---|---|
| Southeastern Tibetan Plateau | | | | |
| | March in 2014 | 248 | March in 2015 | 237 |
| | April in 2014 | 288 | April in 2015 | 259 |
| | May in 2014 | 305 | May in 2015 | 280 |
| | June in 2014 | 265 | June in 2015 | 250 |
| Namco | | | | |
| | | | March in 2015 | 264 |
| | | | April in 2015 | 276 |
| | | | May in 2015 | 306 |
| | | | June in 2015 | 314 |
| Tanggula | | | | |
| | March in 2014 | 210 | | |
| | April in2014 | 226 | | |
| | May in 2014 | 245 | | |
| | June in 2014 | 271 | | |
| Laohugou | | | | |
| | March in 2014 | 229 | March in 2015 | 238 |
| | April in2014 | 269 | April in 2015 | 294 |
| | May in 2014 | 311 | May in 2015 | 305 |
| | June in 2014 | 258 | June in 2015 | 269 |

**Fig. 5.**

**Supplement:**

Oct10, 2017

Dear editor,

The manuscript: ***Black carbon and mineral dust in snow cover on the Tibetan Plateau*** by Zhang Y.L. et al.

The manuscript number: tc-2017-111

We greatly appreciate the reviewers' constructive comments to improve the paper. We have revised our manuscript according to these comments (blue color in the main text), and hope the revised manuscript is suitable for publication in The Cryosphere.

The "point to point" response to comments are listed as below.

Sincerely yours,
Yulan Zhang and Shichang Kang
* * *
**Response to Referee #1**

General opinion: The authors report the observation of light-absorbing impurities in snow over Tibetan Plateau. Based on these field observed data, they calculated the albedo reduction induced by black carbon and mineral dust, and the corresponding impact on snow energy budget. The field data reported in this work are valuable for quantifying the impact of light-absorbing particles on snow albedo and the analysis based on this field data are informative. However, some discussion and conclusions given in this paper are not accurate and need to be modified. This manuscript also contains a lot of types/grammar errors (some obvious errors are listed below) that need to be corrected before this manuscript can be considered for publication. Here are the suggestions/comments that the authors may find useful.

Answer: Thank you very much for the comments. We have carefully revised according to the comments. The manuscript has also been improved by a native speaker to reduce the types/grammar errors.

Comments and Questions
Page 1 Line 30-32: BC is recognized as an important climate forcer not only because it absorbs sunlight, but also because a large fraction of BC are emitted from anthropogenic sources. Please be more accurate at here.

Answer: We have revised as following in the main text (Page 1-2):
Atmospheric BC is a distinct type of carbonaceous material from incomplete combustion of biomass/biofuel and fossil fuel. A large fraction of BC is emitted from anthropogenic activities

(Bond et al., 2013). Because BC can absorb solar radiation, influence cloud processes, and alter the melting of snow cover and glaciers, it has been considered to be the second most important climate forcer in the Earth's climate system only after carbon dioxide (Andreae and Ramanathan, 2013; Bond et al., 2013; Ramanathan and Carmichael, 2008).

Answer: Has been revised. (Page 3, Line 8)

Page 3 Line 1: "snowpack on the TP is associated with the : : :" by associated you mean the snowpack on the TP is influenced by summer monsoon? Or the snowpack will influence summer monsoon? or both? Please be more accurate here.

Answer: Changes in snow cover over the TP have attracted much attention in recent years owing to climate change (Xu et al., 2017). Previous studies have indicated that changes in snow cover across the TP can markedly affect summer monsoons and precipitation over the Indian Ocean (Bai and Feng, 1994; Chen et al., 2000; Vernekar et al., 1995). Through the analysis of spatial and temporal variability of Tibetan snow cover and its relationship with the Indian summer monsoon, Zhao and Moore (2004) showed that there existed an east-west dipole-like correlation pattern between snow cover over the TP and Indian monsoon rainfall that underwent a change in sign around 1985. They also argued that variability in the TP monsoon was responsible for the spatial and temporal variability in the relationship between Tibetan snow cover and the Indian summer monsoon.

The relationship between TP snow cover and the East Asian summer monsoon indicated that winter snow cover played an important role in cooling local air temperature through the snow–albedo effect (Xiao and Duan, 2016). However, data analysis demonstrated that persistent effects of winter snow cover were limited to the period from winter to spring over most parts of the central and eastern TP. Therefore, the preceding snow cover over these regions exerted little influence over either the in situ summer atmospheric heat source or the East Asian summer monsoon, because of its short duration. In contrast, the effects of winter or spring snow cover anomalies over the western TP and the Himalayas can last until summer, and these anomalies further influenced the East Asian

summer monsoon by modulating moisture transport to eastern China and favoring eastward-propagating synoptic disturbances that were generated over the TP.

Here in the main text has been revised as following (Page3, Line 10-13):

Changes in snow cover over the TP have attracted much attention in recent years owing to climate change (Xiao and Duan, 2016; Xu et al., 2017). Snow cover on the TP plays an important role on the Asian summer monsoon, and serves as a crucial water source for several major rivers of Asia (Bai and Feng, 1994; Lau et al., 2010; Vernekar et al., 1995; Yao et al., 2012a; Zhao and Moore, 2004).

Page 4 Line10: " : : : and is dominated by the Indian monsoon" ! A region cannot be dominated by monsoon. Do you mean "affected by"?
Answer: It should be "affected by". The sentence has been changed in the main text (Page4, Line27).

Line 11: "Region III has one site (LHG) is located in the northeastern part of the TP"! "Region III has one site (LHG) that is located in the northeastern part of the TP"
Answer: Has changed in the main text as following (Page4, Line28-29).
Intensified sampling has been carried out in the LHG region within Region III, which is located in the northeastern part of the TP.

Line 15: it seems the snow samples were only collected from the top 5 cm, so how did you calculate the snow albedo for snowpacks thicker than 5 cm? Did you assume the BC/MD mass mixing ratio is constant through the entire snow column? Please clarify this.
Answer: We did not measure the BC/MD mass mixing ratio through the entire snow column. Usually, snow samples were collected from the top 5−10 cm. In general, the snow cover is vertically inhomogeneous, and sometimes it is optically thin (Voisin et al., 2012). The stratified structure of snow cover will lead to the discrepancy between computed spectral albedo from semi-infinite snow cover and measured albedo (Grenfell, 1994; Aoki et al., 2000; Zhou et al., 2003; Kuipers Munneke et al., 2008). Recently, the propagation of light in snow with impurities has been extensively investigated with different physical-based numerical models: WW (Grenfell et al., 1994; Jin et al., 2008; Nair et al., 2013), SNICAR (Flanner et al., 2005), PBSAM (Aoki et al., 2000; 2012), GOSWIM (Yasunari et al., 2012; 2014), TARTES (Libios et al., 2013).
In this study, we use SNICAR model to simulate the albedo. This simulator is a single-layer implementation of the Snow, Ice, and Aerosol Radiation model (Flanner et al., 2007, 2009), which utilized the two-stream radiative transfer solution (Toon et al., 1989). Therefore, in this study, we only focused on the surface albedo based on SNICAR model which requests a snow depth of 5 cm. When the snow cover was thicker than 5 cm, the input parameter of "snowpack thickness" (Fig. R1) was the observed snow cover depth.

In the main text (section 2.5), we have added the related information. (Page7, Line24-28):
In general, the snow cover is vertically inhomogeneous, and sometimes it is optically thin (Voisin et al., 2012). The stratified structure of snow cover will lead to the discrepancy between computed spectral albedo from semi-infinite snow cover and measured albedo (Grenfell, 1994; Aoki et al., 2000; Zhou et al., 2003; Kuipers Munneke et al., 2008). In this study, we usually collected snow

samples from the top 5−10 cm. Because the SNICAR model is a single-layer implementation model (Flanner et al., 2007, 2009), the input parameter of "snowpack thickness" was the observed snow cover depth.

Figure R1. The information of SNICAR model input parameters.

Line 17: how did you measure snow grain size in the field? Please clarify this.

Answer: In the field, the snow grains were sprayed on the MIG paper. Then we took a photo by using portable digital microscope (Anyty 3R-MSV500) (Fig. R2a). In the lab, we can measure the snow grain shape and obtain the lengths of a and b (Fig. R2b). Mugnai and Wiscombe (1987) demonstrated that a collection of unoriented non-spheroids produce the same scattering results as spheres, and Grenfell and Warren (1999) showed that the radius of a non-spherical particle was equal to that of a spherical particle that has the same volume-to-surface-area (V/A) ratio. Consequently, V/A ratio was used to transfer the measured snow grain size into the effective grain size.

On this basis of field measurement, Hao (2009) proposed two assumptions: 1) The snow particle is an inequilateral spheroid; and 2) The major axis, minor axis, and height of the inequilateral spheroid is denoted by a, b, and c, respectively, in such a way that the relationship a = 2b exists. According to Kokhanovsky and Zege (2004), the effective snow grain radius ($R_{ef}$) can be calculated equal to the radius of the volume-to-surface equivalent sphere by the following equation (1):

$$R_{ef} = \frac{3\bar{V}}{4\bar{A}} \tag{1}$$

Where, $\bar{V} = \frac{4}{3}\pi r^3$ and $\bar{A} = \pi r^2$, are the average volume and the average cross-section (geometric shadow) area of the snow grains, respectively. And, $r$ is the radius of geometric optics, $r = \frac{a+b}{4}$.

Thus, $R_{ef} \approx 0.35a$

Eq.1 was then employed to calculate the effective snow grain size.

We have added related information in the main text (Page 5, Line2-12):

For the snow grain size observation, we sprayed the snow grains on MIG paper and took a photo using a portable digital microscope (Anyty 3R-MSV500) to calculate the major and minor axis (Fig. S2). Based on field measurements from previous studies (Mugnai and Wiscombe, 1980; Grenfell and Warren, 1999), two assumptions were proposed: 1) The snow grain particle is an inequilateral spheroid, and 2) The major axis (a), minor axis (b), and height (c) of the inequilateral spheroid are denoted by a, b, and c, respectively, in such a way that the relationship a=2b exists (Hao, 2009). According to Kokhanovsky and Zege (2004), the effective snow grain radius ($R_{ef}$) can be calculated as equal to the radius of the volume-to-surface equivalent sphere by the following equation (1):

$$R_{ef} = \frac{3\bar{V}}{4\bar{A}} \tag{1}$$

where $\bar{V} = \frac{4}{3}\pi r^3$ and $\bar{A} = \pi r^2$, are the average volume and the average cross-section (geometric shadow) area of the snow grains, respectively. In addition, $r$ is the radius of geometric optics, $r = \frac{a+b}{4}$.

Thus

$$R_{ef} \approx 0.35a \tag{2}$$

[Figure]

Figure R2. Snow grain size observation in the field (a) and measurement in the lab (b).

Line 24-26: what is the accuracy of the weight you used to measure filter before and after filtration? It seems the author assume BC/OC/MD are the only insoluble particles deposited on the quartz filter, could you please provide more evidence about this? If MD were the only other particle in snow (besides OC and BC), do all MD absorb sunlight? It might be a good idea to include these discussions in the uncertainty analysis.

Answer: The accuracy of the weight for measurement of filter before and after filtration is less than ±5%.

We assumed BC/OC/MD are the only insoluble particles deposited on the quartz filter. Not all MD absorb sunlight. Iron oxide minerals are efficient light scattering and absorption materials which can enhance absorption at UV and visible wavelengths (Di Mauro et al., 2015; Moosmuller et al., 2012; Zhang et al., 2015). Goethite and hematite are the most abundant forms of iron oxides in dust and the major light absorbers in the shortwave spectrum in snow (Wu et al., 2016a). The light absorption by mineral dust in snow is thought to be due to iron oxides (Wang et al., 2013). Two peaks in the first derivative value of the spectra at 430 and 560 nm were determined to be goethite and hematite, respectively. When the iron content reaches a threshold, the iron oxides have little or no impact on the reflectance spectra (Wu et al., 2016a). The fine fraction of glacier dust has a greater abundance of iron (2016b), and the first derivative values of hematite are higher than goethite, indicating that hematite might be concentrated in the fine fraction (2016a).

Dust optical properties depend strongly on source material and these properties are designed to represent "global-mean" characteristics as closely as possible (Flanner et al., 2007). The SNICAR model applies the Maxwell-Garnett approximation for combining indices of refractions, assuming a mixture of quartz, limestone, montmorillonite, illite, and hematite. We agree that some dust particles (e.g., those containing a large proportion of strongly-absorbing hematite) can have a larger impact on snow albedo than the dust applied in this work (Aoki et al., 2006; Painter et al., 2007).

We have added related discussion in the section 3.4 as following (Page12, Line25-33):
We also have to pay attention to the fact that dust optical properties depend strongly on source material and their properties are designed to represent "global-mean" characteristics as closely as possible (Flanner et al., 2007). The light absorption by mineral dust in snow is thought to be due to iron oxides (Wang et al., 2013), which are efficient light scattering and absorption materials that can enhance absorption at UV and visible wavelengths (Di Mauro et al., 2015; Moosmüller et al., 2012; Zhang et al., 2015). Goethite and hematite are the most abundant forms of iron oxide in dust and the major light absorbers in the shortwave spectrum in snow (Wu et al., 2016). The SNICAR model applies the Maxwell-Garnett approximation for combining indices of refractions, assuming a mixture of quartz, limestone, montmorillonite, illite, and hematite. We should note that some dust particles (e.g., those containing a large proportion of strongly absorbing hematite) can have a larger impact on snow albedo than the dust applied in this work (Aoki et al., 2006; Painter et al., 2007).

Page 5: Line 6: what is the filter blank for? is this a blank filter? It seems there is some particles on filter blank as well since it weighs more than 0 C/cm-2.

Answer: We used the filter blank to detect the sampling and storage processes. It is a blank filter. The blank filters were brought to the field and stored with the filters together. The results showed that the filter blank is much less (1.23±0.38 µg C cm$^{-2}$) than those in the sampled filters (more than 10 µg C cm$^{-2}$), indicating the OC/BC data were reasonable.

Line 8: "separately analyzed" ! analyzed separately

Answer: Has been changed.( (Page6, Line2)

Line 18: ": : : is or not influenced by BC emissions" ! is influenced by BC emissions or not

Answer: Has been changed.( (Page6, Line21)

Line 23: what is "down-sun"?

Answer: It's type wrong. Spectral albedo measurements were collected using an ASD FieldSpec 4 (FS4) standard-resolution spectroradiometer in the range of 350–2500 nm using the ASD Remote Cosine Receptor (RCR) foreoptic. The RCR foreoptic was mounted to the end of a 91.4-cm rectangular aluminium boom and levelled at a height of 80 cm above the snow surface using a bubble level located at the observer's end of the boom, collecting sky irradiance when pointed upward and snow irradiance when pointed downward. Spectral albedo was obtained by dividing snow irradiance by sky irradiance from ten consecutive upward and downward measurements collected at site around local solar noon determined using the NOAA SolarNoonCalculator.

We have changed in the main text (Page7, Line2-9).

For the selected sites, a general-purpose spectroradiometer (Analytical Spectral Devices (ASD), FieldSpec 4, Inc.), was used to measure the reflectance of snow cover. The ASD FieldSpec 4 instrument is a general-purpose spectroradiometer that is useful for applications requiring the measurement of reflectance, transmittance, radiance, or irradiance. The instrument is specifically designed for field environment remote sensing to acquire visible and near-infrared and shortwave infrared spectra. It has 3 nm spectral resolution on the visible/near infrared detector (350–1050 nm, silicon photodiode array), and 10–12 nm resolution on the shortwave infrared detectors (900–2 500 nm, InGaAs). In the field, reflectance was measured at two sites (24K and MD in Table S1) with FieldSpec 4 under clear-sky conditions. These measurements of reflectance were calculated using the standard solar irradiance to get the albedos, which were then used for comparison with simulated albedos.

Line 24: "when the weather was clear"! when the sky was clear. It seems the albedo measurement were made for clear sky only, but later in the albedo comparison, the measured albedo is compared

against the albedo calculated using SNICAR for all-sky case. Is this a typo or this is wrong?

Answer: Has been changed.

The "all-sky" is a type wrong (see Table S2 in supplementary material), we have changed in the main text.

Line 28: "e.g. Doherty et al., 2010": Doherty et al 2010 did not use SNICAR, it only reported the observation in the Arctic. Please remove this citation.

Answer: Has removed.

Page 6: Line 9-11: Please revise this sentence. Is this a model or is this just the method you used to calculate snow water melt in this work?

Answer: It is a method we used to calculate snow water melt. We have changed the sentence in the main text (Page7, Line31-32).

To estimate snow melt due to BC and dust, a method was constructed in which the absorptivity of the snow was multiplied by the daily average incoming shortwave radiation from the automatic weather stations (AWS) set up at the meteorological stations near the sampling sites (Fig. S3) (Schmale et al., 2017).

Line 16: "for clean snow: : :" what is cause of albedo reduction for clean snow case? A different snow grain size due to snow aging? Please be more accurate here.

Answer: Yes. For clean snow, different snow grain size played an important role on the albedo reduction. We have revised the sentence in the main text (Page8, Line7-8):

$\Delta\alpha$ is the albedo reduction caused by BC and dust, for clean snow mainly by different snow grain sizes due to snow aging;

Line 18: "assumed snow depth" what is the assumed snow depth? How did you assume it?

Answer: In the following analysis, we assumed the snow depth to be consistent (which is not the reality) for estimating the variations of snow cover durations days in section 3.3. This assumption was based on the snow depth observations from the TP in previous studies (Che et al., 2012; Xu et al., 2017; Zhong et al., 2016).

We have changed this sentence in the main text (Page8, Line8).

statement is too vague. Do you mean the total emission is larger in southern Asia? or the BC deposition rate is larger ? or both? This is a major conclusion you have in this work, please be more specific.

Answer: Here we mean the total emission is larger in southern Asia (see from the AOD in Fig. R3) rather than the BC deposition rate. We have changed this sentence in the main text (Page9, Line28-33):

In the southern TP (MYL), the amount of BC deposition is larger than in the northern TP (NETP) due to the nearby BC sources in south Asia, known to be a regional hotspot of BC-induced atmospheric solar heating (Ramanathan and Carmichael, 2008). Whereas in the central TP (NMC), the amount of BC deposition was higher than in the NETP. This may be because pollutants from the southern side of the Himalayas can traverse the high mountain range not only through the major north-south river valleys but also by being lifted and advected over the Himalayas to reach to the inland TP (Cong et al., 2015b; Lüthi et al., 2015).

[Figure]

Figure R3. MODIS atmospheric optical depth (AOD) fields derived using dark target algorithms over the TP and its surroundings. (http://giovanni.sci.gsfc.nasa.gov).

Line 8: You used SNICAR model to calculate snow albedo for all-sky cases, but the model set up is not stated. For example, what is the cloud fraction for all sky case? As mentioned in the previous comments, the albedo measurement was performed under clear sky, so in what accuracy do you expect this to agree with the SNICAR calculation (Figure 5)?

Answer: Here is not the all-sky cases, it should be clear sky. We have changed in the main text.

Line 14: "The deviation between measured and simulated reflectance by MD may be a result of the upper boundary of the SNICAR model in particle dimension". This does not make sense, is the deviation due to upper boundary condition (and what upper boundary condition)? or MD particle size? Please revise.

Answer: The difference between measured and simulated reflectance by MD may be caused by the upper boundary of particle size (as shown in Fig. R1, the dust concentrations) in the SNICAR model (Flanner et al., 2007).

Thus, we have changed this sentence in the main text (Page10, Lin10-11).

Line 18-19: "This result is import, showing that the SNICAR model simulations can represent albedo changes of snow cover in the Third Pole region". This is a really strong conclusion. I don't think the authors can make this conclusion based on the results shown in this paper. Especially it is unclear to the readers that how did the authors set up the SNICAR calculation. Please remove this or include more details about SNICAR calculation.

Answer: Agree, and removed this sentence.

Line 26. As the author said, the BC snow albedo forcing over TP is highly uncertain partly due to the uncertainty in simulated BC concentration. But it is also important to point out that a large fraction of such uncertainty is resulted form uncertainty in simulated snow-cover fraction/snow depth. Please consider including this in the discussion.

Answer: Agree, we have added this section in the discussion (Page13, Line1-10).

It is further important to note that a large fraction of such uncertainty can also result from uncertainty in the simulated snow-cover fraction/snow depth across the plateau. The TP covers a large area (more than 2.5 million $km^2$) with an average elevation exceeding 4000 m a.s.l. (Yao et al., 2012a). Considerable heterogeneity in the topography and climate has led to complex spatial and temporal snow cover patterns (Xu et al., 2017). Most studies on the snow cover distribution were based on satellite-based observations (Che et al., 2008, 2012). Surface observations of snow depth showed an increase over the TP from 1957–1998 (Ma and Qin, 2012). However, snow cover depth and the number of snow covered days during the current decade under intense climate warming showed a decreasing trend mainly occurring in the southeast TP in winter. Whereas in spring, snow cover depth showed an increasing trend in the eastern TP (Xu et al., 2017). These differences can also affect the estimation in this study. Nevertheless, the method provides a theoretical approach for evaluating how the presence of LAIs affects the lower parts of the glacier subjected to summer melt.

We have added these information in the supplementary material as Table S3.

[Figure]

Figure R4. Automatic weather stations selected in the Tibetan Plateau.

Table R1. Monthly short-wave radiation from automatic weather stations near the snow sampling sites during the snow melting season.

| Sites | Month of the year | Monthly Short-wave radiation (W m$^{-2}$) | Month of the year | Monthly Shortwave radiation (W m$^{-2}$) |
|---|---|---|---|---|
| Southeastern Tibetan Plateau | | | | |
| | March in 2014 | 248 | March in 2015 | 237 |
| | April in 2014 | 288 | April in 2015 | 259 |
| | May in 2014 | 305 | May in 2015 | 280 |
| | June in 2014 | 265 | June in 2015 | 250 |
| Namco | | | | |
| | | | March in 2015 | 264 |
| | | | April in 2015 | 276 |
| | | | May in 2015 | 306 |
| | | | June in 2015 | 314 |
| Tanggula | | | | |
| | March in 2014 | 210 | | |
| | April in2014 | 226 | | |
| | May in 2014 | 245 | | |
| | June in 2014 | 271 | | |
| Laohugou | | | | |
| | March in 2014 | 229 | March in 2015 | 238 |
| | April in2014 | 269 | April in 2015 | 294 |
| | May in 2014 | 311 | May in 2015 | 305 |
| | June in 2014 | 258 | June in 2015 | 269 |

Line 27: "SD plays an import role" ! SD plays an important role? What do you mean by SD plays a role here? Deeper snow is supposed to melt slower given the same amount of radiative forcing; it is not because snow depths play a role here. Please clarify this point.

Answer: We agree that deeper snow is supposed to melt slower given the same amount of radiative forcing. In this study, we calculated the changes of snow cover duration days rather than the total days of snow cover duration. We mean that SD plays an important role on our estimation of numbers of snow cover duration days. Thus, we have changed the sentence in the main text as following (Page11, Line28-29).

The result indicated that estimation of changes of snow cover duration days caused by the same level of LAIs was also affected by different SD.

Page 10: Line 13: "However, the results presented in this study : : :. for which these assumptions are not critical": This is not true. All the quantities listed in this paragraph will influence the snow albedo and most of them will influence the albedo reduction induced by BC. For example: BC-snow internal mixing increases the albedo forcing by 40-60% compared with external mixing (He et al. (2014). The author should discuss the uncertainty of this study resulted from the assumptions they made, instead of claim these quantities will not impact their results.

References: He, C., Q. Li, K. N. Liou, Y. Takano, Y. Gu, L. Qi, Y. Mao, and L. R. Leung, 2014: Black Carbon Radiative Forcing over the Tibetan Plateau. Geophy. Res. Lett., 41, 7806-7813, doi: 10.1002/2014GL062191.

Answer: Agree, and we have added related discussion in this section in the main text as suggested.

Line 32: "Our study confirms that : : :.. and further reduces snow albedo,: : :.": this is not true. BC and other LAI can reduce the snow albedo even if the snow aging process is not accelerated. Please revise this.

Answer: Agree, and have changed (Page13, Line26).

Our study confirms that BC and other water-insoluble LAIs in snow on land and ice can darken the surface, and further reduces snow albedo, and increases the speed of snow cover melt

Figures3: Is the color bar showing height? Please define the color bar and unit.

Answer: Yes, the color bar shows the height. We have added the information in the figure caption.

**Figure 4: Footprint analyses for six selected sites over the Tibetan Plateau during the winter season (Nov 2015-Feb 2016). LHG and NETP in the northern Tibetan Plateau (Region III), TGL and NMC in the central Tibetan Plateau (Region II), and MYL and SETP in the southern Tibetan Plateau (Region I). (The right color bar means the height (m a.s.l.). The trajectories starting below 500 hPa were taken into account. Black dots: the air parcel did not pass near a fire during the 96 h prior to arrival at the studied site; Green dots: the air parcel did pass by a fire between -96 h and -48 h, but not afterwards, i.e., the 'contact' to a fire lies back at least 48 h before the air parcel arrived at Renlongba glacier; Magenta dots: contact with fires occurred between -48 h and 0 h before arriving at site; Red dots: contact with a fire before and after -48 h.)**

Figure 5: Are MA1-4 measured albedo? Are the dashed lines albedo calculated using SNICAR? Please clarify these details and modify the corresponding text.

Answer: In this figure, MA1-4 (solid lines) are measured albedo. Dashed lines mean the albedo calculated using SNICAR. We have added the information in the figure caption.

**Figure 6:** **Measured albedo by ASD (solid lines, MA1-4) and simulated effects of BC and dust on albedo (dashed lines) at the selected snow site on the Tibetan Plateau.**

Figure 6-8: please clarify the figure convention in each figure. Do the boxes represent average values from central estimate? For example, in Figure 6, you say the rectangles are central estimate, so what does the box mean? standard deviation of central estimate? maximum and minimum of central estimate?

Answer: We have added related information and legend for each figure in the main text.

[revised manuscript text omitted]

18 **Including:**

19 **Supplementary Tables S1-S3**
20 **Supplementary Figure S1-S5**

**Supplementary Tables**

**Table S1.** Sampling information for snow cover over the Tibetan Plateau.

**Table S2.** Parameters for sensitivity analysis with SNICAR model for snow cover in the Tibetan Plateau. 1-Incident radiation (a. Direct, b. Diffuse); 2- Solar zenith angle; 3- Surface spectral distribution (a. Mid-latitude winter, clear-sky, cloud amount<5. b. Mid-latitude winter, cloudy, cloud amount $\geq$5); 4- Snow grain effective radius (µm); 5- Snowpack thickness (m); 6- Snowpack density (kg m$^{-3}$); 7- Albedo of underlying ground (a. Visible, 0.3–0.7µm, b. Near-infrared, 0.7–5.0µm); 8-MAC scaling factor (experimental) for BC; 9-BC concentration (ppb, sulfate-coated); 10- Dust concentration (µg g$^{-1}$, 5.0–10.0 µm diameter); 11- Volcanic ash concentration (ppm); 12- Experimental particle 1 concentration (ng g$^{-1}$).

**Table S3.** Short-wave radiation input data at selected sites across the Tibetan Plateau close to the snow sampling sites.

 **Table S1.** Sampling information for snow cover over the Tibetan Plateau regions.

| Sites | Time | Lat. (°) | Long. (°) | Elevation (m) | Snow depth (cm) | Density (kg m⁻³) | Observed snow grain size (µm) | Snow type | Snow temperature (°C) |
|---|---|---|---|---|---|---|---|---|---|
| LHG | 2015-12-29 12:00 | 39.50 | 96.52 | 4250 | 20 | | | | |
| RYS | 2014-12-03 12:25 | 36.50 | 101.52 | 3396 | | | | | |
| TP-S-1511-36 | 2015-11-30 16:48 | 35.44 | 98.65 | 4363 | 11 | 300 | 2000 | Coarse firn with hoar | −6.5 |
| TP-S-1511-37 | 2015-11-30 18:10 | 35.37 | 99.29 | 4297 | 16 | 250 | 2000 | Coarse firn with wind crust and dirty layer | −7.0 |
| TP-S-1511-38 | 2015-12-01 10:09 | 35.39 | 99.31 | 4298 | 18 | 225 | 2000 | | −15.0 |
| TP-S-1511-39 | 2015-12-01 11:23 | 35.51 | 99.51 | 4405 | 14 | 225 | 2000 | Coarse firn | −6.5 |
| TP-S-1511-35 | 2015-11-30 13:25 | 34.11 | 97.65 | 4781 | 10 | 325 | 2000 | Coarse firn with dirty layer | −5.0 |
| MD | 2014-12-03 12:10 | 34.86 | 98.17 | 4220 | | | | | |
| TP-S-1511-33 | 2015-11-30 11:12 | 33.47 | 97.24 | 4259 | 7 | 225 | 1000 | Coarse firn | −4 |
| TP-S-1511-34 | 2015-11-30 11:41 | 33.61 | 97.24 | 4366 | 12 | 350 | 2000 | Coarse firn with wind crust | −6.0 |
| TGL | 2014-12-10 12:40 | 32.90 | 91.92 | 5163 | | | | | |
| LWQ | 2014-12-04 12:20 | 31.22 | 96.57 | 4370 | | | | | |
| 24K | 2014-12-06 12:47 | 30.99 | 91.68 | 5100 | | | | | |
| NMC | 2014-12-09 12:30 | 30.75 | 91.68 | 4729 | | | | | |
| MYL | 2014-12-10 12:30 | 30.11 | 83.39 | 4610 | | | | | |
| TP-S-1511-1 | 2015-11-15 11:30 | 30.45 | 91.05 | 4216 | 3 | | 200 | Firn | −5.0 |
| TP-S-1511-2 | 2015-11-15 13:33 | 30.67 | 91.10 | 5036 | 6.8 | | 1000 | Firn | −3.7 |
| TP-S-1511-3 | 2015-11-15 14:00 | 30.68 | 91.10 | 5124 | 9 | | 1000 | Fine firn | −0.14 |
| TP-S-1511-4 | 2015-11-15 16:44 | 30.79 | 90.96 | 4687 | 5 | | 1000 | Fine firn | −2 |
| TP-S-1511-5 | 2015-11-15 17:30 | 30.68 | 91.10 | 5096 | 17.1 | | 500 | Firn with wind crust | −4.8 |
| TP-S-1511-6 | 2015-11-16 13:52 | 29.19 | 90.62 | 4762 | 6.8 | | 1000 | Firn | −3.7 |
| TP-S-1511-16 | 2015-11-20 09:13 | 29.00 | 87.47 | 4798 | 9 | 300 | 1000 | Fine firn | −17 |
| TP-S-1511-27 | 2015-11-26 12:37 | 29.83 | 92.34 | 4971 | 17 | 150 | 3500 | Coarse firn | −5 |
| TP-S-1511-28 | 2015-11-27 10:55 | 29.63 | 94.63 | 4304 | 12 | 200 | 2000 | | −1.5 |
| TP-S-1511-29 | 2015-11-27 11:25 | 29.61 | 94.67 | 4523 | 24 | 200 | 3000 | Hoar at the bottom | −4.5 |
| TP-S-1511-30 | 2015-11-27 17:48 | 29.77 | 95.70 | 3728 | 11 | 400 | 2000 | | −1.5 |
| TP-S-1511-31 | 2015-11-28 12:10 | 29.26 | 96.94 | 4582 | 12 | 250 | 2000 | Coarse firn, ice at bottom | −0.05 |
| TP-S-1511-32 | 2015-11-28 1:45 | 29.32 | 97.03 | 4745 | 20 | 175 | 2000 | Fresh snow and firn | −1.0 |
| TP-S-1511-9 | 2015-11-18 12:13 | 28.82 | 87.93 | 4851 | 12.5 | 230 | 1500 | Firn with wind crust and dirty layer | −12.6 |
| TP-S-1511-17 | 2015-11-20 10:02 | 28.96 | 87.44 | 5163 | 15 | 270 | 2000 | Firn with wind crust and dirty layer | −17 |
| TP-S-1511-18 | 2015-11-20 10:35 | 28.91 | 87.43 | 5042 | 8 | 250 | 3000 | Firn with wind crust and dirty layer | −15 |
| TP-S-1511-21 | 2015-11-21 13:13 | 28.67 | 86.13 | 4565 | 9 | 200 | 2000 | Coarse firn | −2.0 |
| TP-S-1511-22 | 2015-11-21 14:09 | 28.52 | 86.17 | 5096 | 5 | 200 | 2000 | Coarse firn | 0 |
| TP-S-1511-23 | 2015-11-22 13:12 | 28.65 | 86.08 | 4592 | 9 | 325 | 3000 | Coarse firn with wind crust and hoar | |
| TP-S-1511-24 | 2015-11-22 14:11 | 28.76 | 85.66 | 4614 | 14 | 250 | 3000 | Coarse firn | 0.05 |
| TP-S-1511-25 | 2015-11-22 15:07 | 28.90 | 85.40 | 4861 | 11.5 | 225 | 3000 | Coarse firn | −2.5 |
| TP-S-1511-26 | 2015-11-22 15:55 | 28.93 | 85.40 | 5166 | 12 | 275 | 1000 | Fine firn with wind crust | 0 |
| TP-S-1511-8 | 2015-11-17 13:02 | 27.39 | 88.83 | 4150 | 5 | 275 | 1000 | Fine firn | −0.4 |

34 **Table S2.** Parameters for sensitivity analysis with SNICAR model for snow cover in the Tibetan
35 Plateau.

| Site | 1 | 2 | 3 | 4 | 5 | 6 | 7a | 7b | 8 | 9-BC | 10-Dust | 11 | 12 |
|---|---|---|---|---|---|---|---|---|---|---|---|---|---|
| LHG* | direct | 62.75 | a | | 20 | | 0.15 | 0.3 | 1 | 3745.46 | 224.49 | 0 | 0 |
| RYS | direct | 58.58 | a | | 11 | | 0.15 | 0.3 | 1 | 8922.53 | 417.65 | 0 | 0 |
| TP-S-1511-36 | direct | 57.02 | a | | 11 | | 0.15 | 0.3 | 1 | 3097.59 | 172.04 | 0 | 0 |
| TP-S-1511-37 | direct | 56.95 | a | | 16 | | 0.15 | 0.3 | 1 | 7373.45 | 292.97 | 0 | 0 |
| TP-S-1511-38 | direct | 57.13 | a | | 18 | | 0.15 | 0.3 | 1 | 4122.43 | 233.86 | 0 | 0 |
| TP-S-1511-39 | direct | 57.25 | a | | 14 | | 0.15 | 0.3 | 1 | 2894.96 | 157.78 | 0 | 0 |
| TP-S-1511-35 | direct | 55.69 | a | | 10 | | 0.15 | 0.3 | 1 | 4217.19 | 295.80 | 0 | 0 |
| MD | direct | 56.94 | a | | 11 | | 0.15 | 0.3 | 1 | 3122.68 | 231.03 | 0 | 0 |
| TP-S-1511-33 | direct | 55.05 | a | | 7 | | 0.15 | 0.3 | 1 | 17468.24 | 846.39 | 0 | 0 |
| TP-S-1511-34 | direct | 55.19 | a | | 12 | | 0.15 | 0.3 | 1 | 4009.22 | 211.41 | 0 | 0 |
| TGL | direct | 55.8 | a | | 11 | | 0.15 | 0.3 | 1 | 1014.18 | 89.61 | 0 | 0 |
| LWQ | direct | 53.44 | a | | 11 | | 0.15 | 0.3 | 1 | 1734.68 | 71.11 | 0 | 0 |
| 24K | direct | 53.46 | a | | 11 | | 0.15 | 0.3 | 1 | 876.75 | 113.52 | 0 | 0 |
| NMC | direct | 53.55 | a | | 11 | | 0.15 | 0.3 | 1 | 600.68 | 45.91 | 0 | 0 |
| MYL | direct | 53.01 | a | | 11 | | 0.15 | 0.3 | 1 | 319.89 | 23.92 | 0 | 0 |
| TP-S-1511-1 | direct | 48.83 | a | | 3 | | 0.15 | 0.3 | 1 | 1283.33 | 125.53 | 0 | 0 |
| TP-S-1511-2 | direct | 49.05 | a | | 6.8 | | 0.15 | 0.3 | 1 | 468.71 | 54.14 | 0 | 0 |
| TP-S-1511-3 | direct | 49.06 | a | | 9 | | 0.15 | 0.3 | 1 | 5093.10 | 352.58 | 0 | 0 |
| TP-S-1511-4 | direct | 49.17 | a | | 5 | | 0.15 | 0.3 | 1 | 4192.93 | 213.11 | 0 | 0 |
| TP-S-1511-5 | direct | 49.06 | a | | 17.1 | | 0.15 | 0.3 | 1 | 1084.93 | 109.12 | 0 | 0 |
| TP-S-1511-6 | direct | 47.83 | a | | 6.8 | | 0.15 | 0.3 | 1 | 250.69 | 42.80 | 0 | 0 |
| TP-S-1511-16 | direct | 48.6 | a | | 9 | | 0.15 | 0.3 | 1 | 886.69 | 86.78 | 0 | 0 |
| TP-S-1511-27 | direct | 50.69 | a | | 17 | | 0.15 | 0.3 | 1 | 3586.99 | 323.06 | 0 | 0 |
| TP-S-1511-28 | direct | 50.68 | a | | 12 | | 0.15 | 0.3 | 1 | 952.23 | 55.50 | 0 | 0 |
| TP-S-1511-29 | direct | 50.66 | a | | 24 | | 0.15 | 0.3 | 1 | 255.93 | 26.86 | 0 | 0 |
| TP-S-1511-30 | direct | 50.82 | a | | 11 | | 0.15 | 0.3 | 1 | 1343.22 | 127.57 | 0 | 0 |
| TP-S-1511-31 | direct | 50.49 | a | | 12 | | 0.15 | 0.3 | 1 | 2117.49 | 238.03 | 0 | 0 |
| TP-S-1511-32 | direct | 50.55 | a | | 20 | | 0.15 | 0.3 | 1 | 201.70 | 21.58 | 0 | 0 |
| TP-S-1511-9 | direct | 47.95 | a | | 12.5 | | 0.15 | 0.3 | 1 | 3417.06 | 344.93 | 0 | 0 |
| TP-S-1511-17 | direct | 48.56 | a | | 15 | | 0.15 | 0.3 | 1 | 822.01 | 92.20 | 0 | 0 |
| TP-S-1511-18 | direct | 48.51 | a | | 8 | | 0.15 | 0.3 | 1 | 1277.70 | 101.71 | 0 | 0 |
| TP-S-1511-21 | direct | 48.5 | a | | 9 | | 0.15 | 0.3 | 1 | 1079.43 | 101.39 | 0 | 0 |
| TP-S-1511-22 | direct | 48.35 | a | | 5 | | 0.15 | 0.3 | 1 | 2201.12 | 226.49 | 0 | 0 |
| TP-S-1511-23 | direct | 48.7 | a | | 9 | | 0.15 | 0.3 | 1 | 1072.04 | 104.53 | 0 | 0 |
| TP-S-1511-24 | direct | 48.81 | a | | 14 | | 0.15 | 0.3 | 1 | 1042.25 | 109.11 | 0 | 0 |
| TP-S-1511-25 | direct | 48.95 | a | | 11.5 | | 0.15 | 0.3 | 1 | 931.68 | 35.29 | 0 | 0 |
| TP-S-1511-26 | direct | 48.98 | a | | 12 | | 0.15 | 0.3 | 1 | 759.41 | 65.26 | 0 | 0 |
| TP-S-1511-8 | direct | 46.28 | a | | 5 | | 0.15 | 0.3 | 1 | 574.95 | 22.39 | 0 | 0 |

37 **Table S3.** Short-wave radiation input data at selected sites across the Tibetan Plateau close to the
38 snow sampling sites.

| | SW (W m$^{-2}$) | | SW (W m$^{-2}$) |
|---|---|---|---|
| SETP | | | |
| March, 2014 | 248 | March, 2015 | 237 |
| April, 2014 | 288 | April, 2015 | 259 |
| May, 2014 | 305 | May, 2015 | 280 |
| June, 2014 | 265 | June, 2015 | 250 |
| NMC | | | |
| | | March, 2015 | 264 |
| | | April, 2015 | 276 |
| | | May, 2015 | 306 |
| | | June, 2015 | 314 |
| TGL | | | |
| March, 2014 | 210 | | |
| April, 2014 | 226 | | |
| May, 2014 | 245 | | |
| June, 2014 | 271 | | |
| LHG | | | |
| March, 2014 | 229 | March, 2015 | 238 |
| April, 2014 | 269 | April, 2015 | 294 |
| May, 2014 | 311 | May, 2015 | 305 |
| June, 2014 | 258 | June, 2015 | 269 |

**Supplementary Figures**

**Figure S1.** Example of snow sampling sites over the Tibetan Plateau.

**Figure S2.** Snow grain size observation in the field (a) and measurement in the lab (b).

**Figure S3.** BC emissions arrived at the selected sites over the Tibetan Plateau based on data drawn from http://inventory.pku.edu.cn/download/download.html (Wang et al., 2014). (A value 2 on the x axis means $2\times10^6$ g month$^{-1}$. The BC value is averaged over 48 h along the back trajectory path. And the value on the y axis is the number of trajectories in the respective BC bin. The bin width is 0.05, i.e. from 0 to 5 on the horizontal axis there are 100 bins.)

**Figure S3.** Automatic weather stations selected for the short-wave radiation input data across the Tibetan Plateau.

**Figure S4.** BC emissions arrived at the selected sites over the Tibetan Plateau region based on data drawn from http://inventory.pku.edu.cn/download/download.html (Wang et al., 2014). (A value 2 on the x axis means $2\times10^6$ g month$^{-1}$. The BC value is averaged over 48 h along the back trajectory path. And the value on the y axis is the number of trajectories in the respective BC bin. The bin width is 0.05, i.e. from 0 to 5 on the horizontal axis there are 100 bins.)

**Figure S5.** Distributions of changes of snow cover duration days caused by BC and mineral dust.

59    **Fig S1.**

[Figure]

62    **Figure S1.** Example of snow sampling sites over the Tibetan Plateau.

63 **Fig S2.**

[Figure]

64
65 **Figure S2.** Snow grain size observation in the field (a) and measurement in the lab (b).

66 **Fig S3.**

[Figure]

67
68 Figure S3. Automatic weather stations selected in the Tibetan Plateau.

**Fig S4.**

[Figure]

**Figure S4.** BC emissions arrived at the selected sites over the Tibetan Plateau region based on data drawn from http://inventory.pku.edu.cn/download/download.html (Wang et al., 2014). (A value 2 on the x axis means $2 \times 10^6$ g month$^{-1}$. The BC value is averaged over 48 h along the back trajectory path. And the value on the y axis is the number of trajectories in the respective BC bin. The bin width is 0.05, i.e. from 0 to 5 on the horizontal axis there are 100 bins.)

76 **Fig S5.**

[Figure]

77

**Figure S5.** Distributions of changes of snow cover duration days caused by BC and mineral dust.

79

80

81

---

## Author Comment (AC2) · 11 Oct 2017

Oct10, 2017

Dear editor,

The manuscript: Black carbon and mineral dust in snow cover on the Tibetan Plateau by Zhang Y.L. et al.

The manuscript number: tc-2017-111

We greatly appreciate the reviewers' constructive comments to improve the paper. We have revised our manuscript according to these comments (blue color in the main text), and hope the revised manuscript is suitable for publication in The Cryosphere.

The "point to point" response to comments are listed as below.

Sincerely yours, Yulan Zhang and Shichang Kang

Response to Referee #2 This manuscript presents results from snow sampling, laboratory analysis, and related modeling efforts for the presence and impact of light absorbing impurities in the Tibetan Plateau region. Although the manuscript is not wrong in stating more measurements are valuable in constraining our understanding of the spatial and temporal variability of light absorbing particulates, the techniques presented in this paper are unclear and as written does not contribute to the state of knowledge of light absorbing particulates in snow. Major changes are needed before this paper can be reviewed again for publication. I will not, at the point, do line by line corrections because they are numerous. The authors need to revisit the writing in each section for editing and to clarify there justification, methods, and results. Particularly, snow sampling and automatic weather stations need to be described in significantly more detail. Answer: Thank you very much for the comments and suggestions. We have tried to improve the methods and results in the main text (in blue color). Snow sampling and automatic weather stations are also described in detailed (Section2.2, Section3.3, in blue color).

Comment 1: My first and foremost concern is that samples were collected in November and December, and yet they are attempting to quantify the impacts of light absorbing particulates on melt. This does not make sense to me and if I misunderstood this it is because it is not made clear in the manuscript. Although the sample collection timing may be after the summer monsoon, these samples do not represent the impurities that are present during the ablation season- and therefore it is inappropriate to use these values to quantify reduction in snowpack duration. Particularly for dust, which tends to deposit in the spring when source regions dry out (peak radiative forcing by dust in snow is observed from MODIS imagery over the Himalayan region in April and May). Answer: Agree, these samples were collected during winter season when the snow cover were more stable and continuous, which might not represent the true impurities

as these during the ablation season. However, during melting season, because of the poor accessibility on the snowpack in the TP, it was hardly to collect the snow cover samples. Besides, considerable heterogeneity in the topography and climate has led to complex spatial and temporal snow cover patterns (Xu et al., 2017). Discontinued snow cover during melting season may be a problem to represent the true impurities in snow. Thus in our study, we gave an estimation based on the different snow grain size and density, rather than the fixed data. We have to admit there existed uncertainties at present. In the future, we should do more related works to fill in gaps.

Comment 2: The backtrajectory footprint modeling was also very unclear to me, how the model runs were carried out needs to be described better, but it is unclear to me why the model runs were only completed for the winter. And were they run continuously? Typically particulates are deposited in episodic events so running it continuously does not inform the source region, it just informs of the regional synoptics. I suggest seeing Skiles et al., 2015 (Hydrological Processes) for how that study produced and described backtrajectory footprints. Answer: We agree. Since most of the snow cover will not exist during summer season in the TP, thus we only calculated the model runs in winter when the snow cover accumulated. The calculation is continuous in the winter season since we did not know the exact snow events during the periods. It is an estimation of all air mass which can be transported to the sampling sites. We have learned the recommended reference (Skiles et al., 2015), and tried to improve the descriptions in the main text as following (Page6, Line6-34). Calculation of air parcel trajectories is a widely used approach in different areas of atmospheric research. Conceptually, footprint analysis was considered as changes in concentration at the receptor site that can be attributed to different upwind source areas along the backward trajectories (Skiles et al., 2015). To determine the potential origins of the LAIs deposited on the snow cover of the TP, back trajectory analyses were performed using the European Centre for Medium-Range Weather Forecasts (ECMWF) analysis fields with the Lagrangian analysis tool LAGRANTO (Sprenger and Wernli, 2015), launched every six hours for six selected sampling sites as receptors (including three sites of MYL, NMC, and SETP

in region I, two sites of TGL and NETP in region II, and LHG in region III) during the winter season. The ECMWF fields (horizontal and vertical wind components) were retrieved on 137 model levels and then interpolated onto a $0.25° \times 0.25°$ latitude-longitude grid. Trajectory starting positions can be defined easily and flexibly based on different geometrical and meteorological conditions; after the computation of the trajectories, a versatile selection is offered based on single or combined criteria (Sprenger and Wernli, 2015). First, starting positions are initialized with a suitable domain over the TP at 12:00 UTC on November 1, 2015 (or 2014). For this case, a domain from 0 to 75 and 60 °E to 120 °E was chosen. We choose starting positions in this domain that are horizontally equidistant with 80 km horizontal spacing and extend vertically from 1030 to 790 hPa with 30 hPa vertical spacing. Then, the trajectories are calculated from all starting positions 96 h backward in time. Finally, biomass burning emission data is traced along the calculated trajectories to estimate whether an air parcel at the receptor site is influenced by BC fire emissions or not. In this study, the Fire INventory from NCAR (FINN) v1.5 global fire emissions flux in 2012−2014, speciated with the GEOS-chem mechanism, was used to estimate contributions of BC fire emission at the six selected receptor sites. FINN emission estimates are based on the framework described by Wiedinmyer et al. (2011). FINN used the satellite observations of active fires and land cover, together with emission factors and estimated fuel loadings to provide daily, highly resolved (1 km) open biomass burning emissions estimates for use in regional and global chemical transport models. Then, the same calculation was performed for the Eclipse V5 inventory for the anthropogenic contributions of BC. The Eclipse V5 inventory was widely used in the simulations. The historical data for the period 1990−2010 were revised compared to preceding sets using the latest IAE (the International Energy Agency) and FAO (the Food and Agriculture Organization) statistics extending to 2010, as well as recent country reporting where available. Note that this analysis was not exactly quantitative and did not take into account wet and dry deposition. We also assumed that the anthropogenic emissions have not changed significantly from 2010 compared to the period from 2012−2014. Thus, the comparison of

natural and anthropogenic BC contributions was reasonable. The relative differences between the BC contributions can provide information on regional differences in this study. References: Skiles, S. M., Painter, T. H., belnap, J., Holland, L., Reynolds, R. L., Goldstein, H. L., Lin, J.: Regional variability in dust-on-snow processes and impacts in the Upper Colorado River Basin, Hydrol. Process., doi:10.1002/hyp.10569, 2015.

Comment 3: The albedo measurements need to be better described. And how was snow effective grain radius retrieved? This should be an optically equivalent grain size, not an observable grain size. If an effective/optical grain size was used, the retrieval should be described. If observed grain sizes were used, the large error introduced by this (see Painter et al., 2007 in Journal of Glaciology) needs to be mentioned. Estimates of changes in snow cover duration should be removed or significant more detail and justification needs to be made for the timing of sample collection. Answer: In this study, we used an optically equivalent grain size, not an observable grain size.

In the field, the snow grains were sprayed on the MIG paper. Then we took a photo by using portable digital microscope (Anyty 3R-MSV500) (Fig. R1a). In the lab, we can measure the snow grain shape and obtain the lengths of a and b (Fig. R1b). Mugnai and Wiscombe (1987) demonstrated that a collection of unoriented non-spheroids produce the same scattering results as spheres, and Grenfell and Warren (1999) showed that the radius of a non-spherical particle was equal to that of a spherical particle that has the same volume-to-surface-area (V/A) ratio. Consequently, V/A ratio was used to transfer the measured snow grain size into the effective grain size. On this basis of field measurement, Hao (2009) proposed two assumptions: 1) The snow particle is an inequilateral spheroid; and 2) The major axis, minor axis, and height of the inequilateral spheroid is denoted by a, b, and c, respectively, in such a way that the relationship $a = 2b$ exists. According to Kokhanovsky and Zege (2004), the effective snow grain radius (Ref) can be calculated equal to the radius of the volume-to-surface equivalent sphere by the following equation (1): $R\_ef=(3V ÌĚ)/(4A ÌĚ )$ (1) Where, $V ÌĚ=4/3 \pi rˆ3$ and $A ÌĚ=\pi rˆ2$, are the average volume and the average cross-section (geometric

shadow) area of the snow grains, respectively. And, r is the radius of geometric optics, r=(a+b)/4. Thus, R_ef≈0.35a Eq.1 was then employed to calculate the effective snow grain size.

We have added related information in the main text (Page 5, Line2-12): For the snow grain size observation, we sprayed the snow grains on MIG paper and took a photo using a portable digital microscope (Anyty 3R-MSV500) to calculate the major and minor axis (Fig. S2). Based on field measurements from previous studies (Mugnai and Wiscombe, 1980; Grenfell and Warren, 1999), two assumptions were proposed: 1) The snow grain particle is an inequilateral spheroid, and 2) The major axis (a), minor axis (b), and height (c) of the inequilateral spheroid are denoted by a, b, and c, respectively, in such a way that the relationship a=2b exists (Hao, 2009). According to Kokhanovsky and Zege (2004), the effective snow grain radius (Ref) can be calculated as equal to the radius of the volume-to-surface equivalent sphere by the following equation (1): R_ef=(3V Ì Ě)/(4A Ì Ě ) (1) where V Ì Ě=4/3 $\pi r^3$ and A Ì Ě=$\pi r^2$, are the average volume and the average cross-section (geometric shadow) area of the snow grains, respectively. In addition, r is the radius of geometric optics, r=(a+b)/4. Thus R_ef≈0.35a (2)

Figure R1. Snow grain size observation in the field (a) and measurement in the lab (b).

For the estimates of changes in snow cover duration days, we have tried to improve the related details in the main text.

References: Grenfell, T. C., and Warren, S. G.: Representation of a nonspherical ice particle by a collection of independent sphere for scattering and obsorption of radiation, J. geophys. Res., 104(D24), 31697–31709, 1999. Hao, X.: Retrieval of alpine snow cover area and grain size basing on optical remote sensing. PhD thesis, Cold and Arid Regions Environment Engineering Research Institute, Lanzhou, China, pp. 103–104, 2009. Mugnai, A., and Wiscombe, W. J.: Scattering of radiation by moderately nonspherical particles. J. Atmos. Sci., 37, 1291–1307, 1987.

Comment 4: Section 3.3 and 3.4 are generally confusing- was shortwave radiation (uplooking and downlooking pyranometers) actually measured? This analysis seems far too simplified. Furthermore, the discussion of snow depth is misleading. This is a straightforward energy balance calculation, so less snow will melt faster than more snow for an equal amount of forcing- this is basic mass balance. So snow depth does not itself play an important role in the reduction of snow cover by particulates. This is a mixing up of forcing and state functions. The citations are also outdated or incorrect in many cases, and I suggest the authors revisit these. Answer: Shortwave radiation data used in this study was actually measurement by automatic weather station at different selected site near the snow sampling sites in region I, II, and III of the TP (Fig R2). Table R1 showed monthly short-wave radiation from automatic weather stations near the snow sampling sites during the snow melting season (March-May) when the temperature began to increase. On average, shortwave radiations in March, April, May, and June is about 238, 269, 292, and 271 W m–2, respectively. Short-wave radiation in March showed the minimum value in Tanggula (210 W m–2, lower than the average). Short-wave radiation in May showed the maximum value in Namco (314 W m–2, higher the average). Thus, in order to estimate the impact of input short-wave radiation on the snow cover duration, we gave a range of short-wave radiation from 220 to 310 W m–2.

We have added related information in the main text as following (Page11, Line14-23): In Eq. (4), monthly shortwave radiation (SW) input data were obtained from the Automatic Weather Station (AWS) near the snow sampling sites across the TP (Fig. S3). Table S3 shows monthly shortwave radiation data from AWS during the snow melting season (March-May) when the temperature began to increase. On average, shortwave radiation in March, April, May, and June is approximately 238, 269, 292, and 271 W m–2, respectively. Short-wave radiation in March showed the minimum value in Tanggula of the central TP (210 W m–2, lower than the average), whereas in May it showed the maximum value in Namco (314 W m–2, higher the average). Based on these data during the melt season, three scenarios (220, 270, and 310 W m–2)were defined as the minimum, median, and maximum scenarios of input shortwave radiation to estimate its

[Figure]

impact on changes of snow cover duration.

Figure R2. Automatic weather stations selected in the Tibetan Plateau.

Table R1. Monthly short-wave radiation from automatic weather stations near the snow sampling sites during the snow melting season. Sites Month of the year Monthly Short-wave radiation (W m–2) Month of the year Monthly Shortwave radiation (W m–2) South-eastern Tibetan Plateau March in 2014 248 March in 2015 237 April in 2014 288 April in 2015 259 May in 2014 305 May in 2015 280 June in 2014 265 June in 2015 250 Namco March in 2015 264 April in 2015 276 May in 2015 306 June in 2015 314 Tang-gula March in 2014 210 April in2014 226 May in 2014 245 June in 2014 271 Laohugou March in 2014 229 March in 2015 238 April in2014 269 April in 2015 294 May in 2014 311 May in 2015 305 June in 2014 258 June in 2015 269

We agree that the equation for changes of snow cover duration is a straightforward energy balance calculation. Thus in this study, it means the duration days of less snow will be less than the more snow for an equal amount of forcing. So under the same level of LAIs in snow, the reduction of snow cover duration days can be affected by different snow depth. We have tried to revise in the main text (section 3.4).

Comment 5: I take issue with the use of the 'Third Pole' term, this is not universally recognized and is somewhat politicized, why not just use Tibetan Plateau or High Mountain Asia? Also light absorbing impurity is an outdated term, the community has moved toward the use of light absorbing particulates. Also mineral dust and the acronym MD are confusing you can simply say dust- which needs no acronym. Similarly, please be consistent in terminology, for example, albedo and reflectance are not the same thing. Answer: We have changed the term "Third Pole" to be "Tibetan Plateau".

We have used the term "light-absorbing particulates" in the main text.

We have changed the "MD" as "dust" in the main text.

Albedo and reflectance are not the same thing. Albedo is defined as the ratio of ir-

radiance reflected to the irradiance received by a surface. The proportion reflected is not only determined by properties of the surface itself, but also by the spectral and angular distribution of solar radiation reaching the Earth's surface. Unless given for a specific wavelength (spectral albedo), albedo refers to the broadband spectrum of solar radiation.

Reflectance of the surface of a material is its effectiveness in reflecting radiant energy. It is the fraction of incident electromagnetic power that is reflected at an interface. The reflectance spectrum or spectral reflectance curve is the plot of the reflectance as a function of wavelength. When we use ASD, we get the reflectance. In order to compare the albedo simulated by SNICAR model in this study, we have to calculate the reflectance based on the standard solar irradiance. We also carefully use the words consistent in the main text.

Comment 6: Uncertainties in modeling are not only due to lack of observations, and increasing our number of sampling points alone will not reduce our model uncertainty. To state this is misleading. Answer: Agree, and we have tried to revise this section in the section3.4 of the main text.

Please also note the supplement to this comment:
https://www.the-cryosphere-discuss.net/tc-2017-111/tc-2017-111-AC2-supplement.pdf
* * *
[Figure]

Figure R1. Snow grain size observation in the field (a) and measurement in the lab (b).

**Fig. 1.**

Figure·R2.·Automatic·weather·stations·selected·in·the·Tibetan·Plateau.

**Fig. 2.**

Table R1. Monthly short-wave radiation from automatic weather stations near the snow sampling sites during the snow melting season.

| Sites | Month of the year | Monthly Short-wave radiation (W·m⁻²) | Month of the year | Monthly Shortwave radiation (W·m⁻²) |
|---|---|---|---|---|
| Southeastern Tibetan Plateau | | | | |
| | March in 2014 | 248 | March in 2015 | 237 |
| | April in 2014 | 288 | April in 2015 | 259 |
| | May in 2014 | 305 | May in 2015 | 280 |
| | June in 2014 | 265 | June in 2015 | 250 |
| Namco | | | | |
| | | | March in 2015 | 264 |
| | | | April in 2015 | 276 |
| | | | May in 2015 | 306 |
| | | | June in 2015 | 314 |
| Tanggula | | | | |
| | March in 2014 | 210 | | |
| | April in2014 | 226 | | |
| | May in 2014 | 245 | | |
| | June in 2014 | 271 | | |
| Laohugou | | | | |
| | March in 2014 | 229 | March in 2015 | 238 |
| | April in2014 | 269 | April in 2015 | 294 |
| | May in 2014 | 311 | May in 2015 | 305 |
| | June in 2014 | 258 | June in 2015 | 269 |

**Fig. 3.**

**Supplement:**

Oct10, 2017

Dear editor,

The manuscript: ***Black carbon and mineral dust in snow cover on the Tibetan Plateau*** by Zhang Y.L. et al.

The manuscript number: tc-2017-111

We greatly appreciate the reviewers' constructive comments to improve the paper. We have revised our manuscript according to these comments (blue color in the main text), and hope the revised manuscript is suitable for publication in The Cryosphere.

The "point to point" response to comments are listed as below.

Sincerely yours,
Yulan Zhang and Shichang Kang

**Response to Referee #2**

This manuscript presents results from snow sampling, laboratory analysis, and related modeling efforts for the presence and impact of light absorbing impurities in the Tibetan Plateau region. Although the manuscript is not wrong in stating more measurements are valuable in constraining our understanding of the spatial and temporal variability of light absorbing particulates, the techniques presented in this paper are unclear and as written does not contribute to the state of knowledge of light absorbing particulates in snow. Major changes are needed before this paper can be reviewed again for publication. I will not, at the point, do line by line corrections because they are numerous. The authors need to revisit the writing in each section for editing and to clarify there justification, methods, and results. Particularly, snow sampling and automatic weather stations need to be described in significantly more detail.

Answer: Thank you very much for the comments and suggestions. We have tried to improve the methods and results in the main text (in blue color). Snow sampling and automatic weather stations are also described in detailed (Section2.2, Section3.3, in blue color).

**Comment 1:** My first and foremost concern is that samples were collected in November and December, and yet they are attempting to quantify the impacts of light absorbing particulates on melt. This does not make sense to me and if I misunderstood this it is because it is not made clear in the manuscript. Although the sample collection timing may be after the summer monsoon, these samples do not represent the impurities that are present during the ablation season- and therefore it is inappropriate to use these values to quantify reduction in snowpack duration. Particularly for dust, which tends to deposit in the spring when source regions dry out (peak radiative forcing by dust in snow is observed from MODIS imagery over the Himalayan region in April and May).

Answer: Agree, these samples were collected during winter season when the snow cover were more stable and continuous, which might not represent the true impurities as these during the ablation season. However, during melting season, because of the poor accessibility on the snowpack in the TP, it was hardly to collect the snow cover samples. Besides, considerable heterogeneity in the topography and climate has led to complex spatial and temporal snow cover patterns (Xu et al., 2017). Discontinued snow cover during melting season may be a problem to represent the true impurities in snow. Thus in our study, we gave an estimation based on the different snow grain size and density, rather than the fixed data. We have to admit there existed uncertainties at present. In the future, we should do more related works to fill in gaps.

**Comment 2:** The backtrajectory footprint modeling was also very unclear to me, how the model runs were carried out needs to be described better, but it is unclear to me why the model runs were only completed for the winter. And were they run continuously? Typically particulates are deposited in episodic events so running it continuously does not inform the source region, it just informs of the regional synoptics. I suggest seeing Skiles et al., 2015 (Hydrological Processes) for how that study produced and described backtrajectory footprints.

Answer: We agree. Since most of the snow cover will not exist during summer season in the TP, thus we only calculated the model runs in winter when the snow cover accumulated. The calculation is continuous in the winter season since we did not know the exact snow events during the periods. It is an estimation of all air mass which can be transported to the sampling sites.

We have learned the recommended reference (Skiles et al., 2015), and tried to improve the descriptions in the main text as following (Page6, Line6-34).

Calculation of air parcel trajectories is a widely used approach in different areas of atmospheric research. Conceptually, footprint analysis was considered as changes in concentration at the receptor site that can be attributed to different upwind source areas along the backward trajectories (Skiles et al., 2015). To determine the potential origins of the LAIs deposited on the snow cover of the TP, back trajectory analyses were performed using the European Centre for Medium-Range Weather Forecasts (ECMWF) analysis fields with the Lagrangian analysis tool LAGRANTO (Sprenger and Wernli, 2015), launched every six hours for six selected sampling sites as receptors (including three sites of MYL, NMC, and SETP in region I, two sites of TGL and NETP in region II, and LHG in region III) during the winter season. The ECMWF fields (horizontal and vertical wind components) were retrieved on 137 model levels and then interpolated onto a 0.25°×0.25° latitude-longitude grid. Trajectory starting positions can be defined easily and flexibly based on different geometrical and meteorological conditions; after the computation of the trajectories, a versatile selection is offered based on single or combined criteria (Sprenger and Wernli, 2015). First, starting positions are initialized with a suitable domain over the TP at 12:00 UTC on November 1, 2015 (or 2014). For this case, a domain from 0 to 75 and 60 °E to 120 °E was chosen. We choose starting positions in this domain that are horizontally equidistant with 80 km horizontal spacing and extend vertically from 1030 to 790 hPa with 30 hPa vertical spacing. Then, the trajectories are calculated from all starting positions 96 h backward in time. Finally, biomass burning emission data is traced along the calculated trajectories to estimate whether an air parcel at the receptor site is influenced by BC fire emissions or not. In this study, the Fire INventory from NCAR (FINN) v1.5 global fire emissions flux in 2012−2014, speciated with the GEOS-chem mechanism, was used to estimate contributions

of BC fire emission at the six selected receptor sites. FINN emission estimates are based on the framework described by Wiedinmyer et al. (2011). FINN used the satellite observations of active fires and land cover, together with emission factors and estimated fuel loadings to provide daily, highly resolved (1 km) open biomass burning emissions estimates for use in regional and global chemical transport models. Then, the same calculation was performed for the Eclipse V5 inventory for the anthropogenic contributions of BC. The Eclipse V5 inventory was widely used in the simulations. The historical data for the period 1990−2010 were revised compared to preceding sets using the latest IAE (the International Energy Agency) and FAO (the Food and Agriculture Organization) statistics extending to 2010, as well as recent country reporting where available. Note that this analysis was not exactly quantitative and did not take into account wet and dry deposition. We also assumed that the anthropogenic emissions have not changed significantly from 2010 compared to the period from 2012−2014. Thus, the comparison of natural and anthropogenic BC contributions was reasonable. The relative differences between the BC contributions can provide information on regional differences in this study.

Answer: In this study, we used an optically equivalent grain size, not an observable grain size.

In the field, the snow grains were sprayed on the MIG paper. Then we took a photo by using portable digital microscope (Anyty 3R-MSV500) (Fig. R1a). In the lab, we can measure the snow grain shape and obtain the lengths of a and b (Fig. R1b). Mugnai and Wiscombe (1987) demonstrated that a collection of unoriented non-spheroids produce the same scattering results as spheres, and Grenfell and Warren (1999) showed that the radius of a non-spherical particle was equal to that of a spherical particle that has the same volume-to-surface-area (V/A) ratio. Consequently, V/A ratio was used to transfer the measured snow grain size into the effective grain size.

On this basis of field measurement, Hao (2009) proposed two assumptions: 1) The snow particle is an inequilateral spheroid; and 2) The major axis, minor axis, and height of the inequilateral spheroid is denoted by a, b, and c, respectively, in such a way that the relationship a = 2b exists. According to Kokhanovsky and Zege (2004), the effective snow grain radius ($R_{ef}$) can be calculated equal to the radius of the volume-to-surface equivalent sphere by the following equation (1):

$$R_{ef} = \frac{3\bar{V}}{4\bar{A}} \tag{1}$$

Where, $\bar{V} = \frac{4}{3}\pi r^3$ and $\bar{A} = \pi r^2$, are the average volume and the average cross-section (geometric

shadow) area of the snow grains, respectively. And, $r$ is the radius of geometric optics, $r = \frac{a+b}{4}$.

Thus, $R_{ef} \approx 0.35a$

Eq.1 was then employed to calculate the effective snow grain size.

We have added related information in the main text (Page 5, Line2-12):

For the snow grain size observation, we sprayed the snow grains on MIG paper and took a photo using a portable digital microscope (Anyty 3R-MSV500) to calculate the major and minor axis (Fig. S2). Based on field measurements from previous studies (Mugnai and Wiscombe, 1980; Grenfell and Warren, 1999), two assumptions were proposed: 1) The snow grain particle is an inequilateral spheroid, and 2) The major axis (a), minor axis (b), and height (c) of the inequilateral spheroid are denoted by a, b, and c, respectively, in such a way that the relationship a=2b exists (Hao, 2009). According to Kokhanovsky and Zege (2004), the effective snow grain radius ($R_{ef}$) can be calculated as equal to the radius of the volume-to-surface equivalent sphere by the following equation (1):

$$R_{ef} = \frac{3\bar{V}}{4\bar{A}} \tag{1}$$

where $\bar{V} = \frac{4}{3}\pi r^3$ and $\bar{A} = \pi r^2$, are the average volume and the average cross-section (geometric shadow) area of the snow grains, respectively. In addition, $r$ is the radius of geometric optics, $r = \frac{a+b}{4}$.

Thus
$$R_{ef} \approx 0.35a \tag{2}$$

[Figure]

Figure R1. Snow grain size observation in the field (a) and measurement in the lab (b).

For the estimates of changes in snow cover duration days, we have tried to improve the related details in the main text.

**Comment 4:** Section 3.3 and 3.4 are generally confusing- was shortwave radiation (uplooking and downlooking pyranometers) actually measured? This analysis seems far too simplified. Furthermore, the discussion of snow depth is misleading. This is a straightforward energy balance calculation, so less snow will melt faster than more snow for an equal amount of forcing- this is basic mass balance. So snow depth does not itself play an important role in the reduction of snow cover by particulates. This is a mixing up of forcing and state functions. The citations are also outdated or incorrect in many cases, and I suggest the authors revisit these.

Answer: Shortwave radiation data used in this study was actually measurement by automatic weather station at different selected site near the snow sampling sites in region I, II, and III of the TP (Fig R2). Table R1 showed monthly short-wave radiation from automatic weather stations near the snow sampling sites during the snow melting season (March-May) when the temperature began to increase. On average, shortwave radiations in March, April, May, and June is about 238, 269, 292, and 271 W m$^{-2}$, respectively. Short-wave radiation in March showed the minimum value in Tanggula (210 W m$^{-2}$, lower than the average). Short-wave radiation in May showed the maximum value in Namco (314 W m$^{-2}$, higher the average). Thus, in order to estimate the impact of input short-wave radiation on the snow cover duration, we gave a range of short-wave radiation from 220 to 310 W m$^{-2}$.

We have added related information in the main text as following (Page11, Line14-23):
In Eq. (4), monthly shortwave radiation (*SW*) input data were obtained from the Automatic Weather Station (AWS) near the snow sampling sites across the TP (Fig. S3). Table S3 shows monthly shortwave radiation data from AWS during the snow melting season (March-May) when the temperature began to increase. On average, shortwave radiation in March, April, May, and June is approximately 238, 269, 292, and 271 W m$^{-2}$, respectively. Short-wave radiation in March showed the minimum value in Tanggula of the central TP (210 W m$^{-2}$, lower than the average), whereas in May it showed the maximum value in Namco (314 W m$^{-2}$, higher the average). Based on these data during the melt season, three scenarios (220, 270, and 310 W m$^{-2}$)were defined as the minimum, median, and maximum scenarios of input shortwave radiation to estimate its impact on changes of snow cover duration.

[Figure]

Figure R4. Automatic weather stations selected in the Tibetan Plateau.

Table R1. Monthly short-wave radiation from automatic weather stations near the snow sampling sites during the snow melting season.

| Sites | Month of the year | Monthly Short-wave radiation (W m$^{-2}$) | Month of the year | Monthly Shortwave radiation (W m$^{-2}$) |
|---|---|---|---|---|
| Southeastern Tibetan Plateau | | | | |
| | March in 2014 | 248 | March in 2015 | 237 |
| | April in 2014 | 288 | April in 2015 | 259 |
| | May in 2014 | 305 | May in 2015 | 280 |
| | June in 2014 | 265 | June in 2015 | 250 |
| Namco | | | | |
| | | | March in 2015 | 264 |
| | | | April in 2015 | 276 |
| | | | May in 2015 | 306 |
| | | | June in 2015 | 314 |
| Tanggula | | | | |
| | March in 2014 | 210 | | |
| | April in2014 | 226 | | |
| | May in 2014 | 245 | | |
| | June in 2014 | 271 | | |
| Laohugou | | | | |
| | March in 2014 | 229 | March in 2015 | 238 |
| | April in2014 | 269 | April in 2015 | 294 |
| | May in 2014 | 311 | May in 2015 | 305 |
| | June in 2014 | 258 | June in 2015 | 269 |

We agree that the equation for changes of snow cover duration is a straightforward energy balance calculation. Thus in this study, it means the duration days of less snow will be less than the more snow for an equal amount of forcing. So under the same level of LAIs in snow, the reduction of snow cover duration days can be affected by different snow depth. We have tried to revise in the main text (section 3.4).

**Comment 5:** I take issue with the use of the 'Third Pole' term, this is not universally recognized and is somewhat politicized, why not just use Tibetan Plateau or High Mountain Asia? Also light absorbing impurity is an outdated term, the community has moved toward the use of light absorbing particulates. Also mineral dust and the acronym MD are confusing you can simply say dust- which needs no acronym. Similarly, please be consistent in terminology, for example, albedo and reflectance are not the same thing.
Answer: We have changed the term "Third Pole" to be "Tibetan Plateau".

We have used the term "light-absorbing particulates" in the main text.

We have changed the "MD" as "dust" in the main text.

Albedo and reflectance are not the same thing. Albedo is defined as the ratio of irradiance reflected to the irradiance received by a surface. The proportion reflected is not only determined by properties of the surface itself, but also by the spectral and angular distribution of solar radiation reaching the Earth's surface. Unless given for a specific wavelength (spectral albedo), albedo refers to the broadband spectrum of solar radiation.

Reflectance of the surface of a material is its effectiveness in reflecting radiant energy. It is the fraction of incident electromagnetic power that is reflected at an interface. The reflectance spectrum or spectral reflectance curve is the plot of the reflectance as a function of wavelength. When we use ASD, we get the reflectance. In order to compare the albedo simulated by SNICAR model in this study, we have to calculate the reflectance based on the standard solar irradiance.
We also carefully use the words consistent in the main text.

**Comment 6:** Uncertainties in modeling are not only due to lack of observations, and increasing our number of sampling points alone will not reduce our model uncertainty. To state this is misleading.
Answer: Agree, and we have tried to revise this section in the section3.4 of the main text.

[revised manuscript text omitted]

18 **Including:**

19 **Supplementary Tables S1-S3**
20 **Supplementary Figure S1-S5**

**Supplementary Tables**

**Table S1.** Sampling information for snow cover over the Tibetan Plateau.

**Table S2.** Parameters for sensitivity analysis with SNICAR model for snow cover in the Tibetan Plateau. 1-Incident radiation (a. Direct, b. Diffuse); 2- Solar zenith angle; 3- Surface spectral distribution (a. Mid-latitude winter, clear-sky, cloud amount<5. b. Mid-latitude winter, cloudy, cloud amount ≥5); 4- Snow grain effective radius (µm); 5- Snowpack thickness (m); 6- Snowpack density (kg m$^{-3}$); 7- Albedo of underlying ground (a. Visible, 0.3–0.7µm, b. Near-infrared, 0.7–5.0µm); 8-MAC scaling factor (experimental) for BC; 9-BC concentration (ppb, sulfate-coated); 10- Dust concentration (µg g$^{-1}$, 5.0–10.0 µm diameter); 11- Volcanic ash concentration (ppm); 12- Experimental particle 1 concentration (ng g$^{-1}$).

**Table S3.** Short-wave radiation input data at selected sites across the Tibetan Plateau close to the snow sampling sites.

 **Table S1.** Sampling information for snow cover over the Tibetan Plateau regions.

| Sites | Time | Lat. (°) | Long. (°) | Elevation (m) | Snow depth (cm) | Density (kg m$^{-3}$) | Observed snow grain size (µm) | Snow type | Snow temperature (°C) |
|---|---|---|---|---|---|---|---|---|---|
| LHG | 2015-12-29 12:00 | 39.50 | 96.52 | 4250 | 20 | | | | |
| RYS | 2014-12-03 12:25 | 36.50 | 101.52 | 3396 | | | | | |
| TP-S-1511-36 | 2015-11-30 16:48 | 35.44 | 98.65 | 4363 | 11 | 300 | 2000 | Coarse firn with hoar | −6.5 |
| TP-S-1511-37 | 2015-11-30 18:10 | 35.37 | 99.29 | 4297 | 16 | 250 | 2000 | Coarse firn with wind crust and dirty layer | −7.0 |
| TP-S-1511-38 | 2015-12-01 10:09 | 35.39 | 99.31 | 4298 | 18 | 225 | 2000 | | −15.0 |
| TP-S-1511-39 | 2015-12-01 11:23 | 35.51 | 99.51 | 4405 | 14 | 225 | 2000 | Coarse firn | −6.5 |
| TP-S-1511-35 | 2015-11-30 13:25 | 34.11 | 97.65 | 4781 | 10 | 325 | 2000 | Coarse firn with dirty layer | −5.0 |
| MD | 2014-12-03 12:10 | 34.86 | 98.17 | 4220 | | | | | |
| TP-S-1511-33 | 2015-11-30 11:12 | 33.47 | 97.24 | 4259 | 7 | 225 | 1000 | Coarse firn | −4 |
| TP-S-1511-34 | 2015-11-30 11:41 | 33.61 | 97.24 | 4366 | 12 | 350 | 2000 | Coarse firn with wind crust | −6.0 |
| TGL | 2014-12-10 12:40 | 32.90 | 91.92 | 5163 | | | | | |
| LWQ | 2014-12-04 12:20 | 31.22 | 96.57 | 4370 | | | | | |
| 24K | 2014-12-06 12:47 | 30.99 | 91.68 | 5100 | | | | | |
| NMC | 2014-12-09 12:30 | 30.75 | 91.68 | 4729 | | | | | |
| MYL | 2014-12-10 12:30 | 30.11 | 83.39 | 4610 | | | | | |
| TP-S-1511-1 | 2015-11-15 11:30 | 30.45 | 91.05 | 4216 | 3 | | 200 | Firn | −5.0 |
| TP-S-1511-2 | 2015-11-15 13:33 | 30.67 | 91.10 | 5036 | 6.8 | | 1000 | Firn | −3.7 |
| TP-S-1511-3 | 2015-11-15 14:00 | 30.68 | 91.10 | 5124 | 9 | | 1000 | Fine firn | −0.14 |
| TP-S-1511-4 | 2015-11-15 16:44 | 30.79 | 90.96 | 4687 | 5 | | 1000 | Fine firn | −2 |
| TP-S-1511-5 | 2015-11-15 17:30 | 30.68 | 91.10 | 5096 | 17.1 | | 500 | Firn with wind crust | −4.8 |
| TP-S-1511-6 | 2015-11-16 13:52 | 29.19 | 90.62 | 4762 | 6.8 | | 1000 | Firn | −3.7 |
| TP-S-1511-16 | 2015-11-20 09:13 | 29.00 | 87.47 | 4798 | 9 | 300 | 1000 | Fine firn | −17 |
| TP-S-1511-27 | 2015-11-26 12:37 | 29.83 | 92.34 | 4971 | 17 | 150 | 3500 | Coarse firn | −5 |
| TP-S-1511-28 | 2015-11-27 10:55 | 29.63 | 94.63 | 4304 | 12 | 200 | 2000 | | −1.5 |
| TP-S-1511-29 | 2015-11-27 11:25 | 29.61 | 94.67 | 4523 | 24 | 200 | 3000 | Hoar at the bottom | −4.5 |
| TP-S-1511-30 | 2015-11-27 17:48 | 29.77 | 95.70 | 3728 | 11 | 400 | 2000 | | −1.5 |
| TP-S-1511-31 | 2015-11-28 12:10 | 29.26 | 96.94 | 4582 | 12 | 250 | 2000 | Coarse firn, ice at bottom | −0.05 |
| TP-S-1511-32 | 2015-11-28 1:45 | 29.32 | 97.03 | 4745 | 20 | 175 | 2000 | Fresh snow and firn | −1.0 |
| TP-S-1511-9 | 2015-11-18 12:13 | 28.82 | 87.93 | 4851 | 12.5 | 230 | 1500 | Firn with wind crust and dirty layer | −12.6 |
| TP-S-1511-17 | 2015-11-20 10:02 | 28.96 | 87.44 | 5163 | 15 | 270 | 2000 | Firn with wind crust and dirty layer | −17 |
| TP-S-1511-18 | 2015-11-20 10:35 | 28.91 | 87.43 | 5042 | 8 | 250 | 3000 | Firn with wind crust and dirty layer | −15 |
| TP-S-1511-21 | 2015-11-21 13:13 | 28.67 | 86.13 | 4565 | 9 | 200 | 2000 | Coarse firn | −2.0 |
| TP-S-1511-22 | 2015-11-21 14:09 | 28.52 | 86.17 | 5096 | 5 | 200 | 2000 | Coarse firn | 0 |
| TP-S-1511-23 | 2015-11-22 13:12 | 28.65 | 86.08 | 4592 | 9 | 325 | 3000 | Coarse firn with wind crust and hoar | |
| TP-S-1511-24 | 2015-11-22 14:11 | 28.76 | 85.66 | 4614 | 14 | 250 | 3000 | Coarse firn | 0.05 |
| TP-S-1511-25 | 2015-11-22 15:07 | 28.90 | 85.40 | 4861 | 11.5 | 225 | 3000 | Coarse firn | −2.5 |
| TP-S-1511-26 | 2015-11-22 15:55 | 28.93 | 85.40 | 5166 | 12 | 275 | 1000 | Fine firn with wind crust | 0 |
| TP-S-1511-8 | 2015-11-17 13:02 | 27.39 | 88.83 | 4150 | 5 | 275 | 1000 | Fine firn | −0.4 |

**Table S2.** Parameters for sensitivity analysis with SNICAR model for snow cover in the Tibetan

Plateau.

| Site | 1 | 2 | 3 | 4 | 5 | 6 | 7a | 7b | 8 | 9-BC | 10-Dust | 11 | 12 |
|------|---|---|---|---|---|---|----|----|---|------|---------|----|----|
| LHG* | direct | 62.75 | a | | 20 | | 0.15 | 0.3 | 1 | 3745.46 | 224.49 | 0 | 0 |
| RYS | direct | 58.58 | a | | 11 | | 0.15 | 0.3 | 1 | 8922.53 | 417.65 | 0 | 0 |
| TP-S-1511-36 | direct | 57.02 | a | | 11 | | 0.15 | 0.3 | 1 | 3097.59 | 172.04 | 0 | 0 |
| TP-S-1511-37 | direct | 56.95 | a | | 16 | | 0.15 | 0.3 | 1 | 7373.45 | 292.97 | 0 | 0 |
| TP-S-1511-38 | direct | 57.13 | a | | 18 | | 0.15 | 0.3 | 1 | 4122.43 | 233.86 | 0 | 0 |
| TP-S-1511-39 | direct | 57.25 | a | | 14 | | 0.15 | 0.3 | 1 | 2894.96 | 157.78 | 0 | 0 |
| TP-S-1511-35 | direct | 55.69 | a | | 10 | | 0.15 | 0.3 | 1 | 4217.19 | 295.80 | 0 | 0 |
| MD | direct | 56.94 | a | | 11 | | 0.15 | 0.3 | 1 | 3122.68 | 231.03 | 0 | 0 |
| TP-S-1511-33 | direct | 55.05 | a | | 7 | | 0.15 | 0.3 | 1 | 17468.24 | 846.39 | 0 | 0 |
| TP-S-1511-34 | direct | 55.19 | a | | 12 | | 0.15 | 0.3 | 1 | 4009.22 | 211.41 | 0 | 0 |
| TGL | direct | 55.8 | a | | 11 | | 0.15 | 0.3 | 1 | 1014.18 | 89.61 | 0 | 0 |
| LWQ | direct | 53.44 | a | | 11 | | 0.15 | 0.3 | 1 | 1734.68 | 71.11 | 0 | 0 |
| 24K | direct | 53.46 | a | | 11 | | 0.15 | 0.3 | 1 | 876.75 | 113.52 | 0 | 0 |
| NMC | direct | 53.55 | a | | 11 | | 0.15 | 0.3 | 1 | 600.68 | 45.91 | 0 | 0 |
| MYL | direct | 53.01 | a | | 11 | | 0.15 | 0.3 | 1 | 319.89 | 23.92 | 0 | 0 |
| TP-S-1511-1 | direct | 48.83 | a | | 3 | | 0.15 | 0.3 | 1 | 1283.33 | 125.53 | 0 | 0 |
| TP-S-1511-2 | direct | 49.05 | a | | 6.8 | | 0.15 | 0.3 | 1 | 468.71 | 54.14 | 0 | 0 |
| TP-S-1511-3 | direct | 49.06 | a | | 9 | | 0.15 | 0.3 | 1 | 5093.10 | 352.58 | 0 | 0 |
| TP-S-1511-4 | direct | 49.17 | a | | 5 | | 0.15 | 0.3 | 1 | 4192.93 | 213.11 | 0 | 0 |
| TP-S-1511-5 | direct | 49.06 | a | | 17.1 | | 0.15 | 0.3 | 1 | 1084.93 | 109.12 | 0 | 0 |
| TP-S-1511-6 | direct | 47.83 | a | | 6.8 | | 0.15 | 0.3 | 1 | 250.69 | 42.80 | 0 | 0 |
| TP-S-1511-16 | direct | 48.6 | a | | 9 | | 0.15 | 0.3 | 1 | 886.69 | 86.78 | 0 | 0 |
| TP-S-1511-27 | direct | 50.69 | a | | 17 | | 0.15 | 0.3 | 1 | 3586.99 | 323.06 | 0 | 0 |
| TP-S-1511-28 | direct | 50.68 | a | | 12 | | 0.15 | 0.3 | 1 | 952.23 | 55.50 | 0 | 0 |
| TP-S-1511-29 | direct | 50.66 | a | | 24 | | 0.15 | 0.3 | 1 | 255.93 | 26.86 | 0 | 0 |
| TP-S-1511-30 | direct | 50.82 | a | | 11 | | 0.15 | 0.3 | 1 | 1343.22 | 127.57 | 0 | 0 |
| TP-S-1511-31 | direct | 50.49 | a | | 12 | | 0.15 | 0.3 | 1 | 2117.49 | 238.03 | 0 | 0 |
| TP-S-1511-32 | direct | 50.55 | a | | 20 | | 0.15 | 0.3 | 1 | 201.70 | 21.58 | 0 | 0 |
| TP-S-1511-9 | direct | 47.95 | a | | 12.5 | | 0.15 | 0.3 | 1 | 3417.06 | 344.93 | 0 | 0 |
| TP-S-1511-17 | direct | 48.56 | a | | 15 | | 0.15 | 0.3 | 1 | 822.01 | 92.20 | 0 | 0 |
| TP-S-1511-18 | direct | 48.51 | a | | 8 | | 0.15 | 0.3 | 1 | 1277.70 | 101.71 | 0 | 0 |
| TP-S-1511-21 | direct | 48.5 | a | | 9 | | 0.15 | 0.3 | 1 | 1079.43 | 101.39 | 0 | 0 |
| TP-S-1511-22 | direct | 48.35 | a | | 5 | | 0.15 | 0.3 | 1 | 2201.12 | 226.49 | 0 | 0 |
| TP-S-1511-23 | direct | 48.7 | a | | 9 | | 0.15 | 0.3 | 1 | 1072.04 | 104.53 | 0 | 0 |
| TP-S-1511-24 | direct | 48.81 | a | | 14 | | 0.15 | 0.3 | 1 | 1042.25 | 109.11 | 0 | 0 |
| TP-S-1511-25 | direct | 48.95 | a | | 11.5 | | 0.15 | 0.3 | 1 | 931.68 | 35.29 | 0 | 0 |
| TP-S-1511-26 | direct | 48.98 | a | | 12 | | 0.15 | 0.3 | 1 | 759.41 | 65.26 | 0 | 0 |
| TP-S-1511-8 | direct | 46.28 | a | | 5 | | 0.15 | 0.3 | 1 | 574.95 | 22.39 | 0 | 0 |

37 **Table S3.** Short-wave radiation input data at selected sites across the Tibetan Plateau close to the
38 snow sampling sites.

| | | SW (W m$^{-2}$) | | SW (W m$^{-2}$) |
|---|---|---|---|---|
| SETP | | | | |
| | March, 2014 | 248 | March, 2015 | 237 |
| | April, 2014 | 288 | April, 2015 | 259 |
| | May, 2014 | 305 | May, 2015 | 280 |
| | June, 2014 | 265 | June, 2015 | 250 |
| NMC | | | | |
| | | | March, 2015 | 264 |
| | | | April, 2015 | 276 |
| | | | May, 2015 | 306 |
| | | | June, 2015 | 314 |
| TGL | | | | |
| | March, 2014 | 210 | | |
| | April, 2014 | 226 | | |
| | May, 2014 | 245 | | |
| | June, 2014 | 271 | | |
| LHG | | | | |
| | March, 2014 | 229 | March, 2015 | 238 |
| | April, 2014 | 269 | April, 2015 | 294 |
| | May, 2014 | 311 | May, 2015 | 305 |
| | June, 2014 | 258 | June, 2015 | 269 |

**Supplementary Figures**

**Figure S1.** Example of snow sampling sites over the Tibetan Plateau.

**Figure S2.** Snow grain size observation in the field (a) and measurement in the lab (b).

**Figure S3.** BC emissions arrived at the selected sites over the Tibetan Plateau based on data drawn from http://inventory.pku.edu.cn/download/download.html (Wang et al., 2014). (A value 2 on the x axis means $2\times10^6$ g month$^{-1}$. The BC value is averaged over 48 h along the back trajectory path. And the value on the y axis is the number of trajectories in the respective BC bin. The bin width is 0.05, i.e. from 0 to 5 on the horizontal axis there are 100 bins.)

**Figure S3.** Automatic weather stations selected for the short-wave radiation input data across the Tibetan Plateau.

**Figure S4.** BC emissions arrived at the selected sites over the Tibetan Plateau region based on data drawn from http://inventory.pku.edu.cn/download/download.html (Wang et al., 2014). (A value 2 on the x axis means $2\times10^6$ g month$^{-1}$. The BC value is averaged over 48 h along the back trajectory path. And the value on the y axis is the number of trajectories in the respective BC bin. The bin width is 0.05, i.e. from 0 to 5 on the horizontal axis there are 100 bins.)

**Figure S5.** Distributions of changes of snow cover duration days caused by BC and mineral dust.

59    **Fig S1.**

[Figure]

60

61
62    **Figure S1.** Example of snow sampling sites over the Tibetan Plateau.

63 **Fig S2.**

[Figure]

64
65 **Figure S2.** Snow grain size observation in the field (a) and measurement in the lab (b).

66 **Fig S3.**

[Figure]

67
68 Figure S3. Automatic weather stations selected in the Tibetan Plateau.

 **Fig S4.**

[Figure]

**Figure S4.** BC emissions arrived at the selected sites over the Tibetan Plateau region based on data drawn from http://inventory.pku.edu.cn/download/download.html (Wang et al., 2014). (A value 2 on the x axis means $2\times10^6$ g month$^{-1}$. The BC value is averaged over 48 h along the back trajectory path. And the value on the y axis is the number of trajectories in the respective BC bin. The bin width is 0.05, i.e. from 0 to 5 on the horizontal axis there are 100 bins.)

**Fig S5.**

[Figure]

**Figure S5.** Distributions of changes of snow cover duration days caused by BC and mineral dust.

References:

Wang, R., Tao, S., Shen, H., Huang, Y., Chen, H., Baljanski, Y., Boucher, O., Ciais, P., Shen, G., Li, W., Zhang, Y., Chen, Y., Lin, N., Su, S., Li, B., Liu, J., Liu, W.: Trend in black carbon emissions from 1960 to 2007, Environ. Sci. Technol., doi:10.1021/es5021422, 2014.

---

## Editor Decision (ED1)

Dear Professor Kang,

Both reviewers have significant concerns regarding the description of your methods (both observations and modeling) and the analysis of your results. Unfortunately they have not been willing to review your responses to their comments or your revised manuscript. I have carefully read through your responses and your revised manuscript and have provided my comments below. There are a couple of referee comments that I don't feel were adequately addressed. Consider my comments and submit your revisions after you've had a native English speaker read through any new text. Please respond point by point, and be sure to highlight all new changes in your revised manuscript.

Regards,

Becky

**General comments:**

- It is unclear what "footprint analysis" means. Please replace with "back trajectory analysis" in all instances or define what you mean by footprint if they are not the same.
- Data available upon request does not meet TC policy. You need to provide your BC and LAP data here as supplementary information or provide a DOI containing the data per TC policy: http://www.the-cryosphere.net/about/data_policy.html#data_availability

**Referee comments:**

- Referee #1's comment on page18 (page 10 line 13) seems not to have been addressed. You need to address this on page 13.
- Referee #2 comment #1: Please mention this as a source of uncertainty in your analysis (section 3.4) in the main text.
- Page 5 lines 10-12: You describe grain size, but not snow depth or density measurements. Please describe methods for all measurements, as requested by the reviewers.
- Page 7 lines 24-28: The lines in blue do not answer the reviewer's question. Did you assume uniform BC throughout the snow depth? It looks like that is the only option in the inputs. If so, what sort of error does this represent?
- Page 9 line 19: "kind of biomass burning" Can you be more explicit here? I'm still not sure what you mean by "open".
-

**Unclear wording or missing information. Much of this is in the methods section, which has improved somewhat, but I think still will not satisfy the reviewers concerns on this point. Specific comments are below.**

- Page 2 line 12: insert "a wavelength ($\lambda$) of" before 440 nm. Elsewhere, insert "$\lambda$ = " before any number representing a wavelength.
- Page 2 line 25: 142 – 271 mm of what? Snow or ice mass?
- Page 2 line 27: "days during ablation through" doesn't make sense. Please check this and reword.
- Page 2 line 31: It doesn't make sense that the presence of dust suggests a relative role for BC. Dust and BC have different sources.

- Page 4 lines 28-29: You characterize region I and II, but not region III. What characterizes region III and differentiates it from I and II? What do you mean by "intensified sampling"? What is "LHG"?
- Page 4 line 30: Is "LHG" the same as region III?
- Page 5 line 6: Is there a region you mention (c) but don't use it in the equations below?
- Page 5 line 17: This is too qualitative. What is the mass fraction of dust compared to BC and OC?
- Page 5 line 24: which is it, 550 or 850? Why one or the other?
- Page 5 line 27: How was the carbon converted to methane?
- Page 5 line 31: Is this the range for the detection limit? 750 seems high.
- Page 6 line 17: need units for the numbers 0 and 75
- Page 7 line 9: What is "standard solar irradiance"? Does this mean you don't have observations of surface irradiance? Remove "to get the albedos".
- Page 7 line 22-23: provide units for all variables.
- Page 7 line 31: Is incoming shortwave radiation the same as surface irradiance?
- Page 8 line 1: How is "solar shortwave insolation" different from "incoming shortwave radiation"?
- Page 8 line 8: Is the observed snow depth based on previous studies? If so reference them here.
- Page 8 lines 6-9: Include units in parentheses for all variables in the text where they are described.
- Page 8 line 32: Have higher OC/EC ratios than what? Fossil fuels?
- Page 9 line 20: "for half of the BC deposition" How was "half" determined, quantitatively? Same with 30% on line 22 and 70% on line 26.
- Page 10 line 11: What is the upper boundary and how does it compare to your grain size measurements?
- Page 10 line 7: insert $\lambda$ to be more explicit what these numbers are referring to.
- Page 10 line 22: What do you mean by "radiative flux"? Can you express in terms of previously used variables, such as RF or albedo?
- Page 11 line 15: What is shortwave radiation "input data"? Is it surface irradiance? You need to be consistent in your defined terms throughout the manuscript.
- Page 12 line 18: What are Koch snowflakes?
- Page 13 line 11: "Snow cover durations were shortened during the melt season from X-Y days…" Insert numbers for X and Y.
- Page 13 line 13: "annual mass budget" Is this a loss of mass? "mass budget" does not necessarily imply a loss.
- Page 14 line 2: Wasn't this study a survey of LAP, and not snow cover?
- Page 14 line 13: What is radiation input data? Is this observed surface irradiance?

**Although the grammar and wording are somewhat improved, there are still some issues. Some specific suggestions on word changes are below which should help to clarify the manuscript.**

- Page 1 line 20: insert "relative" before "biomass burning"
- Page 1 line 21: insert "relative" before "contribution"
- Page 1 line 23: delete "changes of"

- Page 2 line 10: change "by" to "of"
- Page 2 line 16: replace "the simulation showed" with "suggested"
- Page 2 line 17: deposited **to** land snow, not in Page 2 line 17: put a comma after W m$^{-2}$
- Page 2 line 17: replace "or as large as" with "contributing as much as"
- Page 2 line 19: replace "in" with "for the"
- Page 2 line 20: insert "also" between "can" and "change"
- Page 2 line 23: replace "in particular, from" with "especially $\lambda$ = "
- Page 2 line 24: replace "on Claridenfirn of" with "in"
- Page 2 line 26: remove comma after "region"
- Page 2 line 27: what is "disturbed" desert dust?
- Page 2 line 29: remove "even"
- Page 3 line 22: replace "seldom" with "sparse"
- Page 3 line 26: remove "cover" after "snow"
- Page 3 line 27: change first comma to a period to end the sentence. Then start the next sentence with "We further…"
- Page 3 line 28: insert "observed" before LAPs.
- Page 3 line 28-29: should read "We use back trajectory analysis coupled with BC fire emission inventories to approximate natural/anthropogenic contributions"
- Page 3 line 29: increase *our* understanding
- Page 3 line 30: insert comma after TP and remove the next word "and"
- Page 3 line 30: replace "this is also helpful for" with "inform"
- Page 4 line 5: remove "in the earth system"
- Page 4 line 17: remove "on"
- Page 4 line 27: replace "contains" with "includes"
- Page 5 line 23: remove "special"
- Page 5 line 25: Should read "The sample is reheated further in a stepwise fashion to near 900…"
- Page 5 line 26: Should read "burn out all remaining BC"
- Page 5 lines 27-30: Move this sentence up to after "…of the OC in the sample."  Change "modified .. arranged" to "limited the initial temperature plateau"
- Page 6 line 6: Should read: "…used approach for determining source regions of various atmospheric species."
- Page 6 line 7: remove "that"
- Page 6 line 7: can be *qualitatively* attributed
- Page 6 line 19: Remove "Then,"
- Page 6 line 23: Should read "FINN used satellite" (remove "the")
- Page 6 line 25: Remove "Then,"
- Page 6 lines 29-30: Should read: "Note that this analysis is qualitative and does not take into account loss from wet and dry deposition"
- Page 6 lines 31-32: Remove sentence beginning "Thus,"
- Page 6 line 32: Should read "Despite these uncertainties, the relative differences between the BC contributions are used to…"
- Page 7 line 2: remove second comma
- Page 7 line 25: Should read "will lead to discrepancies between"

- Page 8 line 6: insert "and" before "for"
- Page 8 line 7: should read "clean snow due to changing snow grain sizes with show age"
- Page 8 line 28: replace "values" with "concentrations"
- Page 8 line 29: replace "by the soil" with "from the soil"
- Page 8 line 30: replace "represent" with "examine"
- Page 9 line 3: replace "little" with "slightly"
- Page 9 line 13: remove "arrived at"
- Page 10 line 2: replace "simulate" with "estimate the"
- Page 10 line 13: Should read "albedo differences between measurements and simulations is less"
- Page 10 line 20: remove "when testing regional to global scale models"
- Page 11 line 5: Replace "validation" with "assessment". It is impossible to "validate" a model (although I realize this term is frequently used in the literature)
- Page 11 line 11: Should read "Thus, this does doesn't include the effect of OC on estimates of"
- Page 11 line 14: Changes *in* snow cover
- Page 11 line 24: low *SD* scenarios
- Page 12 line 19: A lack of OC *consideration* due to
- Page 12 line 25: Remove "We also have to pay attention to the fact that". Replace "their" with "dust"
- Page 12 line 26: replace designed with "assumed".
- Page 12 line 27: efficient *at* light scattering
- Page 12 line 28: replace "materials that" with "and"
- Page 13 line 8: replace "the estimation" with "estimates"
- Page 13 line 30-31: "would have been marginal" depends on assumed concentrations. Maybe say "uncertain" instead of "marginal"
- Page 14 line 14: reduced *snow cover* from several
- Page 14 line 19-20: "total mass balance" should read "mass lost" if I'm understanding this correctly

---

## Author Response (AR2)

Dec22, 2017

Dear editor,

Thank you and the reviewers very much for all your time and efforts regarding our manuscript (**tc-2017-111: Black carbon and mineral dust in snow cover on the Tibetan Plateau**). We have carefully revised the manuscript according to the reviewers' comments. Detailed responses are in blue in the main text. We also included the revised version below with changes shown in blue. We hope that you and reviewers find the revisions appropriate and adequate.

Yours sincerely,

Yulan ZHANG and Shichang KANG
* * *
**General comments:**

- It is unclear what "footprint analysis" means. Please replace with "back trajectory analysis" in all instances or define what you mean by footprint if they are not the same.

  **Answer:** Thanks and agree, and have revised in the main text.

- Data available upon request does not meet TC policy. You need to provide your BC and LAP data here as supplementary information or provide a DOI containing the data per TC policy: http://www.the-cryosphere.net/about/data_policy.html#data_availability

  **Answer:** We have provided the data of light-absorbing particulates in this study as a supplementary information.

**Referee comments:**

- Referee #1's comment on page18 (page 10 line 13) seems not to have been addressed. You need to address this on page 13.

**Referee #1's comment** [Page 10: Line 13: "However, the results presented in this study : : :. for which these assumptions are not critical": This is not true. All the quantities listed in this paragraph will influence the snow albedo and most of them will influence the albedo reduction induced by BC. For example: BC-snow internal mixing increases the albedo forcing by 40-60% compared with external mixing (He et al. (2014). The author should discuss the uncertainty of this study resulted from the assumptions they made, instead of claim these quantities will not impact their results. References: He, C., Q. Li, K. N. Liou, Y. Takano, Y. Gu, L. Qi, Y. Mao, and L. R. Leung, 2014: Black Carbon Radiative Forcing over the Tibetan Plateau. Geophy. Res. Lett., 41, 7806-7813, doi: 10.1002/2014GL062191.]

  **Answer:** Agree, and we have added related discussion in this section in the main text as suggested. (Page 12, Line 31-33).

  ``````BC in the atmosphere tended to mix with OC and inorganic salts during aging enhancing its absorption (Gustafsson and Ramanathan, 2016). He et al. (2014) noted that BC-snow internal mixing

increases the albedo forcing by 40–60 % compared with external mixing, and coated BC increases the forcing by 30–50 % compared with uncoated BC aggregates, whereas Koch snowflakes reduce the forcing by 20–40 % relative to spherical snow grains.```````

- Referee #2 comment #1: Please mention this as a source of uncertainty in your analysis (section 3.4) in the main text.

**Referee #2 comment #1** [My first and foremost concern is that samples were collected in November and December, and yet they are attempting to quantify the impacts of light absorbing particulates on melt. This does not make sense to me and if I misunderstood this it is because it is not made clear in the manuscript. Although the sample collection timing may be after the summer monsoon, these samples do not represent the impurities that are present during the ablation season- and therefore it is inappropriate to use these values to quantify reduction in snowpack duration. Particularly for dust, which tends to deposit in the spring when source regions dry out (peak radiative forcing by dust in snow is observed from MODIS imagery over the Himalayan region in April and May).]

**Answer:** Agree, we have added the related information in the main text in section 3.4 (Page 12, Line 22-28).

These samples were collected during winter season when the snow cover were more stable and continuous, which might not represent the true impurities as these during the ablation season. However, during melting season, because of the poor accessibility on the snowpack in the TP, it was hardly to collect the snow cover samples. Besides, considerable heterogeneity in the topography and climate has led to complex spatial and temporal snow cover patterns (Xu et al., 2017). Discontinued snow cover during melting season may be a problem to represent the true impurities in snow. Thus in our study, we gave an estimation based on the different snow grain size and density, rather than the fixed data. More work needs to do to fill in gaps in the future.

- Page 5 lines 10-12: You describe grain size, but not snow depth or density measurements. Please describe methods for all measurements, as requested by the reviewers.

**Answer:** In this study, snow depth was measured using a ruler. Snow density was measured using weighing method of specific volume of snow. We have added these information in the main text of section 2.2 (Page 5, Line 4-5).

- Page 7 lines 24-28: The lines in blue do not answer the reviewer's question. Did you assume uniform BC throughout the snow depth? It looks like that is the only option in the inputs. If so, what sort of error does this represent?

**Answer:** In this study, BC was assumed to be uniform throughout the snow depth as described in the SNICAR model. This model is a single-layer implementation model (Flanner et al., 2007, 2009). Due to melting processes, BC may be accumulated on the snow surface, while in the subsurface of snow pack, BC concentration was lower. In SNICAR, the vertical distributions of BC in snow is not consider. Lacking a radiative transfer approximation for BC, Flanner et al. (2007) assume snow forcing acts homogeneously over snow-covered and snow-free ice. This crude assumption probably underestimates forcing. Because the SNICAR cannot provide the scenarios of BC in vertical distribution, we couldn't estimate the errors caused by BC's distribution in snowpack.

- Page 9 line 19: "kind of biomass burning" Can you be more explicit here? I'm still not sure what you mean by "open".

**Answer:** In this study, the open burning of biomass data we used include "wildfire, agricultural fires, and prescribed burning, and dose not included biofuel use and trash burning" (Wiedinmyer et al., 2011). Thus, we have changed in the main text (Page9, Line19-20). We also explained it in the section 2.4 (Page 6, Line 30-31).

**Unclear wording or missing information. Much of this is in the methods section, which has improved somewhat, but I think still will not satisfy the reviewers concerns on this point. Specific comments are below.**

**Answer:** Thank you very much for all the comments. We have carefully learned these suggestions, and revised in the main text.

- Page 2 line 12: insert "a wavelength ( ) of" before 440 nm. Elsewhere, insert " = " before any number representing a wavelength.

**Answer:** Agree, and added the related information before the number representing a wavelength.

- Page 2 line 25: 142 – 271 mm of what? Snow or ice mass?

**Answer:** In the study (Gabbi et al., 2015), maximum deviations in annual mass balance due to Saharan dust were up to -142 mmw.e. $yr^{-1}$ for the upper and -271 mmw.e.$yr^{-1}$ for the lower measurement site in individual years. Thus in this study, it means a range of 142 – 271 mm water equivalent. We have added the information in the main text (Page 2, line 27).

- Page 2 line 27: "days during ablation through" doesn't make sense. Please check this and reword.

**Answer:** We have changed this sentence in the main text.

"```` be shortened by 18 to 35 days during ablation period due to surface shortwave RF caused by deposition of disturbed desert dust"

- Page 2 line 31: It doesn't make sense that the presence of dust suggests a relative role for BC. Dust and BC have different sources.

**Answer:** Agree, they have different sources. Here "the relevant role" means both of BC and dust can reduce snow surface albedo and darkening the surface. We have changed this sentence in the main text. "The presence of dust in snow suggests it also plays an important role in darkening the glacier surface." (Page 2, Line 33)

- Page 4 lines 28-29: You characterize region I and II, but not region III. What characterizes region III and differentiates it from I and II? What do you mean by "intensified sampling"? What is "LHG"?

**Answer:** The stable oxygen isotope ratio ($\delta^{18}O$) in precipitation is an integrated tracer of atmospheric moisture worldwide. In the previous study in the Tibetan Plateau (Yao et al., 2013; Tian et al., 2007), the researchers have established a database of precipitation $\delta^{18}O$ and use different models to evaluate the climatic controls of precipitation $\delta^{18}O$ over the Tibetan Plateau. The spatial and temporal patterns of precipitation $\delta^{18}O$ and their relationships with temperature and precipitation reveal three distinct domains, respectively associated with the influence of the westerlies (northern Tibetan Plateau, region III), Indian

monsoon (southern Tibetan Plateau, region I ), and transition in between (region II). Therefore, we have added related information in the main text (Page 4, Line 26-28).

The "intensified sampling" here means systematic sampling for a whole winter season. We deleted the word "intensified" in the sentence.

"LHG" means the Laohugou No.12 glacier region, we have added the information in the main text (Page 4, Line 24).

- Page 4 line 30: Is "LHG" the same as region III?

**Answer:** "LHG" is not the same as region III. LHG is located in the Region III, it can represent the snow samples we collected in the region III. We have changed the sentence in the main text (Page 4, Line 30-31).

- Page 5 line 6: Is there a region you mention (c) but don't use it in the equations below?

**Answer:** We don't use c in the equations. Thus we revised this sentence in the main text (Page 5, Line 9).

- Page 5 line 17: This is too qualitative. What is the mass fraction of dust compared to BC and OC?

**Answer:** In the Tibetan Plateau, Zhang et al. (2016) showed the dust number concentrations in snowpit from different glaciers (as shown in Table 2 in the publication). The average density of upper crust is about 2.8 g cm$^{-3}$. Median size of the dust in snow of the Tibetan Plateau is about 3 μm (d) (Xu et al., 2010). Thus, the equivalent sphere volume $V = (4/3)\pi(d/2)^3$.

Therefore, the mass concentration of dust in snow of the Tibetan Plateau is about hundreds of μg g$^{-1}$ based on number concentrations. The unit of BC and OC in snow is ng g$^{-1}$, which means around three magnitudes lower than the mass concentration of dust. Thus, in this study, the mass fraction of dust is larger than 90% compared to BC and OC (see the Supplementary data).

We have revised this sentence in the main text (Page 5, Line 20-21).

**Table 2.** Mean values of $\delta^{18}O$ (‰), major ions (ng/g), and microparticles ($10^3$ /mL) in studied snowpits.

| Sites | $\delta^{18}O$ | Na$^+$ | NH$_4^+$ | K$^+$ | Mg$^{2+}$ | Ca$^{2+}$ | Cl$^-$ | SO$_4^{2-}$ | NO$_3^-$ | Microparticles |
|---|---|---|---|---|---|---|---|---|---|---|
| TS | −8.84 | 140 | 213 | 59.4 | 194 | 1159 | 256 | 593 | 359 | 894 |
| LH | −11.94 | 149 | 163 | 30.9 | 345 | 1855 | 364 | 547 | 356 | 740 |
| MS | −17.97 | 75.9 | 118 | 40.0 | 49.4 | 506 | 110 | 109 | 154 | 197 |
| GL | −10.67 | 176 | 142 | 31. 8 | 85.0 | 1186 | 263 | 376 | 291 | 170 |
| ZD | −17.14 | 352 | 210 | 146 | 76.1 | 1658 | 363 | 444 | 256 | 562 |
| ER | −18.53 | 327 | 163 | 108 | 27.5 | 194 | 392 | 419 | 109 | 36.5 |
| DML | −17.12 | 14.7 | 19.6 | 20.6 | 30.5 | 129 | 30.5 | 29.2 | 52.9 | 205 |
| YL | −12.00 | 207 | 229 | 65.2 | 53.5 | 1003 | 245 | 459 | 220 | 116 |

References:

Xu, J., Ho u, S., Qin, D., Kaspari, S., Mayewski, P.A., Petit, J. R., Delmonte, B., Kang, S., Ren, J., Chappellaz, J., Hong, S.: A 108.83-m ice-core record of atmospheric dust deposition at Mt. Qomolangma (Everest), Central Himalaya, Quaternary Research, 73: 33−38, 2010.

Zhang, Y., Kang, S., Zhang, Q., Gao, T., Guo, J., Grigholm, B., Huang, J., Sillanpää, M., Li, X., Du, W., Li, Y., Ge, X.: Chemical records in snowpits from high altitude glaciers in the Tibetan Plateau and its surroundings, PLoS ONE, 11(5), e0155232, doi:10.1371/journal.pone.0155232, 2016.

- Page 5 line 24: which is it, 580 or 80?  Why one or the other?

**Answer:** In the protocol of IMPROVE-A for analysis of OC and EC, different temperature plateaus were used to obtain the signals of OC or EC on the filters. Thus in the main text, we have revised this sentence (Page 5, Line28-31).

- Page 5 line 27: How was the carbon converted to methane?

**Answer:** Fig. R1 of DRI carbon analyzer block diagram shows the analyzer operates by: (1) liberating carbon compounds under different temperature and oxidation environments from a small sample punch taken from a quartz-filter, (2) converting these compounds to carbon dioxide ($CO_2$) by passing the volatilized compounds through an oxidizer ($MnO_2$ at 912 °C), (3) reduction of the $CO_2$ to methane ($CH_4$) by passing the flow through a methanator (firebrick impregnated with nickel catalyst at ~550 °C in a stream of hydrogen), and (4) quantification of $CH_4$ by a flame ionization detector (FID).
We have revised this sentence in the main text (Page 6, Line 1-3).

[Figure]

Fig. R1 DRI thermal/optical carbon analyzer block diagram. (Chow et al., 2004)

References:
Chow, J.C., Watson J. G., Pritchett, L. C., Pierson, W.R., Frazier, C A., Purcell, R.G.: The DRI thermal/optical reflectance carbon analysis system: description, evaluation and applications in U.S. air quality studies. Atmos. Environ., 27(8): 1185−1201, 2004.

- Page 5 line 31: Is this the range for the detection limit?  750 seems high.

**Answer:** The value of 750 µg C cm$^{-2}$ is the detection range (maximum). The detection limit of total carbon is 0.93 µg C cm$^{-2}$. Detection limit of total organic carbon is 0.82 µg C cm$^{-2}$, and the total elemental carbon is 0.19 µg C cm$^{-2}$.

- Page 6 line 17: need units for the numbers 0 and 75

**Answer:** The unit is "°N". We have added in the main text. (Page 6, Line 22)

- Page 7 line 9: What is "standard solar irradiance"? Does this mean you don't have observations of surface irradiance? Remove "to get the albedos".

**Answer:** In this study, we don't have observation of surface irradiance. Thus, we use "reference/standard solar irradiance", which is the power per unit area received from the Sun in the form of electromagnetic radiation in the wavelength range of the measuring instrument. Irradiance on the Earth's surface depends on the tilt of the measuring surface, the height of the sun above the horizon, and atmospheric conditions. Absorptance, reflectance, and transmittance of solar energy are important factors in material degradation studies, solar thermal system performance, solar photovoltaic system performance, biological studies, and solar simulation activities. These optical properties are normally functions of wavelength, which require the spectral distribution of the solar flux be known before the solar-weighted property can be calculated. To compare the relative performance of competitive products, or to compare the performance of products before and after being subjected to weathering or other exposure conditions, a reference standard solar spectral distribution is desirable.

We have deleted the words "to get the albedos".

- Page 7 line 22-23: provide units for all variables.

**Answer:** We have added the units in the main text (Page 7, Line 28-30).

- Page 7 line 31: Is incoming shortwave radiation the same as surface irradiance?

**Answer:** The "incoming shortwave radiation" in this study is observed from the AWS near the sampling site. It is not the same as "surface irradiance".

-
- Page 8 line 1: How is "solar shortwave insolation" different from "incoming shortwave radiation"?

**Answer:** They are the same, in the main text, we have revised.

- Page 8 line 8: Is the observed snow depth based on previous studies? If so reference them here.

**Answer:** Yes. We have added the reference in the main text (Page 8, Line 14).

- Page 8 lines 6-9: Include units in parentheses for all variables in the text where they are described.

**Answer:** We have added the units in the main text. (Page 8, Line 12-14)

- Page 8 line 32: Have higher OC/EC ratios than what? Fossil fuels?

**Answer:** Have higher OC/EC ratios than that from fossil fuels combustion. We have changed this sentence in the main text (Page 9, Line 9).

- Page 9 line 20: "for half of the BC deposition" How was "half" determined, quantitatively? Same with 30% on line 22 and 70% on line 26.

**Answer:** Here in the study means about 50 %. We have changed the sentence in the main text (Page 9, Line 30).

- Page 10 line 11: What is the upper boundary and how does it compare to your grain size measurements?

**Answer:** In this study, the upper boundary (10 μm) is referred to the **dust size** in the SNICAR model (Fig. R2). It is not referred to the snow grain size.

[Figure]

Fig. R2 The input parameters of SNICAR model online. (Flanner et al., 2007)

References:

Flanner, M. G., Zender, C. S., Randerson, J. T., Rasch, P. J: Present-day climate forcing and response from black carbon in snow. J. Geophys. Res., 112, D11202, 2007.

- Page 10 line 7: insert    to be more explicit what these numbers are referring to.
  **Answer:** from wavelength of 350 to 800 nm. We have added information in the main text (Page 10, Line 16).

- Page 10 line 22: What do you mean by "radiative flux"? Can you express in terms of previously used variables, such as RF or albedo?
  **Answer:** Here it means the "radiative forcing".

RF represents the "radiative forcing" in this study. It means the difference between insolation (sunlight) absorbed by the snow surface and energy radiated back to space in this study.

In IPCC AR5 (Myhre et al., 2013), RF means the change in net downward radiative flux at the tropopause after allowing for stratospheric temperatures to readjust to radiative equilibrium, while holding surface and tropospheric temperatures and state variables fixed at the unperturbed values.

In this study, snow albedo is defined as the ratio of irradiance reflected to the irradiance received by snow surface. Snow albedo is highly variable, ranging from as high as 0.9 for freshly fallen snow, to about 0.4 for melting snow, and as low as 0.2 for dirty snow.

References:

Myhre, G., D. Shindell, F.-M. Bréon, W. Collins, J. Fuglestvedt, J. Huang, D. Koch, J.-F. Lamarque, D. Lee, B. Mendoza, T. Nakajima, A. Robock, G. Stephens, T. Takemura and H. Zhang, 2013: Anthropogenic and Natural Radiative Forcing. *In*: Climate Change 2013: The Physical Science Basis. Contribution of Working Group I to the Fifth Assessment Report of the Intergovernmental Panel on Climate Change [Stocker, T.F., D. Qin, G.-K. Plattner, M. Tignor, S.K. Allen,  J. Boschung, A. Nauels, Y. Xia, V. Bex and P.M. Midgley (eds.)]. Cambridge University Press, Cambridge, United Kingdom and New York, NY, USA.

- Page 11 line 15: What is shortwave radiation "input data"? Is it surface irradiance?  You need to be consistent in your defined terms throughout the manuscript.
  **Answer:** Here is the shortwave radiation.

- Page 12 line 18: What are Koch snowflakes?
  **Answer:** Koch snowflakes is one of snow grain types (Fig. R3b) (Liou et al., 2014; von Koch, 1904).
  A Koch snowflake, which has the fractal dimension of 1.262, takes place only on side planes. However, the two basal planes are flat. Thus, a 3-D Koch snowflake contains two flat basal planes and $3 \times 4n$ highly irregular side planes (with six-side symmetry) associated with n fractal iterations (see Figure 1b, where n = 4) (Liou et al., 2014).

[Figure]

**Figure 1.** (a) Geometry for the internal inclusion of BC/dust particles inside a hexagonal plate (a convex-shaped particle) in the *xyz* coordinates where the numbers (1–12) denote the corner position of the hexagon defined by the diameter 2*a* in basal planes and the length 2*c* in the *c* axis. (b) A schematic diagram for a basal plane of a Koch snowflake of the order 4 in which the points $P_i$, ($i = 1, 2,$ and 3) are determined by random numbers, and $Q_i$ are points on the boundary of the snowflake, and $Q_{ij}$ denotes subset points of $Q_i$ to represent the concave condition of a Koch snowflake such that selected random points (BC/dust) are within it.

Fig. R3 Snow gain type of Koch snowflake cited from (Liou et al., 2014).

- Page 13 line 11: "Snow cover durations were shortened during the melt season from X-Y days…" Insert numbers for X and Y.

   **Answer:** Snow cover durations were shortened during the melt season from 1.26−9.4 days…
   We have added the number of days in the main text (Page 13, Line 27).

- Page 13 line 13: "annual mass budget" Is this a loss of mass?  "mass budget" does not necessarily imply a loss.

   **Answer:** Yes, it is "annual mass loss". We have changed in the main text (Page 13, Line 29).

- Page 14 line 2: Wasn't this study a survey of LAP, and not snow cover?

   **Answer:** It is a survey of LAP in snow cover. We have changed in the main text (Page 14, Line 18).

- Page 14 line 13: What is radiation input data? Is this observed surface irradiance?

   **Answer:** It is shortwave radiation. We have deleted "input data".

**Although the grammar and wording are somewhat improved, there are still some issues.  Some specific suggestions on word changes are below which should help to clarify the manuscript.**

**Answer:** Thank you very much for all the suggestions and comments. We have carefully changed in the main text.

- Page 1 line 20: insert "relative" before "biomass burning"
**Done**

- Page 1 line 21: insert "relative" before "contribution"
**Done**

- Page 1 line 23: delete "changes of"
**Done**

- Page 2 line 10: change "by" to "of"

**Done**

- Page 2 line 16: replace "the simulation showed" with "suggested"

**Done**

- Page 2 line 17: deposited *to* land snow, not in Page 2 line 17: put a comma after W m$^{-2}$

**Done**

- Page 2 line 17: replace "or as large as" with "contributing as much as"

**Done**

- Page 2 line 19: replace "in" with "for the"

**Done**

- Page 2 line 20: insert "also" between "can" and "change"

**Done**

- Page 2 line 23: replace "in particular, from" with "especially     = "

**Done**

- Page 2 line 24: replace "on Claridenfirn of" with "in"

**Done**

- Page 2 line 26: remove comma after "region"       Page 2 line 27: what is "disturbed" desert dust?

**Done**

- Page 2 line 29: remove "even"

**Done**

- Page 3 line 22: replace "seldom" with "sparse"

**Done**

- Page 3 line 26: remove "cover" after "snow"

**Done**

- Page 3 line 27: change first comma to a period to end the sentence. Then start the next sentence with "We further…"

**Done**

- Page 3 line 28: insert "observed" before LAPs.

**Done**

- Page 3 line 28-29: should read "We use back trajectory analysis coupled with BC fire emission inventories to approximate natural/anthropogenic contributions"

**Done**

- Page 3 line 29: increase *our* understanding

**Done**

- Page 3 line 30: insert comma after TP and remove the next word "and"

**Done**

- Page 3 line 30: replace "this is also helpful for" with "inform"

**Done**

- Page 4 line 5: remove "in the earth system"

**Done**

- Page 4 line 17: remove "on"

**Done**

- Page 4 line 27: replace "contains" with "includes"

**Done**

- Page 5 line 23: remove "special"

**Done**

- Page 5 line 25: Should read "The sample is reheated further in a stepwise fashion to near 900…"

**Done**

- Page 5 line 26: Should read "burn out all remaining BC"

**Done**

- Page 5 lines 27-30: Move this sentence up to after "…of the OC in the sample." Change "modified .. arranged" to "limited the initial temperature plateau"

**Done**

- Page 6 line 6: Should read: "…used approach for determining source regions of various atmospheric species."

**Done**

- Page 6 line 7: remove "that"

**Done**

- Page 6 line 7: can be *qualitatively* attributed

**Done**

- Page 6 line 19: Remove "Then,"

**Done**

- Page 6 line 23: Should read "FINN used satellite" (remove "the")

**Done**

- Page 6 line 25: Remove "Then,"

**Done**

- Page 6 lines 29-30: Should read: "Note that this analysis is qualitative and does not take into account loss from wet and dry deposition"

**Done**

- Page 6 lines 31-32: Remove sentence beginning "Thus,"
**Done**

- Page 6 line 32: Should read "Despite these uncertainties, the relative differences between the BC contributions are used to…"
**Done**

- Page 7 line 2: remove second comma
**Done**

- Page 7 line 25: Should read "will lead to discrepancies between"
**Done**

- Page 8 line 6: insert "and" before "for"
**Done**

- Page 8 line 7: should read "clean snow due to changing snow grain sizes with show age"
**Done**

- Page 8 line 28: replace "values" with "concentrations"
**Done**

- Page 8 line 29: replace "by the soil" with "from the soil"
**Done**

- Page 8 line 30: replace "represent" with "examine"
**Done**

- Page 9 line 3: replace "little" with "slightly"
**Done**

- Page 9 line 13: remove "arrived at"
**Done**

- Page 10 line 2: replace "simulate" with "estimate the"
**Done**

- Page 10 line 13: Should read "albedo differences between measurements and simulations is less"
**Done**

- Page 10 line 20: remove "when testing regional to global scale models"
**Done**

- Page 11 line 5: Replace "validation" with "assessment". It is impossible to "validate" a model (although I realize this term is frequently used in the literature)
**Done**

- Page 11 line 11: Should read "Thus, this does doesn't include the effect of OC on estimates of"
**Done**

- Page 11 line 14: Changes *in* snow cover

**Done**

- Page 11 line 24: low *SD* scenarios

**Done**

- Page 12 line 19: A lack of OC *consideration* due to

**Done**

- Page 12 line 25: Remove "We also have to pay attention to the fact that".  Replace "their" with "dust"

**Done**

- Page 12 line 26: replace designed with "assumed".

**Done**

- Page 12 line 27: efficient *at* light scattering

**Done**

- Page 12 line 28: replace "materials that" with "and"

**Done**

- Page 13 line 8: replace "the estimation" with "estimates"

**Done**

- Page 13 line 30-31: "would have been marginal" depends on assumed concentrations.  Maybe say "uncertain" instead of "marginal"

**Done**

- Page 14 line 14: reduced *snow cover* from several

**Done**

- Page 14 line 19-20: "total mass balance" should read "mass lost" if I'm understanding this correctly

**Yes, it is "mass lost"**

---

## Editor Decision (ED2)

Dear Professor Kang,

Thank you for your responses and revisions. I have a few grammatical corrections in the new text which are detailed below. There are also 3 comments which didn't seem to be addressed in the manuscript. Please see details below and submit your responses and revisions.

Regards,

Becky

**Grammar or missing information:**

- Page 5 line 5: should read "density was measuring by weighing a specific volume of snow"
- Page 6 line 31: "dose nor included biofuel use and trash burning" should read "does not include biofuel use or trash burning"
- Page 12 line 23: Remove "as these"
- Page 12 line 24-25: Replace sentence beginning with "However" with "It is difficult to sample during the ablation season due to poor accessibility" or some other grammatically correct sentence.
- Page 12 line 25-26: Is this true for all seasons or a specific season? Indicate which season(s).
- Page 12 lines 26-27: I think you mean "Discontinuous", not "Discontinued". Put "the" before "melting". Replace "be a problem" with "also make it difficult"
- Page 12 lines 27-28: Remove this sentence. It doesn't make sense and it seems unnecessary.
- Page 12 line 31: Replace "tended to mix" with "is mixed"

**Responses**:

- My comment starting with "Page 7 lines 24-28". This paragraph should be added to the manuscript. Replace "consider" with "considered" and remove the last sentence.
- My comment starting with Page 8 line 8: I don't see any references on page 8 line 14.
- My comment starting with Page 9 line 20: I understood that half meant 50%. My question was how did you determine these percentages (50, 30, and 70%)?

---

## Author Response (AR3)

Jan04, 2017

Dear editor,

Thank you very much for your time and efforts regarding our manuscript (***tc-2017-111: Black carbon and mineral dust in snow cover on the Tibetan Plateau***). We have carefully revised the manuscript according to the comments. Detailed responses are in blue in the main text. We also included the revised version below with changes shown in blue. We hope that you find the revisions appropriate and adequate.

Yours sincerely,

Yulan ZHANG and Shichang KANG
* * *
**Grammar or missing information:**

1. Page 5 line 5: should read "density was measuring by weighing a specific volume of snow"

Answer: We have changed in the main text.

2. Page 6 line 31: "dose nor included biofuel use and trash burning" should read "does not include biofuel use or trash burning"

Answer: Done

3. Page 12 line 23: Remove "as these"

Answer: Done

4. Page 12 line 24-25: Replace sentence beginning with "However" with "It is difficult to sample during the ablation season due to poor accessibility" or some other grammatically correct sentence.

Answer: Done

5. Page 12 line 25-26: Is this true for all seasons or a specific season? Indicate which season(s).

Answer: In the study by Xu et al. (2017), the sentence "considerable heterogeneity in the topography and climate has led to complex spatial and temporal snow cover patterns" means for all seasons. Their results show a very weak negative trend for the snow depth and the number of snow-cover days in spring and winter from 1961 to 2010, but two different trends were found: an initial increase followed by a decrease; In summer and autumn, snow depth and the number of snow-cover days show a significant decreasing trend for most sites.

Here in our study, we want to indicate the snow cover across the Tibetan Plateau may be discontinuous during ablation season.

6. Page 12 lines 26-27: I think you mean "Discontinuous", not "Discontinued". Put "the" before "melting". Replace "be a problem" with "also make it difficult"

Answer: Done

7.  Page 12 lines 27-28: Remove this sentence. It doesn't make sense and it seems unnecessary.

Answer: Done

8.  Page 12 line 31: Replace "tended to mix" with "is mixed"

Answer: Done

**Responses:**

1.  My comment starting with "Page 7 lines 24-28". This paragraph should be added to the manuscript. Replace "consider" with "considered" and remove the last sentence.

[Comment: Page 7 lines 24-28: The lines in blue do not answer the reviewer's question. Did you assume uniform BC throughout the snow depth? It looks like that is the only option in the inputs. If so, what sort of error does this represent?]

Answer: From Flanner et al (2007, 2009)'s study and personal communication with Prof. Flanner, the SNICAR model online is only a single snow layer, so the BC concentration is necessarily uniform with depth. With the model code, and in the CLM model, however, multiple model layers are specified and hence vertically-heterogeneous BC concentrations can be specified or simulated. In this study, we assumed uniform BC throughout the snow depth. The error in not doing so depends entirely on the environmental conditions. In conditions of strong melt where hydrophobic impurities accumulated near the top, the concentrations can vary strongly with depth.
We have revised in the main text (Page7 Line29-32, and Page8, Line1-3).

References:

Flanner, M. G., Zender, C. S., Hess, P. G., Mahowald, N. M., Painter, T. H., Ramanathan, V., Rasch, P. J.: Springtime warming and reduced snow cover from carbonaceous particles, Atmos. Chem. Phys., 9, 2481–2497, 2009.
Flanner, M. G., Zender, C. S., Randerson, J. T., Rasch, P. J: Present-day climate forcing and response from black carbon in snow. J. Geophys. Res., 112, D11202, 2007.

2.  My comment starting with Page 8 line 8: I don't see any references on page 8 line 14.

[Comment: Page 8 line 8: Is the observed snow depth based on previous studies? If so reference them here.]

Answer: The observed snow depth used were from this study (supplementary Table S1) and previous studies (Che et al., 2012; Xu et al., 2017; Zhong et al., 2016). We have added the related references in the main text.

References:

Che, T., Dai, L., Wang, J., Zhao, K., Liu, Q.: Estimation of snow depth and snow water equivalent distribution using airborne microwave radiometry in the Binggou Watershed, the uppwe reaches of the Heihe River basin, Int. J. Appl. Earth Obs. Info., 17, 23–32, doi:10.1016/j.jag.2011.10.014, 2012.
Xu, W., Ma, L., Ma, M., Zhang, H., Yuan, W.: Spatial-Temperal variability of snow cover and depth in the Qinghai-Tibetan Plateau, J. Clim., 30, 1521–1533, doi:10.1175/JCLI-D-15-0732.1, 2017.
Zhong, X., Zhang, T., Zheng, L., Hu, Y., Wang, H., Kang, S.: Spatiotemporal variability of snow depth aross the Eurasian continent from 1966-2012. Cryosphere Diss., doi:5194/tc-2016-182, 2016.

3. My comment starting with Page 9 line 20: I understood that half meant 50%. My question was how did you determine these percentages (50, 30, and 70%)?

Answer: I am so sorry for the misunderstanding. So shown in the Figure 5 in the main text, we calculated the different biomass contributions to the target site for different hours before they arrived at the site (Table R1). Considering the impact of air mass transportation, we calculated the average contributions based on 0, -24, and -48 hrs.

[Figure]

**Fig. 5** Different source contributions to BC deposition on the snow cover across the Tibetan Plateau.

Table S1 Biomass contributions (%) of BC deposition to the target sites over the Tibetan Plateau.

| Time (h) | MYL | SETP | NMC | TGL | NETP | LHG |
|---|---|---|---|---|---|---|
| 0 | 83% | 78% | 17% | 3% | 2% | 3% |
| -24 | 40% | 50% | 43% | 37% | 19% | 18% |
| -48 | 27% | 37% | 33% | 32% | 32% | 20% |
| Average contributions of biomass burning | 50% | 55% | 31% | 24% | 18% | 14% |